molecul|ar
systems
biology

# Adaptive resistance of melanoma cells to RAF inhibition via reversible induction of a slowly dividing de-differentiated state

Mohammad Fallahi-Sichani[1],* (iD), Verena Becker[1], Benjamin Izar[2,3], Gregory J Baker[1], Jia-Ren Lin[4], Sarah A Boswell[1] (iD), Parin Shah[2] (iD), Asaf Rotem[2] (iD), Levi A Garraway[2,3,5] & Peter K Sorger[1,4,5],** (iD)

## Abstract

Treatment of *BRAF*-mutant melanomas with MAP kinase pathway inhibitors is paradigmatic of the promise of precision cancer therapy but also highlights problems with drug resistance that limit patient benefit. We use live-cell imaging, single-cell analysis, and molecular profiling to show that exposure of tumor cells to RAF/MEK inhibitors elicits a heterogeneous response in which some cells die, some arrest, and the remainder adapt to drug. Drug-adapted cells up-regulate markers of the neural crest (e.g., NGFR), a melanocyte precursor, and grow slowly. This phenotype is transiently stable, reverting to the drug-naïve state within 9 days of drug withdrawal. Transcriptional profiling of cell lines and human tumors implicates a c-Jun/ECM/FAK/Src cascade in de-differentiation in about one-third of cell lines studied; drug-induced changes in c-Jun and NGFR levels are also observed in xenograft and human tumors. Drugs targeting the c-Jun/ECM/FAK/Src cascade as well as BET bromodomain inhibitors increase the maximum effect ($E_{max}$) of RAF/MEK kinase inhibitors by promoting cell killing. Thus, analysis of reversible drug resistance at a single-cell level identifies signaling pathways and inhibitory drugs missed by assays that focus on cell populations.

**Keywords** adaptive and reversible drug resistance; *BRAF^V600E* melanomas; de-differentiated NGFR^High state; RAF and MEK inhibitors
**Subject Categories** Molecular Biology of Disease; Quantitative Biology & Dynamical Systems; Signal Transduction
**Mol Syst Biol. (2017) 13: 905**

## Introduction

Small-molecule inhibitors of MAP kinases (MAPK), such as RAF inhibitors (e.g., vemurafenib and dabrafenib), MEK inhibitors (e.g., selumetinib and trametinib), or their combination, benefit a majority of melanoma patients whose tumors carry activating V600E/K mutations in the *BRAF* oncogene, but they commonly fail to cure disease due to acquired resistance. Acquired resistance has been shown to involve a diversity of oncogenic mutations in components of the MAPK pathway (Nazarian *et al*, 2010; Poulikakos *et al*, 2011; Wagle *et al*, 2011, 2014; Villanueva *et al*, 2013; Long *et al*, 2014; Van Allen *et al*, 2014; Moriceau *et al*, 2015) or parallel signaling networks such as the PI3K/AKT kinase cascade (Shi *et al*, 2014a,b). In some cases, however, the emergence of drug-resistant clones cannot be fully explained by known genetic mechanisms (Hugo *et al*, 2015). It is thought that genetically distinct, fully drug-resistant clones arise from tumor cells that survive the initial phases of therapy due to drug adaptation (or tolerance) (Emmons *et al*, 2016). Reversible (non-genetic) drug adaptation can be reproduced in cultured cells, and combination therapies that block adaptive mechanisms *in vitro* have shown promise in improving rates and durability of response (Lito *et al*, 2013). Thus, better understanding of mechanisms involved in drug adaptation is likely to improve the effectiveness of melanoma therapy by delaying or controlling acquired resistance.

Adaptation to RAF inhibitors involves cell-autonomous changes such as up-regulation or rewiring of mitogenic signaling cascades as well as non-cell-autonomous changes in the microenvironment such as paracrine signaling from stromal cells (Gopal *et al*, 2010; Lito *et al*, 2012; Abel *et al*, 2013; Hirata *et al*, 2015; Obenauf *et al*, 2015). Understanding these mechanisms is made more complex by variability in adaptive responses from one tumor cell line to the next (Fallahi-Sichani *et al*, 2015). Differences in early adaptive signaling (involving the PI3K/AKT, JNK/c-Jun, and NF-κB networks) exist even among *BRAF^V600E* cell lines with comparably high sensitivity to brief (3–4 days of) vemurafenib treatment (Fallahi-Sichani *et al*, 2015).

1  Department of Systems Biology, Program in Therapeutic Sciences, Harvard Medical School, Boston, MA, USA
2  Department of Medical Oncology, Dana–Farber Cancer Institute, Boston, MA, USA
3  Broad Institute of Harvard and MIT, Cambridge, MA, USA
4  HMS LINCS Center and Laboratory of Systems Pharmacology, Harvard Medical School, Boston, MA, USA
5  Ludwig Center at Harvard, Harvard Medical School, Boston, MA, USA
   *Corresponding author. Tel: +1 617 432 6907; E-mail: mohammad_fallahisichani@hms.harvard.edu
   **Corresponding author. Tel: +1 617 432 6901; E-mail: peter_sorger@hms.harvard.edu

There is growing evidence that a reversible drug-tolerant state associated with chromatin modifications (e.g., enhanced histone demethylase activity) can be induced in cancer cells following drug exposure (Sharma et al, 2010). In the case of BRAF-mutant melanomas, a comparable vemurafenib-tolerant state has been associated with changes in the expression of differentiation markers, including: MITF, a key regulator of melanocyte lineage; NGFR (the low affinity nerve growth factor receptor, also known as p75NTR or CD271), a neural crest marker; and receptor tyrosine kinases such as AXL, EGFR, and PDGFRβ (Johannessen et al, 2013; Konieczkowski et al, 2014; Muller et al, 2014; Sun et al, 2014; Ravindran Menon et al, 2015; Smith et al, 2016; Tirosh et al, 2016).

Despite a wealth of data on signaling networks involved in drug adaptation, most of our knowledge comes from studying bulk tumor cell populations. This makes it hard to determine whether proteins involved in adaptation are weakly active in all cells or highly active in a subset of cells. In addition, the phenotypic consequences of drug adaptation (e.g., the emergence of slowly proliferating cells) have primarily been studied using fixed time assays following 1–2 weeks of drug exposure, when drug-adapted cells exhibit high activity in multiple pro-growth signaling cascades (Ravindran Menon et al, 2015). It therefore remains unclear how the initial responses to drug relate to subsequent phenotypes such as cell death or adaptation. Continuous-time, single-cell assays are required to tease out these aspects of drug adaptation.

In this paper, we monitor the responses of $BRAF^{V600E}$ melanoma cells to vemurafenib in real time using live-cell imaging and then analyze the resulting cell states using molecular and phenotypic profiling. We find that vemurafenib-treated $BRAF^{V600E}$ cells exhibit a range of fates over the first 3–4 days of drug exposure; a subset of cells undergoes apoptosis, a second subset remains arrested in the G0/G1 phase of the cell cycle, and a third subset enters a slowly cycling drug-resistant state. The slowly cycling resistant state is maintained when cells are grown in the presence of drug, but it is reversible upon 9 days of outgrowth in medium lacking drug, resulting in the regeneration of a population of cells exhibiting the three behaviors of drug-naïve cells. We find that adaptive resistance is associated with de-differentiation along the melanocyte lineage and up-regulation of neural crest markers such as NGFR. These changes can also be detected in naïve and drug-treated patient-matched human tumors by RNA profiling and histopathology. We identify kinase inhibitors and epigenome modifiers (e.g., BET inhibitors) that appear to block acquisition of the slowly cycling NGFR[High] state in cell lines and in a $BRAF^{V600E}$ melanoma xenograft model and thereby increase sensitivity to vemurafenib. The data and methods used in this paper are freely available and formatted to interchange standards established by the NIH LINCS project (http://www.lincsproject.org/) to promote reuse and enhance reproducibility.

# Results

## Live-cell imaging and single-cell analysis uncover a slowly cycling drug-resistant state involved in adaptation to RAF inhibitors

To study the dynamics of $BRAF^{V600E}$ inhibition in melanoma cells, we performed live-cell imaging on two vemurafenib-sensitive cell lines at concentrations near the IC$_{50}$ for cell killing (COLO858 and

MMACSF; IC$_{50}$ ~0.1–0.5 μM; we subsequently expanded the analysis to additional lines, as described below). The cells expressed a dual cell cycle reporter (Tyson et al, 2012), comprising (i) mCherry-geminin, a protein that is absent during G0/G1, accumulates during S/G2/M, and disappears rapidly late in M phase concurrent with cytokinesis and birth of daughter cells (Sakaue-Sawano et al, 2008), and (ii) H2B-Venus, which labels chromatin, allowing mitotic chromosome condensation and disintegration of nuclei during apoptosis to be scored morphologically (Fig 1A and B). Within 24 h of exposure to 1 μM vemurafenib, COLO858 and MMACSF cells were observed to accumulate in the G0/G1 phase of the cell cycle (with low mCherry-geminin levels and decondensed chromatin; Fig 1C and Movie EV1). Between $t = 24$ and 48 h, ~50% of COLO858 and ~40% of MMACSF cells underwent apoptosis and other cells in the population stayed arrested in G0/G1 (Fig 1C–E).

Subsequently, however, the fates of the two cell lines diverged: MMACSF cells remained arrested, but ~20% of COLO858 cells re-entered S phase between 48 and 84 h (highlighted in pink in Fig 1B and C). Single-cell traces of these adapted cells showed that they underwent division every ~65 h, as compared to doubling time of ~24 h for DMSO-treated COLO858 cells; the additional cell division time was spent in G0/G1 (Fig 1F and Appendix Fig S1). Thus, COLO858 cells exhibited a mixed response to vemurafenib within the first 84 h of exposure to 1 μM vemurafenib with ~60% of cells undergoing apoptosis (primarily between 24 and 48 h of drug exposure), ~20% arresting in the G0/G1 phase of the cell cycle, and ~20% exiting arrest and entering a slowly dividing state.

Exposure of COLO858 cells to 1 μM vemurafenib for an additional ~4 days (for a total of ~8 days) revealed that cells continued to divide slowly in the presence of drug. Approximately half of the cells tracked from day 4 to 8 did not divide at all and the other half divided only once, yielding an estimated doubling time of 48–96 h (Fig 1G). The ratio of cycling versus non-cycling cells in the population appeared to be ~1:1 throughout this period. Thus, slow proliferation and occasional generation of non-dividing cells represent a durable state for COLO858 cells exposed to vemurafenib.

## Generation of dividing adapted cells is not explained by MAPK pathway re-activation

Incomplete inhibition or re-activation of the MAPK pathway has been identified as a major cause of adaptive resistance to vemurafenib (Lito et al, 2012). This arises because MAPK signaling is a key regulator of proliferation in melanoma cells. We observed that exposure of COLO858 and MMACSF cells to vemurafenib or vemurafenib plus trametinib (a potent and selective inhibitor of MEK kinase) inhibited p-ERK[T202/Y204] levels by up to ~15-fold relative to untreated cells; p-ERK[T202/Y204] is a marker of MAPK pathway activity that can be scored by single-cell imaging. Levels of p-ERK[T202/Y204] in drug-treated cells did not significantly change during 24–72 h of treatment suggesting no recovery of MAPK signaling within this period (Fig 2A and B, and Appendix Fig S2A and B). We conclude that COLO858 cells that re-enter the cell cycle do not re-activate the MAPK pathway. To investigate this further, we co-stained cells for p-ERK[T202/Y204] and the proliferation markers p-Rb[S807/811] and Ki-67. These epitopes are present at high levels during S/G2/M, but absent in G0/G1 phases of the cell cycle. No significant difference in p-ERK[T202/Y204] levels was observed between

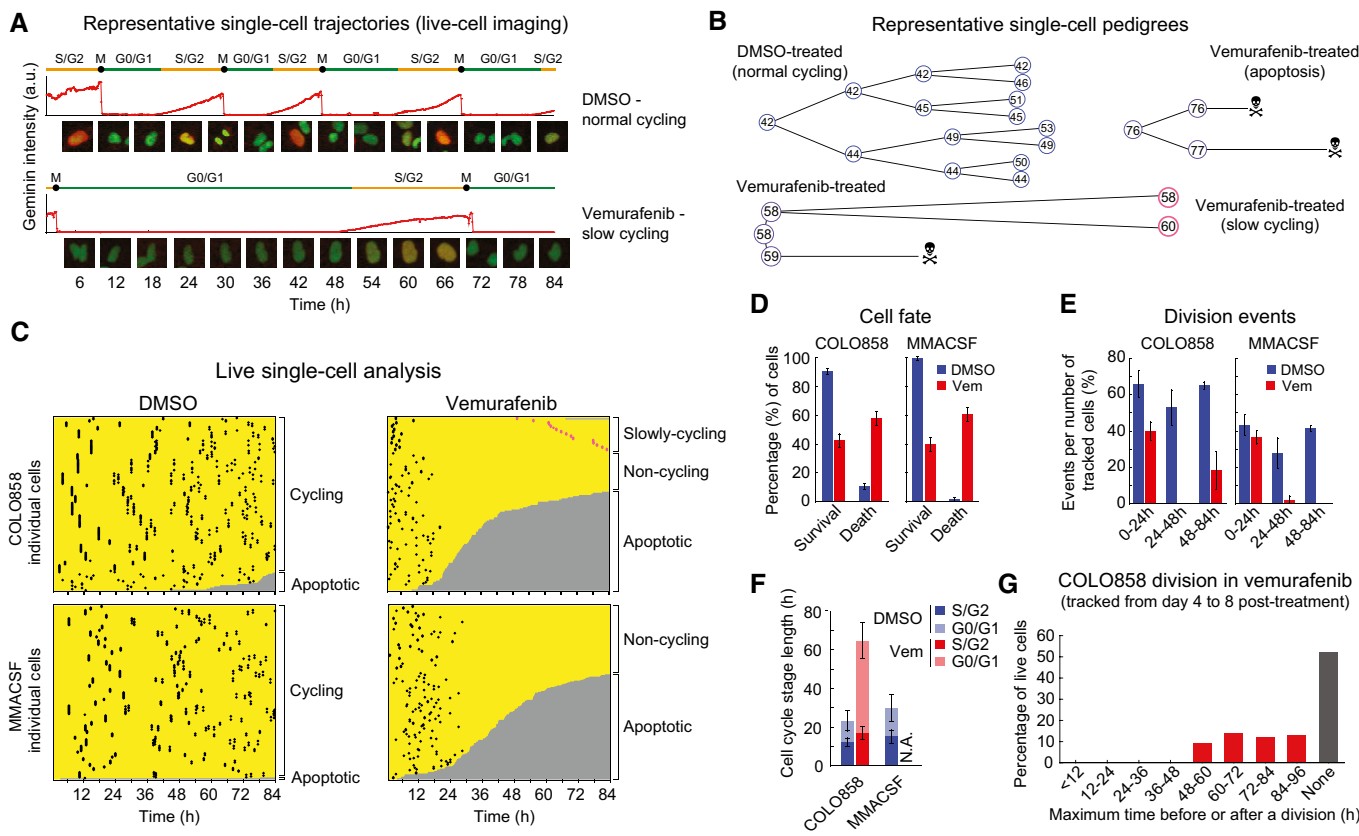

**Figure 1.  Live-cell imaging uncovers a slowly cycling drug-resistant state involved in adaptation to RAF inhibition.**
Time-lapse imaging of COLO858 and MMACSF cells stably expressing H2B-Venus and mCherry-geminin exposed to 1 μM vemurafenib or DMSO for 4–8 days.

A, B    Representative images and cell cycle phases (A) and representative maps of cell lineage (B) are depicted for COLO858 under DMSO and vemurafenib conditions.

C    Single-cell analysis of division and death events. Horizontal axes represent single-cell tracks with time. Division events are displayed as black or pink (in the case of slowly cycling cells) dots. Transition from yellow to gray indicates cell death.

D    Percentage of surviving and dead cells among cells tracked for 84 h.

E    Percentage of division events among cells tracked during indicated time intervals.

F    Length of different cell cycle phases (G0/G1 and S/G2) in cells tracked for 84 h. No data are reported for MMACSF-vemurafenib because all cells stopped dividing ~24 h after treatment and no single cell divided more than once.

G    Division times for COLO858 cells tracked between days 4 and 8 post-treatment with 1 μM vemurafenib. Minimum doubling times were estimated for 100 individual cells by identifying the longest time interval before or after which a cell divides.

Data information: Data in (D–F) are presented as mean ± SD using 3–4 groups of cells imaged from multiple wells (see Materials and Methods).

drug-treated COLO858 cells in G0/G1 (cells that scored as p-Rb$^{Low}$ or Ki-67$^{Low}$) and cells that had re-entered cell division and were in S/G2/M (cells that scored as p-Rb$^{High}$ or Ki-67$^{High}$) (Fig 2C and D, and Appendix Fig S2C). We conclude that the re-entry of a subset of vemurafenib-treated COLO858 cells into the cell cycle does not require detectable re-activation of the MAPK pathway, suggesting an adaptation that makes MAPK signaling less essential for proliferation.

**Adaptation reduces drug maximal effect and the incremental benefit of raising the drug dose**

To better understand how the presence of slowly dividing, drug-adapted cells influences vemurafenib responsiveness at a population level, as conventionally assayed, we performed sequential drug treatment with RAF and MEK inhibitors and then counted surviving cells (Fig 3A). First, COLO858 and MMACSF cells were exposed for

24 h to a dose of vemurafenib below the IC$_{50}$ (0.01–0.32 μM) with the goal of inducing adaptation but minimizing cell death. 1 μM vemurafenib was then added without changing media, and cells were grown for a further 72 h prior to counting the number of viable and apoptotic cells. This protocol resulted in a final total drug concentration of 1.01–1.32 μM, which is above the IC$_{50}$; treatment of cells with DMSO served as a control. We found that pre-treatment of COLO858 cells with vemurafenib reduced cell killing by a second bolus of drug in a dose-dependent manner (Fig 3B): A maximal 2.5-fold increase in the number of surviving cells was observed following 0.1 μM drug pre-treatment (relative to DMSO pre-treated cells). No such effect was observed in MMACSF cells, in which cell viability fell monotonically with increasing total drug concentration. When the concentration of vemurafenib in the second bolus of drug was varied across a range of 0–5 μM, as a means of analyzing dose–response relationships, we observed a significant reduction in E$_{max}$ (the fraction of cells arrested or killed at maximum drug dose) and a

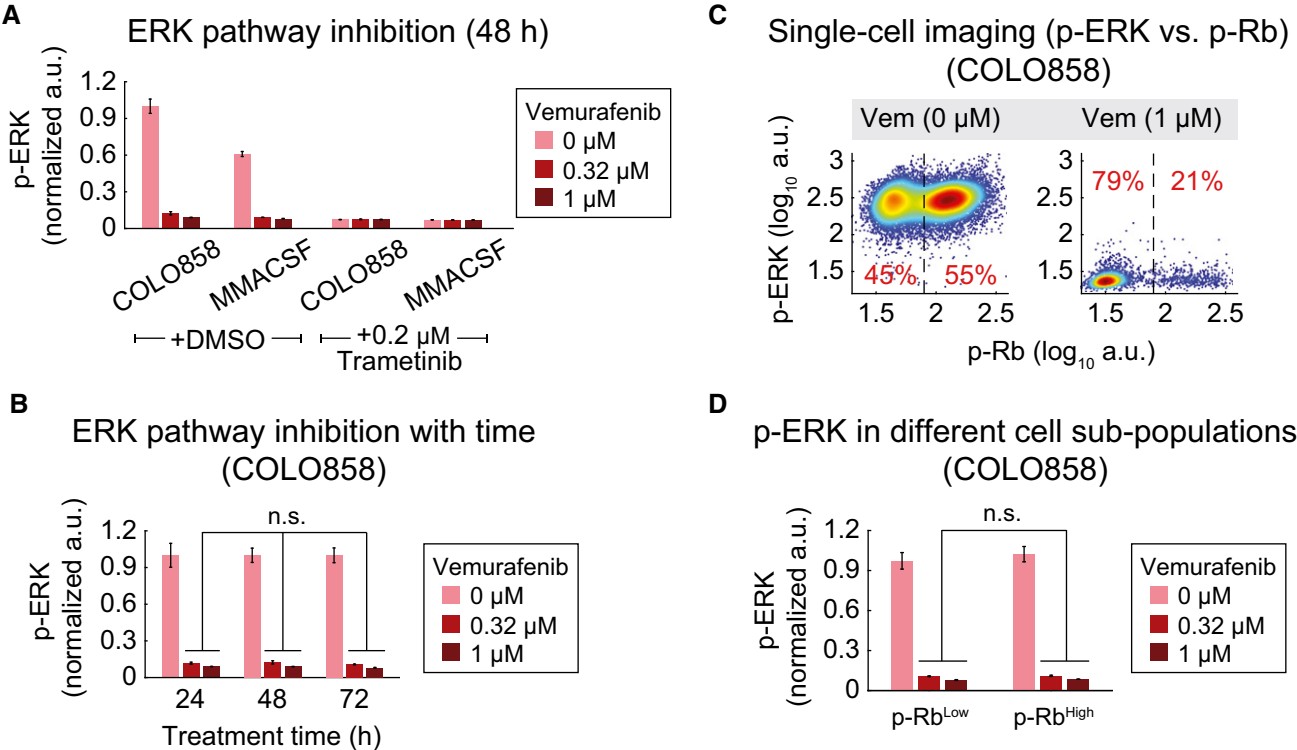

**Figure 2. Drug adaptation is not explained by MAPK pathway re-activation.**

A   p-ERK$^{T202/Y204}$ levels as measured in duplicate by immunofluorescence in COLO858 and MMACSF cells treated for 48 h with vemurafenib in combination with DMSO or trametinib at indicated doses.

B   p-ERK$^{T202/Y204}$ variation with time (24, 48, 72 h) in COLO858 cells treated in duplicate with vemurafenib at indicated doses.

C   Covariate single-cell analysis of p-Rb$^{S807/811}$ versus p-ERK in COLO858 cells 72 h after exposure to indicated doses of vemurafenib. Vertical dashed lines were used to gate p-Rb$^{High}$ versus p-Rb$^{Low}$ cells.

D   Mean p-ERK levels in p-Rb$^{High}$ versus p-Rb$^{Low}$ subpopulations of COLO858 cells treated in duplicate with indicated doses of vemurafenib for 72 h.

Data information: Data in (A, B, D) are presented as mean ± SD and are normalized to DMSO-treated COLO858 cells at each time point. Statistical significance was determined by two-way ANOVA.

---

decrease in Hill slope for COLO858 but not MMACSF cells (Fig EV1A and B). Sequential dosing experiments were also repeated using cells pre-treated with a 1:1 molar ratio of vemurafenib plus trametinib. Pre-treatment of COLO858 with RAF/MEK inhibitor combination also led to subsequent resistance (Figs 3B and EV1C and D). We conclude that pre-treatment of COLO858 with sublethal doses of RAF inhibitor or a RAF/MEK inhibitor combination has a significant effect on subsequent drug response primarily by lowering maximal effect ($E_{max}$) and reducing the incremental effect of rising drug concentration (Hill slope).

**Drug-adapted, slowly cycling cells up-regulate genes associated with a de-differentiated NGFR$^{High}$ state**

To identify genes associated with acquisition of the slowly cycling, vemurafenib-adapted state, we performed RNA sequencing (RNA-seq) on COLO858 cells exposed to drug for 24 and 48 h; drug-treated MMACSF cells served as a control. Genes differentially expressed in COLO858 or MMACSF cells relative to DMSO-treated controls were selected based on a statistical cutoff of $q < 0.01$. Among these genes, we focused on the subset differing in degree of enrichment by twofold or more between the two cell lines (see

Materials and Methods). Genes enriched in vemurafenib-treated COLO858 cells relative to MMACSF cells comprised 479 up- and 646 down-regulated genes at 24 h and 853 up- and 713 down-regulated genes at 48 h (Fig 4A and Dataset EV1). The top GO terms included neural differentiation, neurogenesis, and cytoskeleton regulation (Fig 4B and Dataset EV2): Genes involved in these processes were enriched in COLO858 cells and reduced or unchanged in MMACSF cells. For example, mRNA for NGFR (UniProtKB: P08138), a neural crest marker, increased ~15-fold in vemurafenib-treated COLO858 cells, representing one of the highest fold-changes in the dataset ($q = 3 \times 10^{-4}$); in contrast, NGFR fell ~fourfold in MMACSF cells ($q = 5 \times 10^{-4}$). It has previously been show that cells expressing NGFR represent an intermediate in the process by which melanocytes differentiate from neural crest cells (Mica *et al*, 2013) and NGFR is used clinically as a histopathological marker to distinguish desmoplastic melanomas, tumors that are negative for conventional melanocytic markers, from other skin neoplasms (Lazova *et al*, 2010). In addition to promoting NGFR expression in COLO858 cells, vemurafenib exposure led to up-regulation of neurogenesis genes such as *S100B, CNTN6, L1CAM, FYN, MAP2,* and *NCAM1*, further evidence that cells acquire a less differentiated, more neural crest-like state (Fig 4A). The expression of genes involved in cell cycle

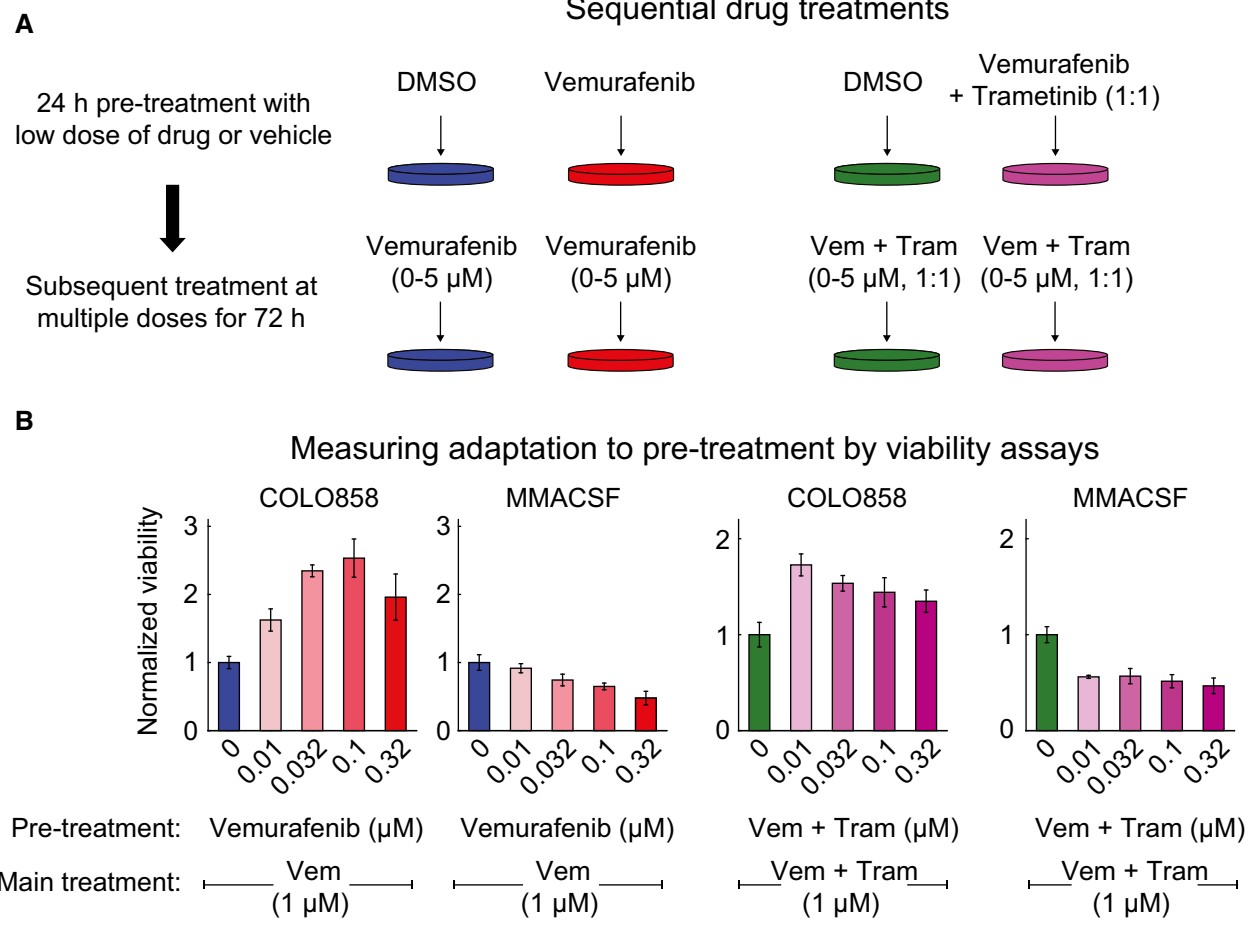

**Figure 3. Sequential drug treatments reveal adaptive resistance to RAF and MEK inhibitors.**

A   Schematic outline of a two-stage drug treatment experiment to measure the impact of drug adaptation on the response of COLO858 and MMACSF cells to RAF/MEK inhibitors.

B   Viability of cells pre-treated with DMSO, multiple doses of vemurafenib, or vemurafenib plus trametinib, and then treated with an additional 1 μM vemurafenib or vemurafenib plus trametinib (without change of media). Cell viability measured in four replicates was normalized to the viability of cells pre-treated with DMSO. Data are presented as mean ± SD.

progression also changed upon drug exposure: In both COLO858 and MMACSF cells, cell cycle genes were down-regulated by 24 h, but in COLO858, they rose again to their original levels by 48 h (Fig 4A and B), consistent with live-cell imaging data showing that COLO858 cells transiently arrest and then re-enter the cell cycle.

To follow changes in NGFR protein levels in control and drug-treated cells, we co-stained for NGFR and the proliferation marker Ki-67; NGFR protein levels were low in drug-naïve COLO858 cells and increased up to ~sevenfold by 48 h and ~25-fold by 72 h of vemurafenib exposure, consistent with mRNA data. In contrast, NGFR levels fell in MMACSF cells following 48–72 h in drug (Fig 4C). Time-course studies of COLO858 cells helped to reveal how drug-adapted NGFR[High] cells arose. Within 24 h of vemurafenib treatment, the population of cells shifted from a largely (> 80%) proliferative Ki-67[High]/NGFR[Low] state to a non-mitotic Ki-67[Low]/NGFR[Low] state (Fig 4D). Non-mitotic cells then up-regulated NGFR, acquiring a Ki-67[Low]/NGFR[High] state by $t = 48$ h, after which they gradually re-entered the cell cycle. By $t = 72$ h, > 90% of cells in the population were NGFR[High]. Among these NGFR[High] cells, ~40%

eventually became Ki-67[High], showing that they had begun to proliferate. We conclude that a subset of cells exposed to vemurafenib transiently exits the cell cycle and induces an adaptive response that makes them drug-resistant and NGFR[High]; cells subsequently re-enter the cell cycle and proliferate slowly.

### Vemurafenib-induced de-differentiation of cells and adaptive resistance are reversible upon drug removal

To determine whether the NGFR[High], drug-adapted state is reversible, we exposed COLO858 cells to vemurafenib (at 0.32 μM) for 48 h and then isolated NGFR[High] and NGFR[Low] cells by fluorescence-activated cell sorting (FACS). These cell populations differed in NGFR levels ~fourfold on average (Fig 5A and Appendix Fig S3). FACS-sorted cells were allowed to grow in fresh medium, and samples were fixed every 24 h and the levels of NGFR and Ki-67 expression measured by immunofluorescence (Fig 5B). We observed that NGFR levels progressively increased in the NGFR[Low] pool and fell in the NGFR[High] pool, so that by day 9 average receptor

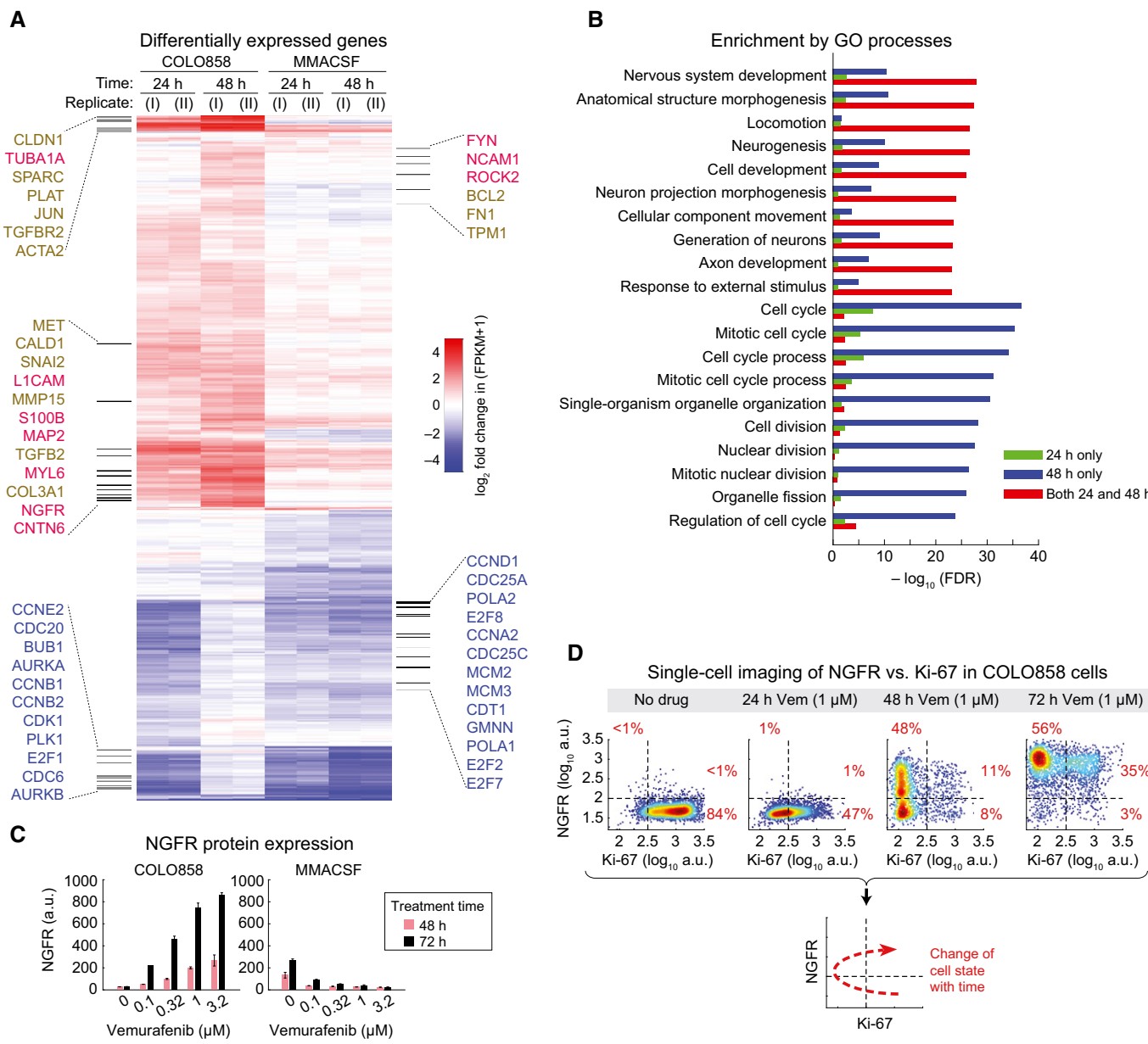

**Figure 4.  Drug resistance is associated with de-differentiation of cells to a slowly cycling NGFR<sup>High</sup> phenotype.**

A   Differentially up-regulated genes in COLO858 relative to MMACSF cells treated with 0.2 μM vemurafenib for 24 and 48 h ($\log_2$ (ratio) ≥ 1). Selected genes involved in neurogenesis, neural differentiation and myelination (red), cell adhesion, ECM remodeling and epithelial–mesenchymal transition (brown), and cell cycle regulation (blue) are highlighted.

B   Top Gene Ontology (GO) biological processes differentially regulated between COLO858 and MMACSF cells.

C   NGFR protein levels measured in duplicate by immunofluorescence in COLO858 and MMACSF cells treated with indicated doses of vemurafenib for 48 or 72 h. Data are presented as mean ± SD.

D   Covariate single-cell analysis of Ki-67 versus NGFR in COLO858 cells 24–72 h after exposure to 1 μM vemurafenib or DMSO.

expression levels were indistinguishable in the two pools of cells. Return of NGFR levels to pre-treatment levels was accompanied by an increase in Ki-67 staining showing that rapid proliferation had resumed.

To measure vemurafenib sensitivity, NGFR<sup>High</sup> and NGFR<sup>Low</sup> cells were exposed to drug, 2 days after sorting (the time required for cells to completely re-adhere), and drug response was measured using a conventional 3-day assay and analyzed using a recently

developed "growth rate inhibition" (GR) metric that corrects for differences in cell proliferation rates (Hafner *et al*, 2016). We observed that NGFR<sup>High</sup> cells were significantly less sensitive to vemurafenib in comparison with NGFR<sup>Low</sup> cells ($P = 6 \times 10^{-5}$) (Fig 5C). In contrast, when cells from NGFR<sup>High</sup> and NGFR<sup>Low</sup> pools were allowed to grow for 9 days in the absence of drug, responsiveness to vemurafenib was indistinguishable (Fig 5D). Moreover, when proliferation rates were scored in freshly isolated NGFR<sup>High</sup> cells

(over a 3-day period), the average cell doubling time was ~32 h as compared to ~18 h for cells in the NGFR$^{Low}$ pool. (Fig 5E). Finally, when cells that had undergone one cycle of vemurafenib-induced NGFR up-regulation were allowed to reset to the pre-treatment state by outgrowth in the absence of drug and then re-exposed to vemurafenib for 48 h, NGFR was up-regulated to the same degree as in drug-naïve cells (Fig 5F).

These experiments demonstrate that the subset of vemurafenib-treated COLO858 cells able to acquire a slowly dividing NGFR$^{High}$

phenotype is more drug-resistant than the subset of cells in the same initial population that remains NGFR$^{Low}$. As expected, the magnitude of the difference in drug resistance and growth rate observed in studies of FACS-sorted cells was smaller than in live-cell imaging experiments. This is because analysis of sorted cells involves waiting for cells to re-adhere in the absence of drug; during this period, the adapted phenotype relaxes to the pre-treatment state. In contrast, in live-cell studies, cells are continuously exposed to drug and the adapted phenotype is maintained. Overall, we

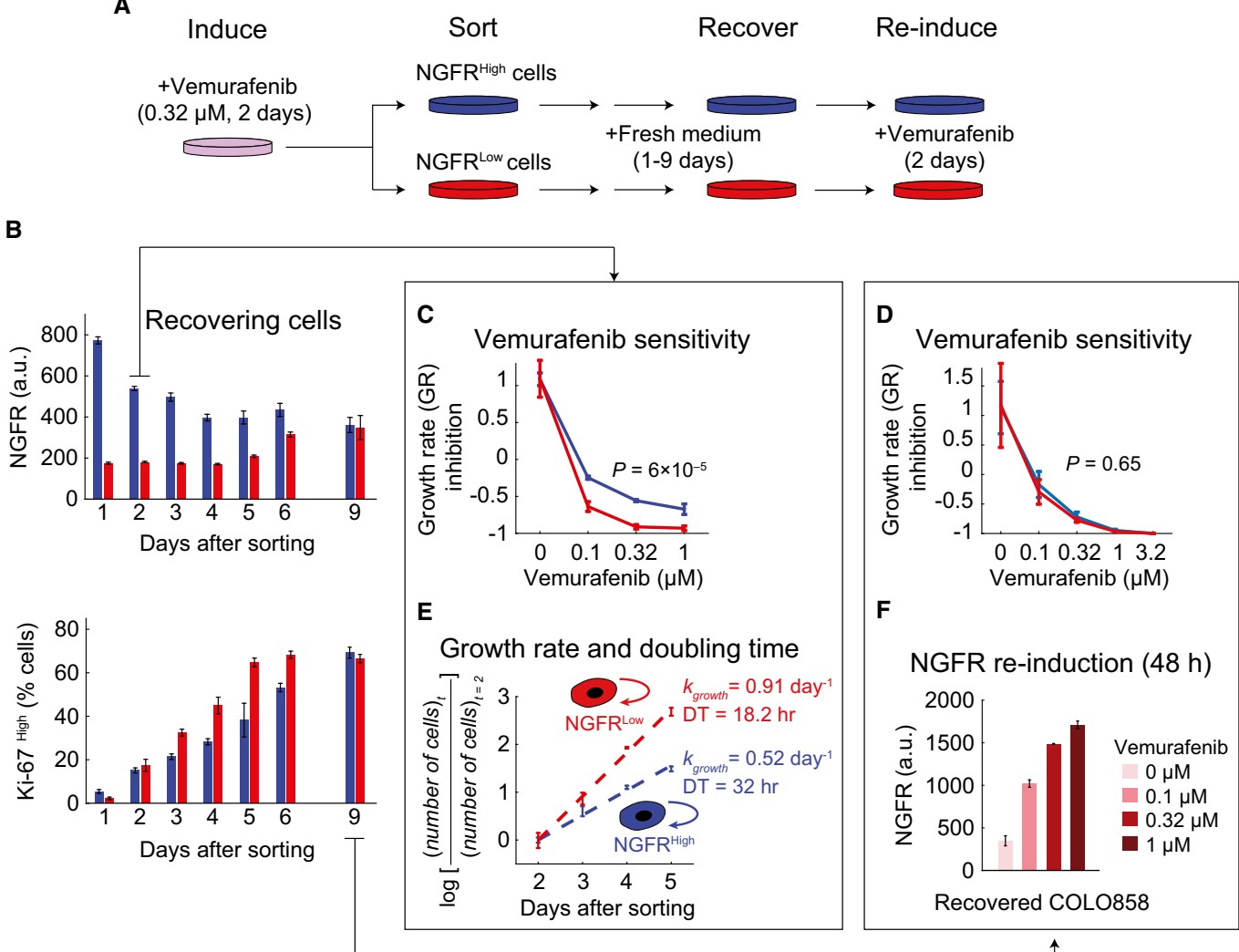

**Figure 5.  Vemurafenib-induced de-differentiation of cells and adaptive resistance are reversible upon drug removal.**

A    Schematic outline of an experiment involving induction of the slowly cycling NGFR$^{High}$ state in COLO858 cells following 48-h treatment with 0.32 μM vemurafenib, sorting cells to obtain NGFR$^{Low}$ and NGFR$^{High}$ subpopulations, recovering each cell subpopulation in fresh growth medium for 1–9 days, and re-inducing recovered cells with vemurafenib.

B    NGFR and Ki-67 protein levels measured by immunofluorescence in cells grown for 9 days in fresh medium ($n = 4$).

C, D    Growth rate (GR) inhibition assay performed on FACS-sorted NGFR$^{High}$ and NGFR$^{Low}$ pools of cells after 2 (C) or 9 (D) days of outgrowth in fresh medium. Measurements were performed in 4 (C) or 6 (D) replicates.

E    Growth rate and doubling time measurements in 4 replicates in FACS-sorted NGFR$^{High}$ and NGFR$^{Low}$ cells during 2–5 days of outgrowth in fresh medium.

F    NGFR levels measured in duplicate by immunofluorescence in COLO858 cells recovered after 9 days of outgrowth in fresh media and subsequently re-exposed for 48 h to four doses of vemurafenib.

Data information: Data in (B–F) are presented as mean ± SD. Statistical significance was determined by two-way ANOVA.

conclude that the vemurafenib-induced, slowly cycling, NGFR^High state is transiently stable allowing NGFR^High and NGFR^Low cells to inter-convert on a time scale of about a week in culture. Such behavior is inconsistent with a genetic difference between the two populations of cells, but similar to the transiently heritable cell-to-cell variability previously shown to play a role in cellular response to pro-apoptotic ligands (Flusberg *et al*, 2013) and other small-molecule drugs (Cohen *et al*, 2008; Sharma *et al*, 2010).

### Induction of an NGFR^High state involves extracellular matrix (ECM) components, focal adhesion, and the AP1 transcription factor c-Jun

To identify biochemical pathways involved in NGFR up-regulation, we performed pathway enrichment analysis on genes differentially regulated in vemurafenib-treated COLO858 and MMACSF cells. Cell

adhesion, ECM remodeling, and epithelial-to-mesenchymal transition (EMT) were among the top enriched pathways (Fig EV2 and Dataset EV2). Genes up-regulated in vemurafenib-treated COLO858 cells included the ECM components thrombospondin-1 (THBS1; TSP-1; UniProtKB: P07996), an adhesive glycoprotein that mediates cell–cell and cell–ECM interactions, the laminin subunits LAMA1 and LAMC1 (UniProtKB: P25391 and P11047), CCN signaling protein NOV (Perbal, 2004) (UniProtKB: P48745), and several integrin family receptors (Fig 6A and B). Gene set enrichment analysis (GSEA) showed that similar molecules and pathways accompany increased NGFR expression in 25 *BRAF^V600E* melanoma cell lines found in the Cancer Cell Line Encyclopedia (CCLE) and 128 *BRAF^V600E* melanoma biopsies in The Cancer Genome Atlas (TCGA) (Fig 6C).

To identify potential transcriptional regulators of genes up-regulated in the NGFR^High state, we used DAVID (http://

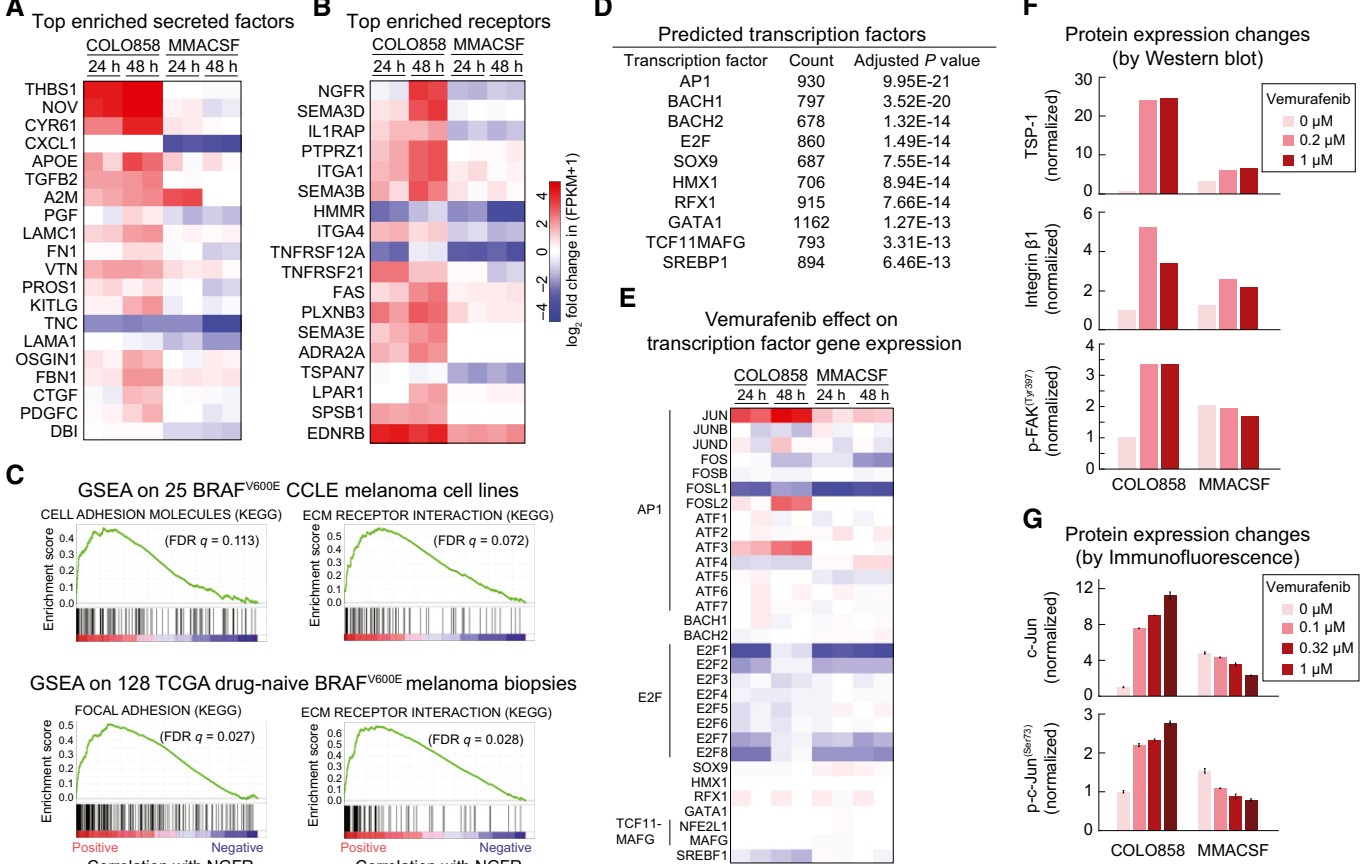

**Figure 6.** The NGFR^High state involves extracellular matrix (ECM) components, focal adhesion, and the AP1 transcription factor c-Jun.

A, B    Top differentially regulated genes encoding secreted proteins (A) and cell surface receptors (B) between COLO858 and MMACSF cells.

C    Ranked GSEA plots of top KEGG pathways significantly correlated with NGFR expression in 25 *BRAF^V600E* melanoma cell lines from the CCLE (top) and tumor biopsies of 128 *BRAF^V600E* melanoma patients in TCGA (bottom).

D, E    A list of transcription factor candidates predicted (by DAVID; see Materials and Methods) to regulate differentially expressed genes between vemurafenib-treated COLO858 and MMACSF cells (D), and the corresponding transcription factor gene expression levels in these cells (E).

F    Quantified Western blot measurements (see Materials and Methods) for thrombospondin-1 (THBS1; TSP-1), integrin β1, and p-FAK^Y397 in COLO858 and MMACSF cells treated for 48 h with indicated doses of vemurafenib. Data are first normalized to HSP90α/β levels in each cell line at each treatment condition and then to DMSO-treated COLO858 cells.

G    c-Jun and p-c-Jun^S73 changes as measured in duplicate by immunofluorescence in COLO858 and MMACSF cells treated for 48 h with indicated doses of vemurafenib. Data are normalized to DMSO-treated COLO858 cells.

Data information: Data in (F, G) are presented as mean ± SD.

david.abcc.ncifcrf.gov) (Fig 6D) and then examined expression levels for the top 10 transcription factor candidates (Fig 6E). DAVID identified the AP1 family of transcription factors as the top candidates for regulators of the adapted state in COLO858 cells ($P \approx 10^{-20}$) (Fig 6D). Moreover, *JUN*—which encodes the AP1 transcription factor c-Jun—was the most differentially enriched candidate: In COLO858, it was up-regulated ~ninefold within 24 h and ~23-fold within 48 h of exposure to vemurafenib, but changed at most ~twofold in MMACSF cells within 48 h of treatment (Fig 6E). When we focused DAVID and differential expression analysis specifically on receptors and secreted/ECM proteins, AP1 factors and *JUN* were again predicted to be key differential regulators of vemurafenib response in COLO858 and MMACSF cells (Fig EV3A).

To investigate the involvement of ECM proteins and receptors in drug adaptation, we performed Western blotting on extracts from COLO858 and MMACSF cells, focusing on the subset of genes for which antibodies are available: TSP-1, integrin β1 (a subunit of a cell adhesion receptor that binds to TSP-1), and an activating phosphorylation site on the focal adhesion kinase (p-FAK$^{Y397}$). Following 48 h in 0.2 or 1 μM vemurafenib, p-FAK levels (normalized to the levels of HSP90α/β) increased ~3.5-fold in COLO858 cells relative to DMSO-treated cells, whereas they fell slightly (~25%) in MMACSF cells (Figs 6F and EV3B). TSP-1 levels (also normalized to HSP90α/β levels) increased by ~25-fold in COLO858 cells but only ~twofold in MMACSF cells. Integrin β1 was induced > fivefold in COLO858 cells, but only ~twofold in MMACSF cells. Both c-Jun and p-c-Jun$^{S73}$ (the active state of the protein) were substantially elevated by vemurafenib treatment (up to ~12-fold and ~threefold, respectively) in COLO858 cells but down-regulated (by ~50%) in MMACSF cells (Fig 6G). Thus, differences detected at the level of mRNA were reflected in the levels and activities of the corresponding proteins.

To obtain functional data on proteins implicated in the drug-adapted state, we depleted *JUN* or *PTK2* (the FAK gene) in COLO858 cells by siRNA. Depletion of either gene significantly reduced vemurafenib-induced NGFR up-regulation (by ~70%) and increased sensitivity to vemurafenib as compared to cells transfected with control siRNA (Fig 7A). NGFR knockdown, however, did not reduce cell viability nor did exposure of cells to NGF, an NGFR ligand (Fig EV3C). Thus, NGFR appears to be a marker of the vemurafenib-resistant cell state rather than a regulator of drug resistance. In contrast, siRNA experiments directly implicate c-Jun and FAK in drug adaptation.

### Concurrent inhibition of RAF/MEK signaling and a putative c-Jun/FAK/Src cascade overcomes vemurafenib resistance in NGFR$^{High}$ cells

To begin to identify signaling proteins involved in acquisition of an NGFR$^{High}$, drug-adapted state, we exposed COLO858 cells to vemurafenib in combination with a range of small-molecule kinase inhibitors, including defactinib and PF562271, two compounds that target FAK; JNK-IN-8, a selective inhibitor of c-Jun N-terminal kinases (JNK); and dasatinib and saracatinib, two inhibitors of Src family non-receptor tyrosine kinases that function downstream of FAK (and other receptor tyrosine kinases); see Fig 7 for details on drug dosing and nominal target information. When COLO858 cells

were treated with these drugs and vemurafenib at 0.32 or 1 μM for 48 h, induction of NGFR was reduced from threefold to fivefold to less than 1.5-fold (and in some cases to levels below those of drug-naïve cells; Fig 7B). In contrast, exposing COLO858 cells to trametinib plus vemurafenib enhanced NGFR induction as compared to vemurafenib alone (Fig 7B). Thus, kinases targeted by the approved drug dasatinib (Sprycel®), investigational drugs defactinib, and saracatinib as well as tool compounds PF562271 and JNK-IN-8 are involved in vemurafenib-induced NGFR up-regulation. Super-induction of NGFR by vemurafenib plus trametinib shows that MAPK signaling is a negative regulator of this process (Appendix Fig S4A).

To better understand the effects of kinase inhibitors on vemurafenib adaptation, we measured NGFR and Ki-67 in single cells by imaging. Assays were performed across doses, and data were z-scored and visualized as a 2D landscape (Fig 7C). This showed that the effects of JNK, FAK, and Src inhibitors in vemurafenib-treated cells (green arrow) were orthogonal to the effects of trametinib (red arrow): Whereas the fraction of NGFR$^{High}$ cells rose as the fraction of Ki-67$^{High}$ cells fell with trametinib, with JNK, FAK, or Src inhibitors both NGFR and Ki-67 were suppressed (Fig 7C). Co-drugging COLO858 cells with trametinib reduced p-ERK$^{T202/Y204}$ levels, but co-drugging with JNK, FAK, or Src inhibitors had no effect on p-ERK$^{T202/Y204}$ levels relative to vemurafenib alone (Fig EV4). We conclude that whereas trametinib acts to enhance both the therapeutic effect of vemurafenib (i.e., inhibition of proliferation) and its counter-therapeutic effect (i.e., induction of the NGFR$^{High}$ state), JNK, FAK, or Src inhibitors have orthogonal activities that reduce drug adaptation.

The biological significance of these findings was confirmed by dose–response studies showing that combining vemurafenib with JNK, FAK, or Src inhibitors increased killing of COLO858 but not MMACSF cells (Fig 7D and Appendix Fig S4B). Moreover, the effectiveness of JNK, FAK, or Src inhibitors rose when MAPK signaling was more fully inhibited by a combination of vemurafenib and trametinib (Fig 7D; right panel). The primary effect of combining MAPK inhibitors with drugs such as dasatinib or saracatinib was an increase in $E_{max}$ and a reduction in the fraction of surviving cells. In Fig 7D, log–log drug dose–response plots highlight that co-drugging primarily affected the 1–10% of *BRAF*$^{V600E}$ cells that survived MAPK inhibitors; such differences were much less obvious when using log-linear dose–response plots conventionally used to score interaction. We conclude that drugs targeting kinases that lie within pathways identified by gene sent enrichment analysis as up-regulated in vemurafenib-treated COLO858 cells substantially increase cell killing.

### A screen identifies chromatin modifications as additional contributors to the NGFR$^{High}$ state

The acquisition of transiently heritable states in drug-treated melanoma cells is reminiscent of reversible drug tolerance previously shown to involve chromatin modifications sensitive to HDAC inhibition (Sharma *et al*, 2010; Ravindran Menon *et al*, 2015). We therefore performed a focused screen with a library of small-molecule inhibitors of epigenome-modifying enzymes to identify those that reduced NGFR induction by vemurafenib. COLO858 cells were exposed for 48 h to a sublethal dose of

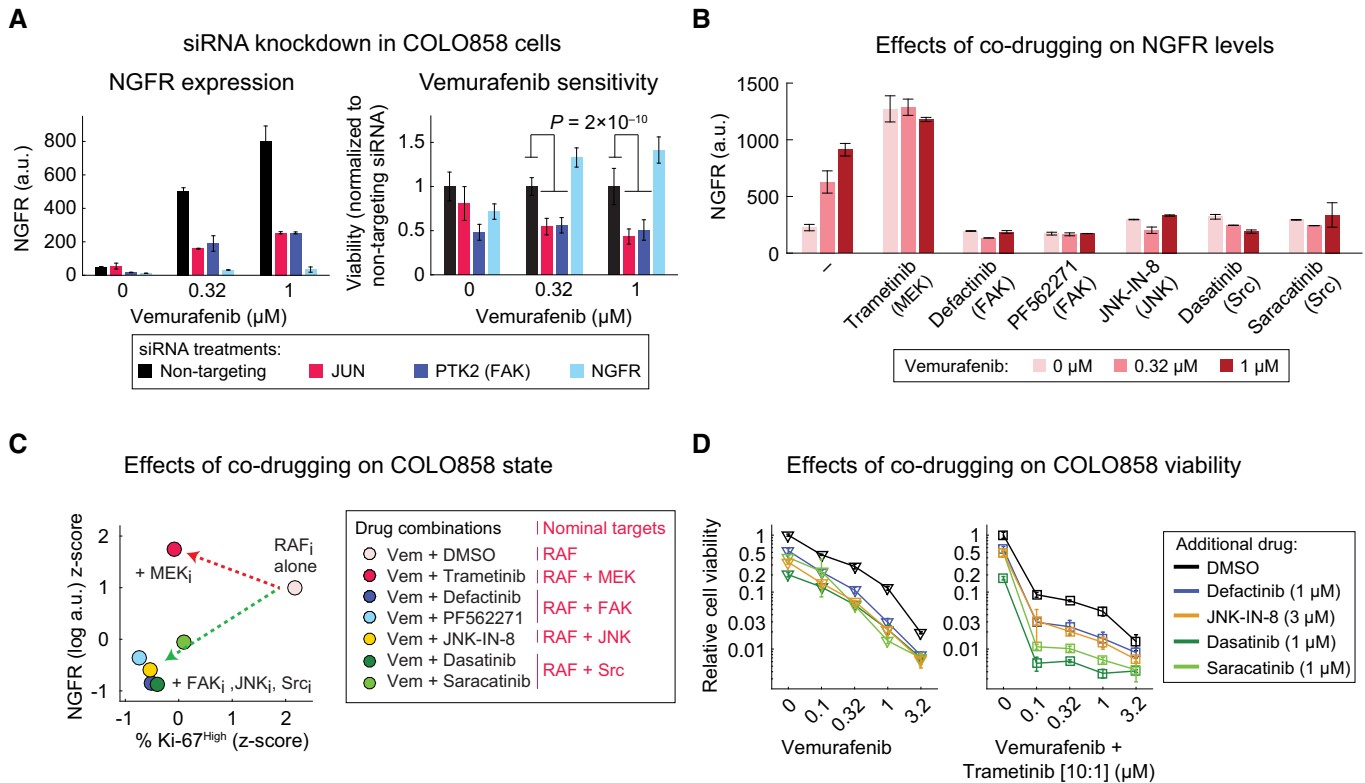

**Figure 7.** Concurrent inhibition of RAF/MEK signaling and the c-Jun/FAK/Src cascade blocks the NGFR$^{High}$ state and increases cell killing.

A   NGFR levels as measured by immunofluorescence (left panel) and relative cell viability (right panel) in COLO858 cells following treatment in duplicate with indicated doses of vemurafenib in the presence of siRNAs targeting *JUN*, *PTK2*, and *NGFR* for 72 h. Viability data for each siRNA condition at each dose of vemurafenib were normalized to cells treated with the same dose and the non-targeting siRNA.

B   NGFR protein levels measured by immunofluorescence in duplicate in COLO858 cells treated for 48 h with indicated doses of vemurafenib, in combination with DMSO, MEK inhibitor trametinib (0.6 μM), FAK inhibitors defactinib (3 μM) and PF562271 (3 μM), JNK inhibitor JNK-IN-8 (3 μM), or Src inhibitors dasatinib (3 μM) and saracatinib (3 μM).

C   Pairwise comparison between drug combination-induced changes in NGFR and Ki-67 in COLO858 cells treated for 48 h with vemurafenib at 0.32 and 1 μM in combination with DMSO or two doses of trametinib (0.2, 0.6 μM), defactinib (1, 3 μM), PF562271 (1, 3 μM), dasatinib (1, 3 μM), saracatinib (1, 3 μM), and JNK-IN-8 (1, 3 μM). NGFR and Ki-67 levels were measured by immunofluorescence. For each signal, data were averaged across two replicates, two doses of vemurafenib, and two doses of the second drug, log-transformed, and z-score-scaled across seven drug combinations.

D   Relative viability of COLO858 cells treated for 72 h with vemurafenib or vemurafenib plus trametinib (10:1 dose ratio) in combination with DMSO, JNK-IN-8, dasatinib, saracatinib, and defactinib at indicated doses. Viability data were measured in three replicates and normalized to DMSO-treated controls.

Data information: Data in (A, B, D) are presented as mean ± SD. Statistical significance was determined by two-way ANOVA.

vemurafenib (0.32 μM) in combination with one of 41 inhibitors of HDAC, BET, and other chromatin-targeting compounds at three doses (see Materials and Methods for a list of compounds); NGFR levels were measured by imaging (Fig 8A). Three BET bromodomain inhibitors, (+)-JQ1, I-BET, and I-BET151, were found to consistently suppress NGFR up-regulation when combined with vemurafenib in technical and biological replicates (Fig 8A and Appendix Fig S5A). All three compounds also reduced Ki-67 levels (Appendix Fig S5B) and increased killing of COLO858 cells when combined with vemurafenib, as measured by 3-day viability assays and evaluation of drug $E_{max}$ (Fig 8B; left panel). For example, when applied to COLO858 cells, 0.32 μM JQ1 slowed down cell division but was not measurably cytotoxic (Movie EV2) and no more apoptosis was detected than in a DMSO-only control (Fig EV5). However, a combination of 0.32 μM JQ1 and 1 μM vemurafenib increased apoptosis to

> 90% of cells (1 μM vemurafenib alone induced 40% apoptosis under these conditions; Fig EV5). Thus, JQ1 and vemurafenib are synergistic in cell killing by conventional Loewe criteria. JQ1 and the other BET inhibitors were even more effective in promoting cell killing when vemurafenib and trametinib were used in combination (Fig 8B; right panel). Moreover, on a plot of Ki-67 versus NGFR levels, the effects of BET inhibitors (green arrow) were orthogonal to those of trametinib (red arrow), a property shared with JNK, FAK, and Src inhibitors (Fig 8C). BET inhibitors reduced c-Jun up-regulation induced by vemurafenib to a significant degree (by an average of ~50%; $P < 2 \times 10^{-8}$) but did not fully block it (Fig 8D). From these data, we conclude that chromatin modifications are likely to be involved in the acquisition of the transiently heritable, NGFR$^{High}$, vemurafenib-resistant cell state and that multiple BET inhibitors can block this effect.

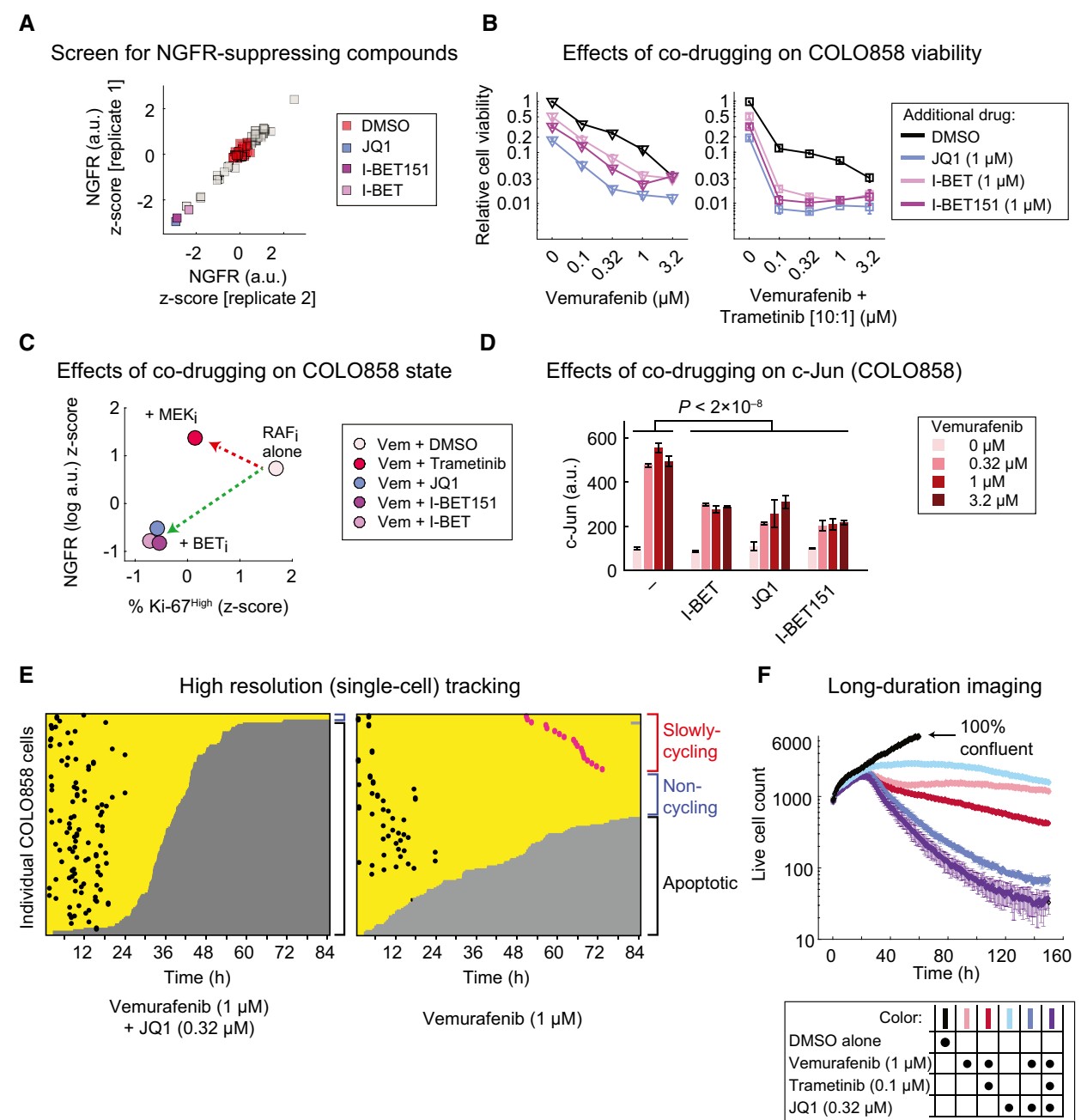

**Figure 8. BET inhibitors suppress the slowly cycling NGFR^High state and effectively reduce the cancer cell population with time.**

A   COLO858 cells were treated for 48 h in duplicate with vemurafenib (at 0.32 μM) in combination with DMSO or three doses (0.11, 0.53, and 2.67 μM) of each of 41 compounds in a chromatin-targeting library. NGFR protein levels were measured by immunofluorescence, averaged across three doses of each compound, and *z*-scored.

B   Relative viability of COLO858 cells treated for 72 h with vemurafenib or vemurafenib plus trametinib (10:1 dose ratio) in combination with DMSO, (+)-JQ1, I-BET, and I-BET151 at indicated doses. Viability data were measured in three replicates and normalized to DMSO-treated controls.

C   Pairwise comparison between drug-induced changes in NGFR and Ki-67 in COLO858 cells treated with vemurafenib at 0.32, 1, and 3.2 μM in combination with DMSO or trametinib (0.2 μM), I-BET (1 μM), I-BET151 (1 μM), and (+)-JQ1 (1 μM) for 48 h. Data for each drug combination were averaged across two replicates and three doses of vemurafenib, log-transformed, and *z*-score-scaled.

D   c-Jun protein levels measured by immunofluorescence in duplicate in COLO858 cells treated for 48 h with indicated doses of vemurafenib, in combination with DMSO, I-BET (1 μM), (+)-JQ1 (1 μM), and I-BET151 (1 μM).

E   Single-cell analysis of division and death events following live-cell imaging of COLO858 cells treated with 1 μM vemurafenib in combination with DMSO or (+)-JQ1 (0.32 μM) for 84 h. Data are presented as described in Figure 1.

F   Time-lapse analysis of COLO858 cells treated in three replicates for ~1 week with different drug combinations at indicated doses. Data for DMSO-treated cells are shown until day 3, the time at which cells reach ~100% confluency.

Data information: Data in (B, D, F) are presented as mean ± SD.

## NGFR-suppressing drug combinations block the emergence of slowly cycling cells and effectively reduce the cancer cell population with time

To link the activities of drugs that inhibit induction of NGFR by vemurafenib to the kinetics of cell killing, we performed live-cell imaging of COLO858 cells in the presence of 1 μM vemurafenib alone or in combination with 0.32 μM JQ1. Co-drugging eliminated the emergence of slowly cycling cells (pink) and increased the fraction of cells undergoing apoptosis to > 95% (Fig 8E and Appendix Fig S5C and D, and Movie EV3). We also monitored cell growth every 45 min for ~7 days using a live-cell microscope (an IncuCyte® Live Cell Analysis System) that is placed in an incubator and results in minimal perturbation of growth conditions. In vemurafenib-treated cells (Fig 8F; pink line), both cell division and cell death were observed between 0 and 30 h after which cell number was nearly constant. Co-drugging with trametinib increased cell killing (red), but by the end of one week, the number of viable cells was still ~50% of the initial number ($t = 0$). Exposure of cells to JQ1 alone induced cytostasis with little cell killing (light blue), whereas the combination of vemurafenib plus JQ1 was highly cytotoxic, resulting in continuous cell killing throughout the 7-day assay period (blue); the triple combination of vemurafenib, trametinib, and JQ1 was even more effective (purple). Live-cell analysis of COLO858 cells exposed to combinations of vemurafenib, trametinib, and the FAK inhibitor defactinib yielded comparable findings (Appendix Fig S5C–F). These data show that a drug identified on the basis of its ability to block acquisition of an NGFR[High] state also blocks the emergence of slowly growing, vemurafenib-adapted cells and, as a consequence, causes a sustained increase in the rate of cell killing.

## JNK, FAK, Src, and BET inhibitors overcome the NGFR[High] state in additional *BRAF[V600E/D]* melanoma lines

To investigate the generality of the biology described above, we analyzed seven additional *BRAF[V600E/D]* melanoma cell lines. In two of these lines (A375 and WM115), NGFR levels were high in the absence of vemurafenib but increased modestly in a dose-dependent manner following 48-h exposure to 0.1–1 μM vemurafenib (Fig 9A and Appendix Fig S6A–C). Concomitantly, these lines exhibited up to ~sevenfold dose-dependent increase in c-Jun levels (Appendix Fig S6D). The five other lines we examined exhibited no detectable increase in NGFR or c-Jun levels upon exposure to vemurafenib. These findings are consistent with previous data showing that

adaptation to vemurafenib is heterogeneous across cell lines (Fallahi-Sichani *et al*, 2015), but overall, a statistically significant and positive correlation was observed between vemurafenib-induced c-Jun and NGFR levels (Pearson's ρ = 0.86, *P* = 0.001) (Fig 9B).

When we measured the levels of TSP-1, integrin β1, and p-FAK[Y397] in A375 and WM115 cells, we observed vemurafenib-induced increases in expression and/or high basal levels, in contrast to low basal levels and an absence of induction in drug-treated NGFR[Low] MZ7MEL cells (Appendix Fig S6E and Fig EV3B). In common with COLO858 cells, co-drugging A375 and WM115 cell lines with JNK-IN-8, dasatinib, saracatinib, defactinib, and either vemurafenib or vemurafenib plus trametinib increased cell killing (and reduced $E_{max}$), but co-drugging had no significant effect on killing of MZ7MEL cells (Fig 9C and Appendix Fig S6F). When we repeated a focused screen for epigenome-targeting compounds in A375 and WM115 cells, we identified the same three BET inhibitors JQ1, I-BET, and I-BET151 as capable of blocking vemurafenib-induced NGFR up-regulation (Appendix Fig S7A–C). All three of these compounds enhanced cell killing when combined with vemurafenib or vemurafenib plus trametinib (Fig 9D). On a plot of NGFR versus Ki-67 levels, the effects of co-drugging A375 or WM115 cells with vemurafenib and inhibitors of BET proteins, JNK, FAK, or Src were orthogonal to those of co-drugging with trametinib, in all cases reducing the fraction of Ki-67[High] and NGFR[High] cells relative to vemurafenib alone but without further reducing p-ERK levels (Fig 9E and F, and Appendix Fig S7D and E). From these data, we conclude that even though basal NGFR levels vary significantly among COLO858, A375, and WM115 cells, all three lines exhibit similar drug adaptation in the presence of MAPK inhibitors.

## The NGFR[High] state is associated with resistance to MAPK inhibitors in some melanoma patients

When tumor biopsies from drug-naïve melanoma patients were immunostained for NGFR, we observed variability from one tumor to the next and, within a single tumor, from one region to the next: NGFR[High]/MITF[Low] and NGFR[Low]/MITF[High] domains were present in 4/11 samples and the former stained less strongly for Ki-67 (Fig 10A and Appendix Fig S8). We obtained biopsies from a patient prior to the onset of therapy, 2 weeks after initiation of therapy with dabrafenib plus trametinib and subsequent to relapse and then measured NGFR, Ki-67, and c-Jun levels by immunostaining. Relative to the pre-treatment biopsy, the on-treatment biopsy exhibited a reduction in the fraction of Ki-67[High] cells from ~23% to ~4%,

---

**Figure 9.  JNK, FAK, Src, and BET inhibitors overcome the NGFR[High] drug-resistant state in additional *BRAF[V600E/D]* melanoma lines.**

A   NGFR protein levels measured in duplicate by immunofluorescence in seven *BRAF[V600E/D]* cell lines treated with vemurafenib at indicated doses for 48 h.

B   Correlation between vemurafenib-induced changes in c-Jun and NGFR protein levels across nine *BRAF[V600E/D]* melanoma cell lines. Cells were treated with five doses of vemurafenib (0, 0.1, 0.32, 1, and 3.2 μM) for 48 h. c-Jun and NGFR protein levels measured by immunofluorescence at each condition were averaged across two replicates and normalized to DMSO-treated controls. The area under the dose–response curve (AUC) for the two measurements (c-Jun and NGFR) was calculated, *z*-score-scaled across nine cell lines, and their pairwise Pearson's correlation was reported.

C, D   Relative viability of A375 and WM115 cells treated in 3 replicates for 72 h with vemurafenib or vemurafenib plus trametinib (10:1 dose ratio) in combination with indicated kinase inhibitors (C) or BET inhibitors (D).

E, F   Pairwise comparison between NGFR and Ki-67 levels in A375 and WM115 cells treated with vemurafenib in combination with indicated kinase inhibitors (E) or BET inhibitors (F). Drug doses, time points, and data normalization are similar to Figs 7C and 8C.

Data information: Data in (A, C, D) are presented as mean ± SD.

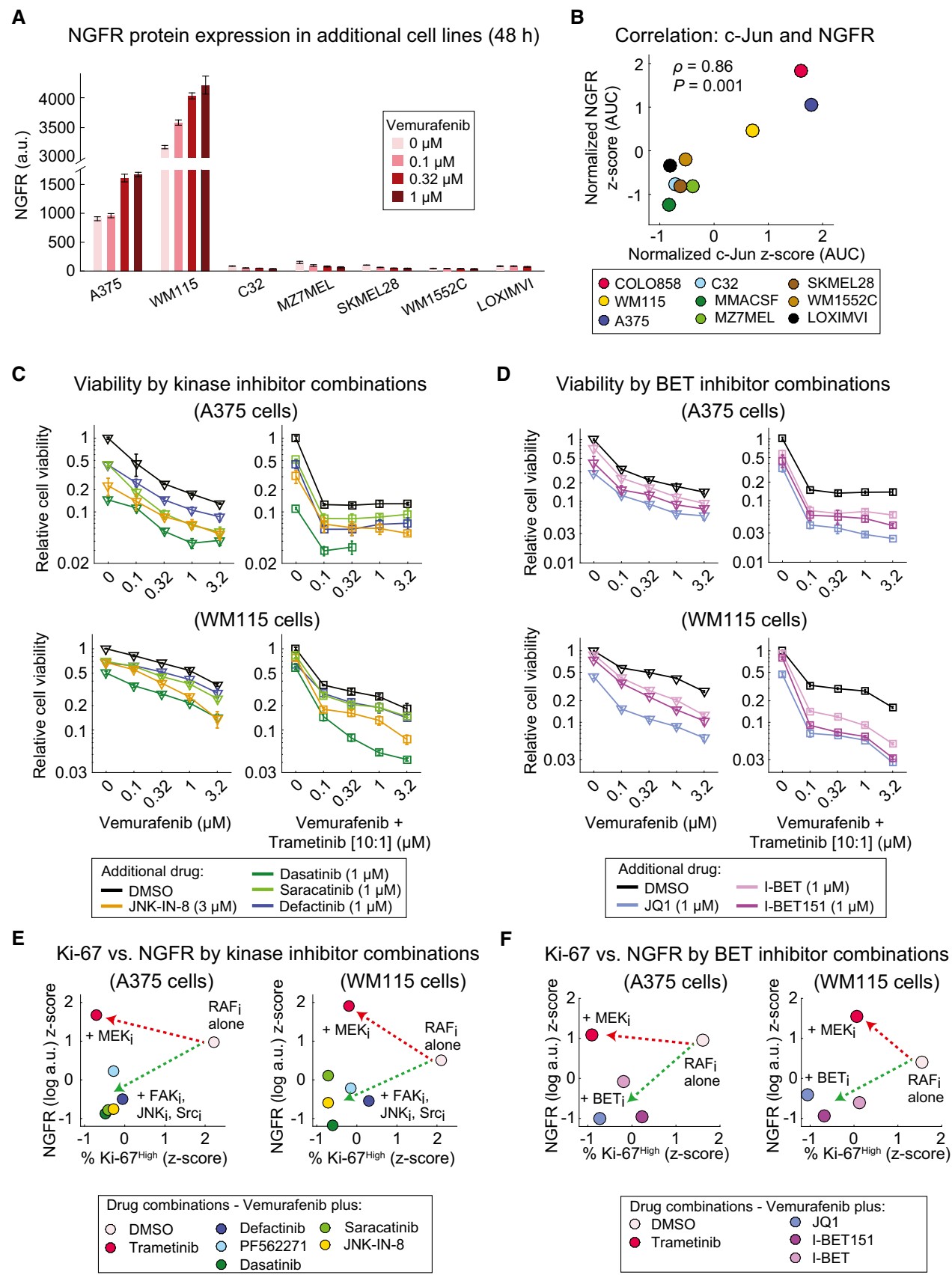

**Figure 9.**

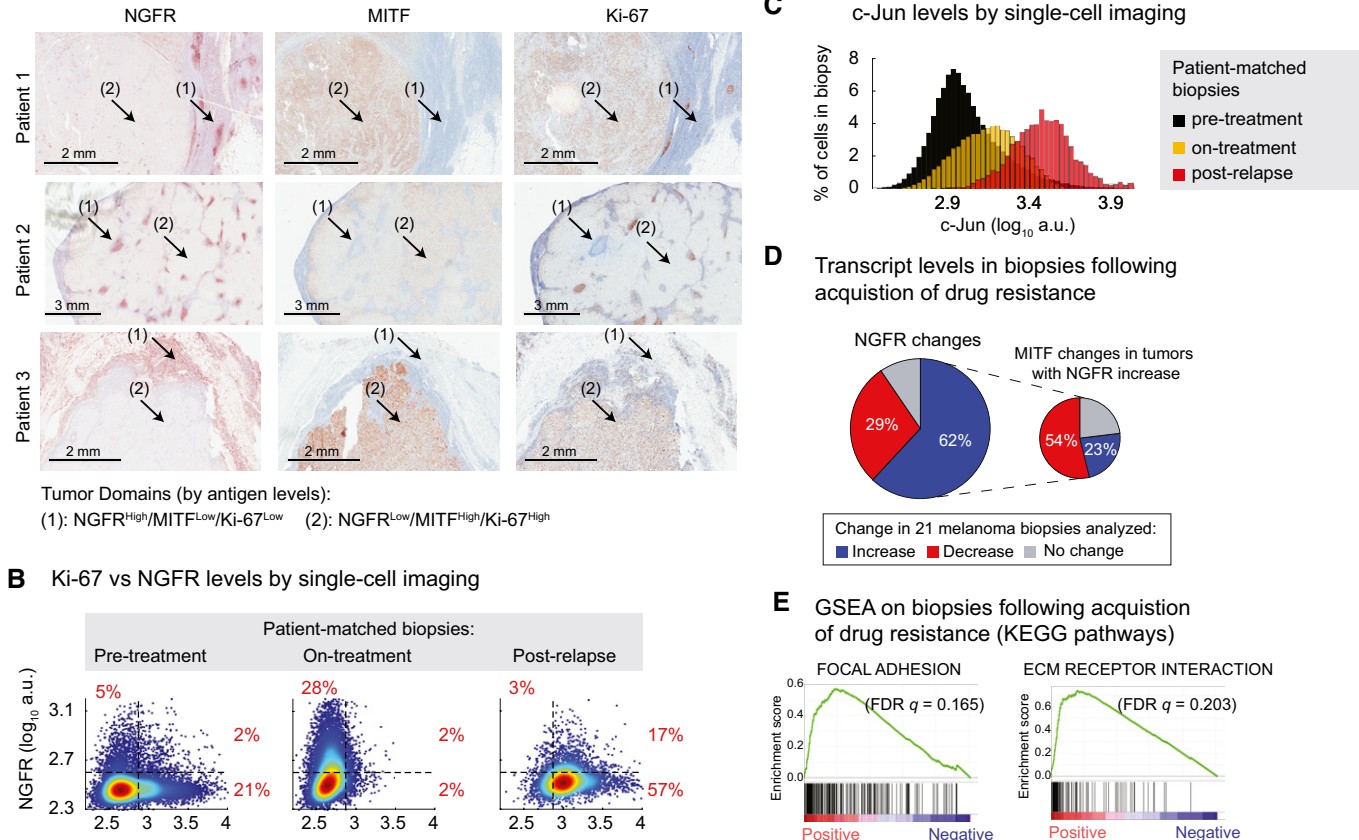

**Figure 10. The NGFR<sup>High</sup> state is associated with resistance to MAPK inhibitors in a subset of melanoma patients.**

A   Immunohistochemical analysis of vemurafenib-naïve tumors from three melanoma patients stained for NGFR, MITF, and Ki-67 (see Materials and Methods for patient clinical information).

B   Covariate single-cell analysis of Ki-67 versus NGFR measured by immunofluorescence in pre-treatment, on-treatment (with dabrafenib and trametinib combination for 2 weeks), and post-relapse tumor biopsies of a *BRAF*-mutant melanoma patient (see Materials and Methods for patient clinical information).

C   Cell population histograms representing c-Jun variations measured by immunofluorescence in the same patient-matched biopsies as shown in (B).

D   NGFR gene expression changes in 21 matched pairs of pre-treatment and post-resistance tumor biopsies analyzed by RNA sequencing. MITF changes are shown for tumors with a post-resistance NGFR increase (increase = $\log_2$ (fold-change) > 0.5, decrease = $\log_2$ (fold-change) < −0.5, no change = |$\log_2$ (fold-change)| ≤ 0.5). Gene expression data from patients treated with RAF inhibitor, MEK inhibitor, or their combination were analyzed by combining two published datasets (Sun *et al*, 2014; Hugo *et al*, 2015).

E   Ranked GSEA plots of top KEGG pathways significantly correlated with NGFR expression in 18 matched pairs of pre-treatment and post-resistance tumor biopsies (Hugo *et al*, 2015).

consistent with the anticipated effects of dabrafenib/trametinib therapy, but the fraction of NGFR<sup>High</sup> cells increased from ~7% to ~30% (Fig 10B). The distribution of signal intensities across single cells suggested that these changes primarily involved a switch from a Ki-67<sup>High</sup>/NGFR<sup>Low</sup> state to a Ki-67<sup>Low</sup>/NGFR<sup>High</sup> state following initiation of therapy. This change was associated with an increase in c-Jun expression (Fig 10C; compare yellow and black distributions). Following relapse, the fraction of Ki-67<sup>High</sup> cells increased dramatically (to ~74%) reflecting re-acquisition of proliferative potential and ~20% of these cells were NGFR<sup>High</sup>. Relapse was also accompanied by a dramatic increase in c-Jun levels (Fig 10C; red distribution). Although the number of samples in this study is low, the data are consistent with a heterogeneous distribution of NGFR<sup>High</sup>/Ki-67<sup>Low</sup> and NGFR<sup>Low</sup>/Ki-67<sup>High</sup> domains in melanoma tumors, and

a drug-mediated induction of NGFR<sup>High</sup>/Ki-67<sup>Low</sup> state that is concomitant with c-Jun up-regulation, a situation reminiscent of our observations in cultured cells.

To investigate changes in NGFR levels across a larger cohort of *BRAF*-mutant melanoma patients, we analyzed two published RNA-seq datasets involving matched samples from 21 tumors pre-treatment and following emergence of resistance to different combinations of RAF/MEK inhibitors (Sun *et al*, 2014; Hugo *et al*, 2015). In 62% of biopsies from patients with acquired drug resistance, NGFR gene expression increased as compared to pre-treatment levels (Fig 10D). MITF expression fell in only 50% of these biopsies, suggesting that therapy-induced NGFR up-regulation and MITF down-regulation do not necessarily occur concomitantly. GSEA of these tumors identified ECM–receptor interactions and

        

focal adhesion among the top enriched KEGG pathways correlated with NGFR gene expression levels, consistent with data obtained in cell lines (Fig 10E, and Datasets EV3 and EV4). We conclude that melanomas exhibit variability in differentiation status pre- and post-treatment but that acquisition of an NGFR$^{High}$ state is associated with resistance to RAF/MEK-targeted therapy in about half of melanomas examined.

### JQ1 suppresses induction of an NGFR$^{High}$ state in *BRAF$^{V600E}$* melanoma xenografts

To test whether the NGFR$^{High}$ phenotype can be blocked *in vivo* by drugs identified as effective in cell lines, we analyzed A375 cells grown as xenografts in nude mice. A375 cells are among the most widely used xenograft models for *BRAF*-mutant melanoma. Mice were exposed for 5 days to RAF inhibitor dabrafenib (at a 25 mg/kg dose) alone or in combination with JQ1 (at a 50 mg/kg dose). Four xenograft tumors per condition were excised, fixed, sectioned, and then co-stained for NGFR and Ki-67. Analysis of staining intensity at a single-cell level revealed heterogeneity from

one region of tumor to the next and a reciprocal relationship between regions of the tumor that were NGFR$^{High}$ and Ki-67$^{High}$ (Fig 11A), a pattern similar to what was observed in human tumors. Treatment of animals with dabrafenib plus JQ1 significantly reduced the fraction of NGFR$^{High}$ cells as compared to dabrafenib alone (or vehicle-treated controls) and the combination also reduced the fraction of Ki-67$^{High}$ cells relative to dabrafenib or vehicle (Fig 11B). These data mimic two key aspects of what we observed in cultured A375 cells (which have high NGFR levels in the basal state): First, JQ1 can reduce NGFR levels, and second, JQ1 and dabrafenib can combine to reduce the fraction of proliferating Ki-67$^{High}$ cells. Moreover, as these experiments were being conducted, a study was published on tumor burden in mice engrafted with A375 tumors. It showed that JQ1 and vemurafenib have synergistic effects of tumor shrinkage (Paoluzzi *et al*, 2016). Together, these findings establish that the effects of co-drugging with JQ1 and MAPK inhibitors observed in cell culture can also be obtained in xenograft models. This sets the stage for large-scale pre-clinical evaluation of drugs such as BET bromodomain inhibitors as a means of blocking drug adaptation and increasing

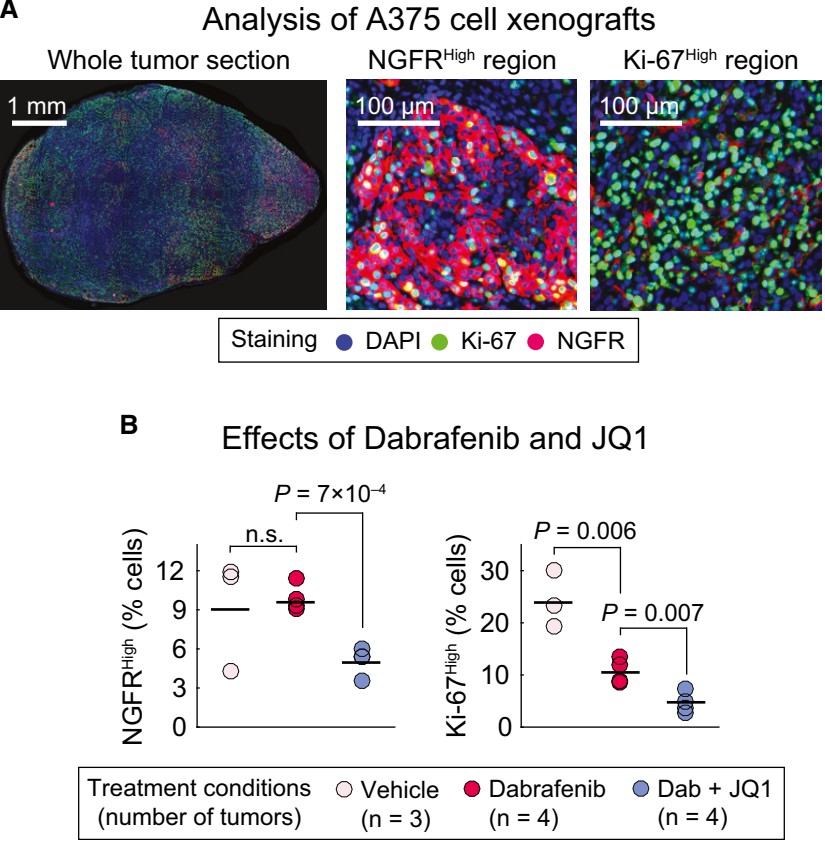

**Figure 11.  The NGFR$^{High}$ phenotype can be suppressed by JQ1 in *BRAF$^{V600E}$* melanoma xenografts.**

A  Immunofluorescence analysis of A375 melanoma xenograft tumors co-stained for Ki-67 and NGFR proteins. Selected images from a whole tumor section as well as NGFR$^{High}$/Ki-67$^{Low}$ and NGFR$^{Low}$/Ki-67$^{High}$ regions of a vehicle-treated tumor are shown to highlight the spatial and cell-to-cell heterogeneity in Ki-67 and NGFR protein expression.

B  Percentage of Ki-67$^{High}$ and NGFR$^{High}$ cells in tumors treated for 5 days with dabrafenib (25 mg/kg) only, dabrafenib (25 mg/kg) in combination with JQ1 (50 mg/kg), or vehicle. Number of tumors (mice) analyzed per condition is shown. Solid horizontal lines represent the mean of measurements. Up to 50,000 individual cells per tumor were analyzed for NGFR and Ki-67 intensities. Statistical significance was determined using two-tailed two-sample *t*-test.

cell killing by MAPK inhibitors in a subset of *BRAF*-mutant melanomas.

## Discussion

In this paper, we use time-lapse, live-cell imaging, and single-cell analysis to show that *BRAF*-mutant melanoma cells exhibit time-variable and heterogeneous phenotypes when exposed to MAPK pathway inhibitors such as vemurafenib, dabrafenib, and trametinib near the $IC_{50}$ for cell killing. Cells initially undergo growth arrest, consistent with the known requirement for MAPK activity in proliferation. Apoptosis peaks between 48 and 72 h and typically kills 40–60% of cells, while other cells enter a G0/G1 arrest. In a subset of lines, a subpopulation of cells overcomes drug-mediated cell cycle arrest and begins to divide threefold to fourfold more slowly than drug-naïve cells. Such adapted cells exhibit elevated neural crest markers including NGFR and neurogenesis genes, suggestive of drug-induced de-differentiation and consistent with previous studies associating increased NGFR levels or loss of melanocyte differentiation markers (e.g., MITF) with resistance to MAPK pathway inhibitors (Konieczkowski *et al*, 2014; Muller *et al*, 2014; Ravindran Menon *et al*, 2015). In culture, the generation of slowly cycling NGFR[High] cells reduces drug maximal effect, as evidenced by short-term (3-day) viability assays and week-long time-lapse imaging. Slowly cycling, drug-adapted cells are likely to contribute to residual disease and eventual emergence of genetically distinct drug-resistant clones (Frick *et al*, 2015; Hata *et al*, 2016).

### Reversible drug resistance

The slowly cycling NFGR[High] state induced by vemurafenib is only transiently stable: After 9 days of outgrowth in drug-free medium, such cells reset to their initial state as measured by restoration of vemurafenib sensitivity, increased proliferation rate, and reduced expression of NGFR. Such metastable, bidirectional changes in cell state are inconsistent with selection of pre-existing genetic variants but are more durable than transiently heritable differences generated by stochastic fluctuation in protein levels (Cohen *et al*, 2008; Gascoigne & Taylor, 2008; Flusberg *et al*, 2013). Instead, the phenomenon is reminiscent of drug-tolerant persisters (DTPs), which constitute < 1% of drug-naïve cell populations, become enriched following exposure to high concentrations of anti-cancer drugs (> 100-fold above $IC_{50}$ values) for > 9 days, and have hyperactive IGF-1R signaling (Sharma *et al*, 2010). Like NGFR[High] melanoma cells, DTPs are sensitive to some kinase inhibitors and to inhibitors of epigenome-modifying enzymes, HDACs in the case of DTPs, and BET inhibitors in the case of vemurafenib-adapted melanoma cells. However, time-lapse imaging shows that melanoma cells responding to vemurafenib induce a slowly dividing drug-adapted state more rapidly and in a larger fraction of cells (> 20% of cells by 3 days near the vemurafenib $IC_{50}$) than has been observed for DTPs. We speculate that differences between these phenomena originate primarily from differences in the strength and timing of imposed selective pressures; whereas acutely high doses of drug lead to selection of a small percentage of intrinsically, highly insensitive cells (i.e., DTPs) (Sharma *et al*, 2010; Roesch *et al*, 2013), lower and more realistic drug doses provide a larger fraction of cells with

sufficient time to induce an adaptive mechanism and become drug insensitive; once induced, this resistance appears to protect cells from higher doses of drug (Ravindran Menon *et al*, 2015).

Our data add to a growing body of research suggesting that tumor cells can reversibly undergo dynamic changes that create subpopulations of cells with different proliferative potentials and sensitivity to apoptosis. Stochastic fluctuation in protein levels (Spencer *et al*, 2009), DTPs, and NGFR[High] melanoma cells represent three distinguishable but related mechanisms of achieving a state of reversible drug resistance. Such cells are thought to be the basis of residual disease and to provide a pool for further genetic or epigenetic changes that eventually induce the growth of drug-resistant clones (Hata *et al*, 2016).

### A speculative pathway for reversible drug resistance in melanoma

Based on data from drug-adapted cells in culture, the efficacy of co-drugging these cells with kinase and BET bromodomain inhibitors and analysis of gene expression profiles in human melanoma biopsies, we propose a speculative model for the adaptive resistance to RAF/MEK inhibition characterized in this paper. Exposure of *BRAF*-mutant melanoma cells to MAPK inhibitors initially induces up-regulation of JNK/c-Jun signaling, a known regulator of EMT-related genes and of cell adhesion and ECM molecules (Liu *et al*, 2015; Ramsdale *et al*, 2015). Up-regulation of cell adhesion proteins is accompanied by activation of FAK and downstream Src kinases, causing cells to acquire a distinct epigenetic state and become more neural crest-like. Such cells divide slowly and have a reduced requirement for ERK signaling.

We and others have recently reported that JNK and c-Jun are activated in a subset of melanomas exposed to MAPK inhibitors (Delmas *et al*, 2015; Fallahi-Sichani *et al*, 2015; Ramsdale *et al*, 2015; Riesenberg *et al*, 2015; Titz *et al*, 2016). Our current study links this phenomenon to transiently heritable (reversible) de-differentiation by a subset of cells in the population and to high expression of NGFR, which has previously been observed in human tumors. We also identify compounds other than JNK inhibitors able to block drug adaptation and increase cell killing. Studies on an as-yet limited number of biopsies show that NGFR expression is induced by MAPK inhibitors in human tumors, concomitant with a reduction in cell proliferation; this effect is highly heterogeneous across a single human tumor and also across a xenograft, the former representing a genetically heterogeneous sample and the latter a more homogenous one. In a human tumor analyzed prior to therapy, on therapy and following progression, we find that c-Jun levels increase upon initial MAPK inhibition and rise further when tumors become drug-resistant. Thus, it seems plausible that mechanisms identified in cultured cells are also operative in real tumors. It is important to note, however, that JNK/c-Jun-dependent adaptation marked by an NGFR[High] state, as described here, appears to occur in only a subset (about one-third) of cell lines studied. Other mechanisms are presumably operative in other cell lines.

### Inhibitors of adaptive drug resistance

By targeted screening, we identify two classes of compounds with the potential to block vemurafenib-induced de-differentiation (as marked by elevated NGFR expression): (i) small-molecule kinase

inhibitors against components of the postulated c-Jun/FAK/Src cascade and (ii) epigenetic modifiers, including BET bromodomain inhibitors presumed to block the de-differentiation program. Combining vemurafenib with JNK, FAK, or Src kinase inhibitors, or with BET inhibitors suppresses acquisition of the NGFR[High] phenotype, prevents the emergence of slowly cycling drug-resistant cells, and enhances cell killing. In the case of BET inhibitor JQ1, we also show that co-drugging suppresses the NGFR[High] state in *BRAF*-mutant melanoma xenografts treated with dabrafenib. The primary effect of co-drugging on cultured cells is on maximum effect ($E_{max}$) and involves reducing viable cell number (in a 3-day assay) from 1% to 10% of the initial population to 0.01–0.1%. In our opinion, such an effect would be missed by most protocols used to screen for drug combinations.

The molecular effects of RAF/MEK and JNK/FAK/Src/BET inhibitors appear to be orthogonal, with the former suppressing MAPK signaling and the latter suppressing the consequent emergence of de-differentiated, adapted cells. Moreover, experiments with vemurafenib and trametinib show that the greater the extent of MAPK inhibition, the greater the extent of adaptation. Thus, inhibitors of adaptation such as dasatinib (Sprycel®) might be expected to combine with MAPK inhibition in therapeutically beneficial ways. Inhibiting Src family kinases has previously been reported to overcome resistance to RAF inhibitors (Girotti et al, 2013), although this was attributed to a role for Src downstream of RTKs rather than FAK. Vemurafenib has also been shown to activate melanoma-associated stromal fibroblasts, increasing ECM production and elevating integrin/FAK/Src signaling to promote vemurafenib resistance in nearby melanoma cells (Hirata et al, 2015). All three of these mechanisms could be involved at the same time, perhaps to different extents in different settings.

Current understanding of biomarkers for vemurafenib-induced de-differentiation in melanomas remains incomplete. For example, the switch of melanoma cells in culture to a drug-resistant NGFR[High] phenotype is not associated with a reduction in MITF levels. Moreover, only about half of NGFR[High] post-resistance biopsies exhibited a reduction in MITF levels, suggesting that therapy-induced NGFR up-regulation and MITF down-regulation are not necessarily concomitant. Low MITF expression in melanomas has previously been linked to increased expression of RTKs such as AXL, EGFR, and PDGFRβ, which activate immediate–early signaling, causing resistance to RAF/MEK inhibitors (Muller et al, 2014). However, the NGFR[High] phenotype we observe is not associated with RTK up-regulation as judged by mRNA expression. These findings raise the question whether we and others are probing different aspects of a unified adaptive mechanism common to all melanomas or whether adaptation is fundamentally different in genetically distinct tumor cells. Answering this question at a single-cell level may help identify novel therapies and biomarkers that have been missed by experiments that focus on bulk tumor cell killing.

# Materials and Methods

## Cell culture

Melanoma cell lines used in this study were obtained from the Massachusetts General Hospital Cancer Center with the following primary sources: COLO858 (ECACC), A375, C32, WM115, SKMEL28, and WM1552C (ATCC), LOXIMV1 (DCTD Tumor Repository, National Cancer Institute), MMACSF (RIKEN BioResource Center), and MZ7MEL (Johannes Gutenberg University Mainz). C32, MMACSF, SKMEL28, and WM115 cell lines were grown in DMEM/F12 (Invitrogen) supplemented with 5% fetal bovine serum (FBS) and 1% sodium pyruvate (Invitrogen). COLO858, LOXIMVI, MZ7MEL, and WM1552C cell lines were grown in RMPI 1640 (Corning cellgro) supplemented with 5% FBS and 1% sodium pyruvate (Invitrogen). A375 cells were grown in DMEM with 4.5 g/l glucose, L-glutamine, and sodium pyruvate (Corning cellgro), supplemented with 5% FBS. We added penicillin (50 U/ml) and streptomycin (50 μg/ml) to all growth media.

## Reagents and antibodies

Chemical inhibitors from the following sources were dissolved in dimethyl sulfoxide (DMSO) as 10 mM stock solution and used in treatments: vemurafenib (MedChem Express), trametinib (GSK1120212), defactinib, PF562271, pictilisib (GDC0941), palbociclib (PD0332991), and AZD8055 (all from Selleck Chemicals), JNK-IN-8 (EMD Millipore), dasatinib, saracatinib, (+)-JQ1, I-BET, I-BET151, and belinostat (all from Haoyuan Chemexpress). The following primary antibodies with specified animal sources and catalogue numbers were used in specified dilution ratios in immunofluorescence analysis of cells and tissues: p-S6[S240/244] rabbit monoclonal antibody (mAb) (clone D68F8, Cat# 5364), 1:800; p-ERK[T202/Y204] rabbit mAb (clone D13.14.4E, Cat# 4370), 1:800; c-Jun rabbit mAb (clone 60A8, Cat# 9165), 1:800; p-c-Jun[S73] rabbit mAb (clone D47G9, Cat# 3270), 1:800; Ki-67 mouse mAb (clone 8D5, Cat# 9449), 1:400; c-Jun mouse mAb (clone L70B11, Cat# 2315), 1:200; p75NTR (NGFR) rabbit mAb (clone D4B3, Cat# 8238), 1:1,600 (for staining cultured cells) or 1:200 (for staining tissue sections); all from Cell Signaling Technology, and p-Rb[S807/811] goat polyclonal antibody (Cat# sc-16670), 1:400, from Santa Cruz Biotechnology. The following antibodies were diluted 1:1,000 and used in Western blots: p-FAK[Y397] rabbit mAb (clone D20B1, Cat# 8556), FAK rabbit mAb (clone D2R2E, Cat# 13009), integrin β1 rabbit mAb (clone D2E5, Cat# 9699), p75NTR (NGFR) rabbit mAb (clone D4B3, Cat# 8238), β-actin rabbit mAb (clone D6A8, Cat# 8457), all from Cell Signaling Technology; HSP90α/β rabbit polyclonal antibody (Cat# sc-7947) from Santa Cruz Biotechnology; and thrombospondin rabbit polyclonal antibody (Cat# ab85762) from Abcam.

## Human tumor specimens

Under IRB-approved protocols, freshly procured and discarded melanoma tumor specimens were formalin-fixed, paraffin-embedded, sectioned, and stained with H&E for histopathological evaluation or immunofluorescence staining. Clinical history of the drug-naive patients (age, sex, *BRAF* mutation, sequencing method, treatment history) is as follows: patient 1 (74, male, wild-type, whole-exome sequencing, no prior treatment), patient 2 (58, male, *BRAF^{V600E}*, targeted sequencing, no prior treatment), patient 3 (86, female, *BRAF^{V600E}*, whole-exome sequencing, no prior treatment), and patient 4 (65, male, *BRAF^{V600E}*, targeted sequencing, interferon). In the case of patient-matched biopsies from pre-treatment, on-treatment (for 2 weeks), and post-relapse tumors, biopsies were

collected from a male patient with metastatic *BRAF*-mutant melanoma, treated with dabrafenib and trametinib combination.

## Immunohistochemistry of human tumor specimens

Tumor sections were deparaffinized, and heat-induced epitope retrieval (HIER) was performed on the unit using EDTA for 20 min at 90°C. Sections were incubated for 30 min with primary antibodies including Ki-67 rabbit monoclonal antibody (clone SP6, Cat# VP-RM04) from Vector Laboratories, p75NTR/NGFR rabbit polyclonal antibody (Cat# 119-11668) from RayBiotech, and MITF mouse monoclonal antibody (clone D5, Cat# MA5-14154) from Thermo Scientific, and were then completed with the Leica Refine detection kit (secondary antibody, the DAB chromogen, and the hematoxylin counterstain).

## Live-cell reporter constructs

To generate cells expressing fluorescently tagged geminin and H2B, we used the pPB-CAG.EBNXN/pCMV-hyPBase transposase vector system (Allan Bradley, Sanger Institute). First, a pPB-CAG vector containing a multiple cloning site (pPB-CAG-MCS) was generated by annealed oligo cloning of the following primers into the EcoRI and SalI restriction sites of pPB-CAG-EKAREV (Albeck *et al*, 2013) containing a puromycin selection cassette: 5′-aattcggatcccatatgca cgtgctcgagg-3′ and 5′-tcgacctcgagcacgtgcatatgggatccg-3′. Next, intermediate pPB-CAG constructs were generated for ERK-KTR-mTurquoise2, H2B-Venus, and mCherry-geminin performing Gibson Assembly (New England Biolabs) at the EcoRI and SalI restriction sites and using the following templates and primers: ERK-KTR (Regot *et al*, 2014) with 5′-tctcatcattttggcaaagaattcggcatgaagggccga aagcct-3′ and 5′-ctcaccatactagtggatgggaattgaaag-3′ and mTurquoise2 (Goedhart *et al*, 2012) with 5′-ccactagtatggtgagcaagggcgag-3′ and 5′-ca cacattccacagggtcgacttacttgtacagctcgtccatg-3′; H2B (Nam & Benezra, 2009) with 5′-tctcatcattttggcaaagaattcggcatgcctgaaccctctaagtctgc-3′ and 5′-ctcaccatggtggcgaccggtggatc-3′ and Venus with 5′-tcgccaccatgg tgagcaagggcgag-3′ and 5′-cacacattccacagggtcgacttatttgtacaattcgtccatc ccc-3′; mCherry with 5′-tctcatcattttggcaaagaattcggcatggtgagcaag ggcgag-3′ and 5′-ggatatcccttgtacagctcgtccatgc-3′ and geminin (Sakaue-Sawano *et al*, 2008) with 5′-ctgtacaagggdatatccatcacactggc-3′ and 5′-cacacattccacagggtcgacttacagcgcctttctccg-3′. These intermediate constructs were used as templates for a final round of Gibson cloning to generate pPB-CAG-ERK-KTR-mTurquoise2-P2A-H2B-Venus-P2A-mCherry-geminin in which the DNA coding for three live-cell reporters is separated by self-cleaving P2A sites: ERK-KTR-mTurquoise2 with 5′ ctgtctcatcattttggcaaag-3′ and 5′-cacgtcgcca gcctgcttaagcaggctgaagttagtagctccgcttcccttgtacagctcgtccatg-3′, H2B-Venus with 5′-ttcagcctgcttaagcaggctggcgacgtggaggagaaccccgggccta tgcctgaaccctctaag-3′ and 5′-gacatcccccgcttgtttcaataacgaaaaattcgtcg cgcccgagcctttgtacaattcgtccatcc-3′, mCherry-geminin with 5′-ttttcgtt attgaaacaagcggggatgtcgaagaaaatccgggcccgatggtgagcaagggcg-3′ and 5′-ctgacacacattccacagggtcgacttacagcgcctttctccgtttttc-3′. Plasmid DNA was provided by Marcus Covert (ERK-KTR), Joachim Goedhart (mTurquoise2), Robert Benezra (H2B-mCherry, Addgene plasmid # 20972), Atsushi Miyawaki (geminin), and Allan Bradley (pPB-CAG.EBNXN and pCMV-hyPBase). Positive clones were confirmed by sequencing. To create stable cell lines, cells were co-transfected with the pPB-CAG triple reporter plasmid and pCMV-hyPBase using

FuGene HD (Promega) and transiently selected with puromycin. To enrich for cells stably expressing the live-cell reporter at comparable levels, cells were subjected twice to FACS. Reporter and parental cell lines were confirmed to grow at comparable rates for different vemurafenib concentrations over 72 h of treatment.

## Live single-cell imaging and analysis

Cell lines stably expressing the live-cell reporter were seeded into Costar 96-well black clear-bottom tissue culture plates (Corning 3603) in 200 μl full growth medium without phenol red at a density of 4,500 cells per well for COLO858 or 4,000 cells per well for MMACSF; cells were counted using a Cellometer Auto T4 Cell Viability Counter (Nexcelom Bioscience). To facilitate cell tracking, COLO858 reporter cells treated with DMSO were mixed with an equal amount of parental cells, maintaining an overall cell density of 4,500 cells/well. The next day, cells were treated with DMSO or 1 μM vemurafenib, or with 1 μM vemurafenib in combination with DMSO, 3 μM defactinib, 1 μM dasatinib, or 0.32 or 1 μM (+)-JQ1 using an Hewlett-Packard (HP) D300 Digital Dispenser. Within 45–80 min after drug treatment, image acquisition was started using a Nikon Ti motorized inverted microscope with a 10× Plan Fluor 0.30 NA Ph1 objective lens and the Perfect Focus System for continuous maintenance of focus. Plates were placed into an OkoLab cage microscope incubator set to 37°C, 5% $CO_2$, and 90% humidity to enable stable environmental conditions throughout the experiment. Images were acquired every 6 min for the indicated times with a Hamamatsu ORCA ER cooled CCD camera controlled with Meta-Morph 7 software, using a 2 × 2 binning. For illumination, the Lumencor Spectra-X light engine in combination with a CFP/YFP/mCherry beam splitter (Chroma ID No. 032357) was used. H2B-Venus fluorescence was collected with a 508/24 excitation and a 540/21 emission filter at 200 ms exposure, and mCherry-geminin fluorescence was collected with a 575/22 excitation and a 632/60 emission filter at 400 ms exposure.

Individual cells from up to 10 wells per condition were analyzed. Cell positions and cell death/division events were manually tracked using a custom MATLAB-based script provided by Jose Reyes, Kyle W. Karhohs, and Galit Lahav (Harvard Medical School). Using H2B, a total of 150–217 cells were manually tracked and cell division and death events were recorded. To derive statistical mean and variance, cells from multiple wells were pooled together to generate three or four groups of wells containing ~50–70 cells. Data from 3 to 4 groups of cells were then used to report the mean ± SD. For extracting the geminin signal, the mean intensity of the centroid dilated by 12 pixels was calculated after using a rolling ball background subtraction. To determine the onset of S/G2, a moving average with a window of 40 frames was calculated for the geminin signal and in general an average value above a threshold of 2.0 (COLO858) or 1.5 (MMACSF) was determined as the start of the S/G2 cell cycle stage.

To measure cell division times following longer periods of vemurafenib treatment (i.e., ~8 days), COLO858 cells were initially exposed to 1 μM vemurafenib for 2 days, medium was then changed (to remove apoptotic cells), and cells were treated with 1 μM vemurafenib for an additional 2 days before they were imaged for ~4 days. Minimum doubling times were estimated for 100 individual cells tracked between days 4 and 8 post-treatment

by identifying the longest time interval before or after which a cell divides.

### Long-term time-lapse live-cell analysis using IncuCyte

COLO858 cells expressing H2B-mVenus were imaged every 45 min for ~1 week after treatment in three replicates with indicated drugs at indicated concentrations with a 4× objective using IncuCyte ZOOM live-cell imager (Essen Bioscience). Dead cells were identified by staining with IncuCyte CytoTox Red Reagent (Essen Bioscience, Cat# 4632). Time-lapse live-cell analysis (following exclusion of dead cells) was performed using ImageJ software.

### Apoptosis, cell viability, and growth rate inhibition assays

For 72–96 h viability, apoptosis, or growth rate inhibition assays, cells were seeded in 3–6 replicates at 2,500–5,000 cells per well in 96-well plates (Corning 3603) in 180 μl of full growth media; cells were counted using a Cellometer Auto T4 Cell Viability Counter (Nexcelom Bioscience). Cells were treated the next day using a Hewlett-Packard (HP) D300 Digital Dispenser with compounds at reported doses. To score viability and apoptosis, we used a dye-based imaging assay: The cell-permeable DNA dye Hoechst 33342 was used to mark nuclei, and DEVD-NucView488 caspase-3 substrate was used to mark apoptosis, as previously described in detail (Fallahi-Sichani et al, 2015). Imaging was performed using a 10× objective using a PerkinElmer Operetta High Content Imaging System. Eleven sites were imaged in each well. Image segmentation and analysis were performed using Acapella software (PerkinElmer). The nuclear segmentation with Hoechst 33342 was used to identify individual nuclei and to count cells. To score apoptotic cells, bright spots were detected by dividing NucView488 channel nuclear intensity by the nucleus area and spots brighter than a separating threshold were scored as apoptotic. Relative viability was calculated by subtracting the number of apoptotic cells from the total number of cells to achieve viable cell count at each condition that was normalized to a DMSO-treated control. To compare drug effect on different cell populations that grow at different rates (e.g., FACS-sorted NGFR$^{High}$ and NGFR$^{Low}$ cells), growth rate (GR) inhibition was calculated by normalizing viable cell count data to the growth rate of untreated cells as described previously (Hafner et al, 2016). Data were analyzed using MATLAB 2014b software.

### RNA extraction, library construction, and RNA-seq analysis

COLO858 and MMACSF cells were seeded in 10-cm plates in two replicates, treated the next day with either DMSO or 0.2 μM vemurafenib for 24 and 48 h. At the time of harvest for RNA, cells were washed once with PBS and then lysed in the dish with RLT buffer (Qiagen). Samples were immediately processed with Qiashredder and RNeasy kits (Qiagen) and frozen until further use. 10 μg of RNA was DNAse-treated and cleaned up with RNeasy MinElute kit (Qiagen). RNA quality was assessed by Bioanalyzer (Agilent), and all samples had RINs of 9.0 or higher. RNA-seq libraries were constructed using Illumina's TruSeq-stranded mRNA library prep kit and protocol with minor modifications. Briefly, 1 μg of RNA was mixed with 2 μl of a 1:100 dilution of ERCC spike in control Mix2

(Life Technologies) before mRNA purification. Elution and fragmentation was done for 6 min at 94°C. cDNA was synthesized and cleaned up with Ampure beads (Agencourt). Fragments were end-modified and adaptors ligated before another Ampure bead cleanup. Final library amplification was done at 13 cycles and again cleaned up with Ampure beads. The resulting libraries were roughly 380 bp in length as assessed by Bioanalyzer. RAN-seq was performed at the Harvard University Sequencing Facility (FAS Division of Science) on Illumina HiSeq 2000 machines using the standard single-read (1 × 50 bp) protocol. The reads were mapped to the human genome (build hg19) using Tophat (Kim et al, 2013) with default settings, and differential expression analysis was performed using Cuffdiff (Trapnell et al, 2010) running on the web-based Galaxy platform (https://usegalaxy.org/). To identify differentially regulated genes between COLO858 and MMACSF cells, we first selected genes whose expression at 24 or 48 h following treatment changed relative to a DMSO-treated control ($q < 0.01$) in at least one of the cell lines; we then identified those genes with FPKM $\geq 1$, that changed in abundance by more than twofold between the two cell lines ($|\log_2 (\text{ratio})| \geq 1$), where "ratio" represents treatment versus DMSO fold-change in COLO858 divided by the treatment versus DMSO fold-change in MMACSF cells. Differentially regulated genes were processed using Metacore (Genego, Inc) software available online (http://portal.genego.com/). Ranked biological processes and pathways were generated using "analyze single experiment" feature with default settings.

### Hierarchical clustering

Unsupervised hierarchical clustering of expression levels of differentially regulated genes between COLO858 and MMACSF cell lines was carried out using MATLAB 2014b, using the Chebyshev distance as the metric. (FPKM + 1) values of vemurafenib-treated conditions were normalized to those of DMSO-treated controls and $\log_2$-transformed prior to clustering.

### Bioinformatics analysis

Gene expression data analysis and heat-map visualization were performed using MATLAB 2014b software. Differentially expressed genes with transcription factor activity and genes associated with cell surface receptors and secreted peptides/proteins were identified using Advanced Search 2.0 feature of the online Genego database (https://portal.genego.com/). Enriched transcriptional regulators for the list of differentially expressed genes were predicted using the Database for Annotation, Visualization and Integrated Discovery (DAVID) v6.7 (http://david.ncifcrf.gov/) (Huang da et al, 2009a,b), and they were compared to the gene expression levels of transcription factors 24–48 h after vemurafenib treatment.

### Gene set enrichment analysis

Gene set enrichment analysis (GSEA) (Mootha et al, 2003; Subramanian et al, 2005) was performed using GSEA v2.2.0 software with 1,000 phenotype permutations. Gene Ontology (GO) biological processes (c5.bp.v5.0.symbols.gmt), and Kyoto Encyclopedia of Genes and Genomes (KEGG) pathway (c2.cp.kegg.v5.0.symbols.gmt) gene sets were obtained from http://www.broadinstitute.

org/gsea/downloads.jsp and used in GSEA. To identify biological processes and pathways most correlated with NGFR expression, we performed GSEA on RNA-seq data of tumors from 128 $BRAF^{V600E}$ melanoma patients included in TCGA (Cancer Genome Atlas Network, 2015), and microarray data of 25 $BRAF^{V600E}$ melanoma cell lines in the CCLE (Barretina *et al*, 2012), and by selecting NGFR expression levels as the "phenotype" and "Pearson" as the metric for ranking genes. A detailed description of GSEA methodology and interpretation is provided at http://www.broadinstitute.org/gsea/doc/GSEAUserGuideFrame.html. Briefly, enrichment score indicates "the degree to which a gene set is overrepresented at the top or bottom of a ranked list of genes". The false discovery rate (FDR *q*-value) is "the estimated probability that a gene set with a given enrichment score represents a false positive finding". "In general, given the lack of coherence in most expression datasets and the relatively small number of gene sets being analyzed, an FDR cutoff of 25% is appropriate".

## Fluorescence-activated cell sorting

COLO858 cells were seeded in 15-cm plates, treated the next day with 0.32 μM vemurafenib for 48 h. The cell monolayer was incubated at 37°C with trypsin 0.05% (Gibco) for 1 min, lifted from the plate with a cell scraper and re-suspended in PBS with 2% FBS. The cell suspension was washed in PBS with 2% FBS twice. Cells were counted and assessed for viability by trypan blue exclusion test on a Vi-CELL Cell Viability Counter instrument (Beckman Coulter). The cell suspension was incubated with a PE-conjugated NGFR monoclonal antibody (clone ME20.4-1.H4, Miltenyi Biotec) and with calcein-AM viability marker (Life Technologies) per manufacturers' recommendations. After incubation, the cell suspension was washed twice in PBS with 2% FBS. Cells were sorted on a BD FACSAria II (BD Biosciences) instrument. Unstained and calcein-AM and NGFR-PE single-color controls were used to set appropriate gates. Cells high for calcein were gated, and ~1 million NGFR$^{High}$ or NGFR$^{Low}$ cells were sorted into 15-ml conical tubes (Falcon) that were prepared with 5 ml RMPI 1640 (Corning cellgro) supplemented with 5% FBS and 1% sodium pyruvate (Invitrogen) and plated using culture conditions described above. FACS-sorted NGFR$^{High}$ and NGFR$^{Low}$ cells were counted using a Countess II FL Automated Cell Counter (Life Technologies) and cultured for 9 days in fresh growth media in 96-well plates or treated with vemurafenib for immunofluorescence and growth rate inhibition assays.

## *In vivo* xenograft studies

All animal experiments were conducted in accordance with procedures approved by the Institutional Animal Care and Use Committee (IACUC) at Harvard Medical School. Six-week-old NU/J mice (Jackson Laboratory, Stock# 002019) were transiently anesthetized using 5% vaporized isoflurane and injected subcutaneously in the right flank with $2.5 \times 10^6$ A375 human melanoma cells suspended in 200 μl of growth factor-reduced Matrigel (Corning 356230) in PBS (1:1). Tumor xenografts were allowed to grow until the mean volume across the tumors reached 150 mm$^3$ as measured by digital calipers. Mice were then randomly assigned to treatment (once-daily) for 5 days with one of the following drug combinations: *Group 1*, 200 μl dabrafenib (25 mg/kg) via oral gavage (OG) plus

500 μl JQ1 (50 mg/kg) via intraperitoneal (IP) injection; *Group 2*, 200 μl of dabrafenib (25 mg/kg) via OG plus 500 μl of IP vehicle control; *Group 3*, OG and IP vehicle controls given at equivalent volumes. Dabrafenib was diluted in 0.5% hydroxypropylmethylcellulose and 0.2% Tween-80 in pH 8.0 distilled water, and JQ1 was diluted in 5% dextrose in water. After 5 days, mice were transcardially perfused with oxygenated and heparinized Tyrode's solution; this allowed for simultaneous euthanasia and exsanguination. Flank xenografts were then surgically removed and fixed in 4% paraformaldehyde (PFA) in PBS and stored at 4°C for 48 h. All fixed tumors from a given treatment group were uniformly paraffin-embedded into a single paraffin block holder (i.e., 1 block holder per treatment group) and sectioned at 5 μm thickness: This allowed each microscope slide bearing a single 5-μm paraffin section to contain a representative sample of all the tumors from a given treatment group.

## Immunofluorescence staining, quantitation, and analysis

Immunofluorescence assays for cultured cells were performed using cells seeded in 96-well plates (Corning 3603) and then treated the next day using Hewlett-Packard (HP) D300 Digital Dispenser with compounds at reported doses for indicated times in 2–3 replicates. Cells were fixed in 4% paraformaldehyde for 20 min at room temperature and washed with PBS with 0.1% Tween-20 (Sigma-Aldrich) (PBS-T), permeabilized in methanol for 10 min at room temperature, rewashed with PBS-T, and blocked in Odyssey blocking buffer (OBB) for 1 h at room temperature. Cells were incubated overnight at 4°C with primary antibodies in OBB. Cells were then stained with rabbit, mouse, and goat secondary antibodies from Molecular Probes (Invitrogen) labeled with Alexa Fluor 647 (Cat# A31573), Alexa Fluor 488 (Cat# A21202), and Alexa Fluor 568 (Cat# A11057). Cells were washed once in PBS-T, once in PBS, and were then incubated in 250 ng/ml Hoechst 33342 and 1:800 Whole Cell Stain (blue; Thermo Scientific) solution for 20 min. Cells were washed twice with PBS and imaged with a 10× objective using a PerkinElmer Operetta High Content Imaging System. 9–11 sites were imaged in each well. Image segmentation, analysis, and signal intensity quantitation were performed using Acapella software (PerkinElmer). Population-average and single-cell data were analyzed using MATLAB 2014b software. Single-cell density scatter plots were generated using signal intensities for individual cells.

For immunofluorescence analysis of xenograft and human tumor sections, processing and staining steps were performed with a customized protocol using a BOND RX automated immunohistochemistry processor (Leica Biosystems). Briefly, tissue slides were baked at 60°C for 30 min, de-waxed using Bond Dewax Solution (Leica Biosystems, Cat# AR9222), antigen retrieved for 20 min using BOND Epitope Retrieval solution 1 (Leica Biosystems, Cat# AR9961), and blocked with OBB for 30 min. Slides were incubated with secondary antibodies (Invitrogen): goat anti-mouse Alexa 647 (Cat# A21236) and goat anti-rabbit Alexa 555 (Cat# A21428) and scanned with a 10× objective using CyteFinder (RareCyte Inc.) to obtain pre-staining images for registering background due to non-specific binding of secondary antibodies (e.g., in necrotic regions of tumors). Slides were then incubated with a fluorophore inactivating solution containing 4.5% $H_2O_2$ and 20 mM NaOH in PBS for 2 h to

quench the background fluorescence (Lin *et al*, 2015). Slides were then incubated at 4°C overnight with primary antibodies in OBB, washed with PBS, and stained with the same secondary antibodies described above for 2 h at room temperature. Slides were re-scanned with a 10× objective using CyteFinder. Images were stitched together to afford a single aggregate immunofluorescence image of the whole area of all tumor sections per drug treatment group. Fluorescence images were flat-field corrected and single-cell fluorescence intensities were quantified using ImageJ software. Briefly, the Hoechst channel was used to generate single-cell masks and ROIs. For Ki-67 nuclear staining, the masks were applied and the mean/integrated pixel intensity per cell was obtained. For NGFR staining, the nuclear masks were then converted to ring-shaped ROIs (2-pixel wide), and the ROIs were applied to quantify NGFR signal intensities per cell. We then used pre-staining images (generated from slides stained with secondary antibodies only) to exclude high-background regions of the tumor (with fluorescence intensities higher than a separating threshold) from further analysis. Removal of high-background regions, and analysis of population-average and single-cell data were performed using MATLAB 2014b software.

## Western blots and quantitation

To prepare protein lysates, cells were seeded in 10-cm plates and treated the next day with either DMSO or two doses of vemurafenib (0.2, 0.32, or 1 µM) for 48 h at 37°C in full growth media. Cells were transferred to ice, washed with ice-cold PBS, and lysed with a 1% NP-40 lysis buffer (1% NP-40, 150 mM NaCl, 20 mM Tris pH 7.5, 10 mM NaF, 1 mM EDTA pH 8.0, 1 mM $ZnCl_2$ pH 4.0, 1 mM $MgCl_2$, 1 mM $Na_3VO_4$, 10% glycerol) supplemented with complete mini/EDTA-free protease inhibitors (Roche, Cat# 11836170001). Protein concentration of cleared lysates was measured using a BCA assay kit (Thermo Scientific, Cat# 23225). Lysates were adjusted to equal protein concentrations for each cell line, 4× NuPage LDS sample buffer (Invitrogen) supplemented with 50 mM DTT was added, and samples were heated for 10 min at 70°C and loaded on Novex 3–8% Tris–Acetate gels (Invitrogen). Western blots were performed using the iBlot Gel Transfer Stacks PVDF system (Invitrogen). After blocking with OBB (Licor), membranes were incubated with primary antibodies diluted 1:1,000 in OBB. As secondary antibody, donkey anti-rabbit IgG coupled to IRDye 800CW (Licor, Cat# 926-32213) was diluted 1:15,000 in OBB including 0.01% SDS. Membranes were scanned on an Odyssey CLx scanner (Licor) with 700 and 800 nm channels set to automatic intensity at a 169 µm resolution. Protein levels were quantified with the Image Studio 4.0.21 software (Licor) using the built-in manual analysis tool with a median local background correction. All intensity values were corrected using the respective HSP90α/β or β-actin levels and then normalized to the DMSO control sample for COLO858. Western blots were performed in three independent replicates, and a representative blot with quantification is shown.

## siRNA transfection

siRNAs against *JUN*, *PTK2*, and *NGFR* genes and a non-targeting control were from Dharmacon. COLO858 cells seeded in 96-well plates were transfected using transfection reagent DharmaFECT 2

(Dharmacon) in antibiotic-free growth medium. Cells were treated the next day with vemurafenib at indicated doses for 48–72 h followed by fixation for immunofluorescence staining.

## Compound screening using a chromatin-targeting library

A375, COLO858, and WM115 cells were plated in two replicates at 4,500, 7,000, and 8,000 cells per well, respectively; cells were counted using a Countess II FL Automated Cell Counter (Life Technologies). Cells were treated the next day with either DMSO or three different doses (0.11, 0.53, and 2.67 µM) of each of 41 compounds in the Harvard Medical School Library of Integrated Network-based Cellular Signature (HMS-LINCS) IV chromatin-targeting library (see https://lincs.hms.harvard.edu/db/libraries/LINCS-IV/ for the list of compounds) using previously prepared 384-well dilution plates and a Seiko pin transfer robot system. Immediately after, A375 and WM115 cells were then treated with 1 µM vemurafenib and COLO858 cells were treated with 0.32 µM vemurafenib using an HP D300 Digital Dispenser. Forty-eight hours after treatment, cells were fixed and analyzed for NGFR expression using immunofluorescence microscopy, as described earlier. To identify candidate compounds with potential for suppressing NGFR expression, the NGFR measurements were averaged across the three doses of chromatin-targeting compounds and *z*-scored. The data were scatter-plotted between the two replicates. Compounds whose NGFR levels were consistently among the five lowest *z*-scores in all the three cell lines were selected for further analysis.

## Statistical analysis

All data (with error bars) are presented as mean ± SD using indicated numbers of replicates. Statistical significance for cell-based experiments performed at different drug or growth factor doses, time points, and replicates was determined based on two-way analysis of variance (ANOVA). Correlation between measurements was evaluated based on pairwise Pearson's correlation coefficient. Statistical significance for xenograft experiments was determined using the two-tailed two-sample *t*-test. Significance was set at $P < 0.05$. Statistical analyses were performed using MATLAB 2014b software.

## Data availability

RNA sequencing data collected in this study were deposited to Gene Expression Omnibus (https://www.ncbi.nlm.nih.gov/geo/) under the accession number GSE87641. The data and methods used in this study are available in a machine-readable format to facilitate re-analysis by others at https://lincs.hms.harvard.edu/fallahi-sichani-molsystbiol-2017/.

**Expanded View** for this article is available online.

## Acknowledgements

This work was funded by NIH grants U54-HL127365, CA139980, and GM107618 (to PKS), and NCI grant K99CA194163 (to MF-S), a Merck Fellowship of the Life Sciences Research Foundation (to MF-S), grants from the Adelson Medical Research Foundation and the Melanoma Research Alliance (to LAG), the Ludwig Center at Harvard (to PKS and LAG), and a DFCI Wong Family Award

(to BI). We thank the Nikon Imaging Center and ICCB-Longwood Screening Facility at HMS, the BWH Pathology Core and J Reyes, SH Davis, KW Karhohs, G Lahav, G Berriz, J Muhlich, E Williams, D Wrobel, R Benezra, A Bradley, MW Covert, J Goedhart, A Miyawaki, C Shamu, NS Gray, C Yoon, LN Kwong, GM Murphy, and C Lian for reagents and assistance.

## Author contributions

MF-S designed the project and performed cell-based experiments and computational/statistical analysis. VB performed cell-based experiments. GJB performed mouse experiments. BI analyzed clinical samples. JRL analyzed clinical and xenograft samples. SAB assisted with gene expression analysis and PS and AR with cell culture. LAG and PKS supervised the research. All authors wrote and reviewed the manuscript.

## Conflict of interest

LAG is a member of the scientific advisory board for Warp Drive, Inc., consultant for Novartis, Inc., and equity holder in Foundation Medicine, Inc. All other authors declare that they have no conflict of interest.

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
