## [Review Process File · Molecular Systems Biology]

Adaptive resistance of melanoma cells to RAF inhibition via reversible induction of a slowly-dividing de-differentiated state

Mohammad Fallahi-Sichani, Verena Becker, Benjamin Izar, Gregory J Baker, Jia-Ren Lin, Sarah A Boswell, Parin Shah, Asaf Rotem, Levi A Garraway, Peter K Sorger

Corresponding author: Peter Sorger & Mohammad Fallahi-Sichani, Harvard Medical School

Review timeline:

Submission date:	10 January 2016
Editorial Decision:	07 March 2016
Revision received:	03 October 2016
Editorial Decision:	26 November 2016
Accepted:	26 November 2016

Editor: Thomas Lemberger

Transaction Report:

1st Editorial Decision

07 March 2016

The reviewers acknowledge that the topic of the study is relevant and they recognise the quality of the data. The overall level of support remains however rather on the cautious side as it stands. The reviewers feel that substantial additional evidence would be required to firmly establish the key conclusions of the paper and demonstrate their physiological relevance. We thus feel that the following major points raised by the reviewers should be convincingly addressed in a major revision of this work:

- Both reviewer #2 and #4 note that xenografts experiments would be necessary to assess the proposed combinatorial regimens.
- More direct evidence would be needed to show that NGFR-high cells are indeed slowly cycling.
- More direct evidence would be needed to demonstrate the resistance of the NGFRhigh sub-population.
- In general, the reviewers felt that the manuscript was sometimes difficult to follow.

REFeree REPORTS

Reviewer #1:

This manuscript of the Sorger group describes a very complex study in which the authors identify a subpopulation of melanoma cells, which are resistant to BRAF inhibition. This subgroup is characterized by high NGFR expression, and the authors explore drugs that block this cell-state such that a larger fraction of cells can be killed by BRAF inhibitors. The overall story and results are very interesting, but the story is told (especially in the second part of the manuscript) overly complicated and convoluted, making it very hard to follow and understand. I think also the title doesn't really reflect the content of the manuscript (why picking the BET inhibitor in the title which is a minor part of the manuscript?). And, most importantly, where I have a real issue is that the authors try to put forward the notion of the "slowly cycling" dedifferentiated NGFR high cells population that drive resistance. First, they don't show that NGFR high cells cycle slowly. What they show is that a subset of cells reenters cell-cycle during treatment. Why don't they show that NGFR cells cycle slowly? Second, they show that NGFR is upregulated during treatment and also high basal NGFR expression is correlated with re-entering cell cycle. But why don't they show that the fraction of NGFR high cells is actually resistant?

Detailed description and comments

The manuscript starts with reporting data from life-cell imaging of cell-cycle/apoptosis reporters in two cell melanoma cell lines. One cell line shows a subpopulation of cells that, when treated with the inhibitor, stop cycling for 5 days and then reenter cell cycle, while most other cells go into apoptosis. In the other cell line a similar number of cells is apoptotic, but does not re-enter the cell cycle. This is very interesting and a nice starting point for the paper. However, as stated above, I have some concern regarding the interpretation of it as a "slowly-cycling" state, which I am not convinced of. Much longer life-cell trajectories would be needed to show that they are indeed slowly cycling when treated with the drug, and not only transiently blocked.

Using a pre-treatment experiment, the authors then show convincingly that the cell line that shows cell cycle entry of a subpopulation become more resistant, when pre-treated for 24 hours with low doses of the drug and a drug cocktail of a MEK and BRAF inhibitor, while the other cell line shows no increased resistance. What is somewhat lacking is a biochemical control: Is the inhibitor acting in terms of low pERK levels when treated for 24hours + 72 hours? The only that is shown is shown for 48 hours treatment.

Next, transcriptome experiments were used to derive a hypothesis about the difference between the cell lines, and by GO-analysis they come up with the hypothesis that the resistant cells are in a less differentiated, more neuron like cell type. By measuring a candidate marker (NGFR), they show that three of their cell lines in a panel of 9 cell lines show high NGFR expression and an increased expression when treated with increased doses of BRAF inhibitor. FACS analysis shows that the cells surviving treatment show high NGFR expression and only few are KI67 positive, i.e. not in G0/G1. The data looks good and are in agreement with the idea that some NGFR high cells survive and do enter the cell cycle under treatment. They however do not support the idea of a "reciprocal relationship between NGFR expression and proliferation". What the data shows is that there is basically hardly any relation between NGFR and KI76 in individual cells, and if there is some correlation than it seems positive.

Using FACS sorting, cells were then sorted into NGFR high and NGFR low cells, and re-cultured. This is an elegant way to determine if cells stochastically flip between NGFR high and NGFR low states and if so on what timescale. It seems that after 6-9 days, the cells originating from the two populations obtained similar levels of NGFR, suggesting that cells retain their state for about a week. Using NGFR/KI67 co-staining, the authors show how the cells relax to a low NGFR and high proliferative state. This is for the first time that there is some (albeit weak) anti-correlation between NGFR/cell cycle. Having that said, looking at the NGFR low state, cell cycle recovery seems very similar. I think if this is going to be a main point of the paper, this has to be strengthened by much more direct measurements and a thorough analysis. As it is, one can interpret the data that drug treatment up-regulates NGFR and blocks cell cycle, and that a subpopulation re-enters cell cycle. What is also puzzling for me is why they don't show drug sensitivity of the NGFR high vs. low cells at early time points. This would be a direct prove that NGFR high cells are resistant. Instead, they show in Fig. 4D that after 9 days there is no difference.

Next, the authors analyze BRAF melanoma patient expression data, cell line data and show that

NGFR level correlates different differentiation markers and cell matrix/contact pathways, suggesting that NGFR might control differentiation and cell matrix/adhesion. Cell culture work shows that vemurafenib treatment, in addition to inducing NGFR also regulates genes involved in these processes. They then screen different FAK and SRC inhibitors if they block NGFR induction and they tend to do so.

From here on, the manuscript becomes hard to follow as now they basically screen for different pathways and drugs with different but overlapping methods and not a consistent drug panel throughout the next figures. I would guess the manuscript would largely benefit for concentrating on the main findings of Figures 6-9 or at least reorganizing those in targeted pathway inhibitors, targeted epigenetic drugs and cytokines, and either drop or move the rest to the supplement.

First they identify AP1/JUN as one of the potential transcription factors that may drive NGFR high state by using TFBS enrichment (it remains unclear why they follow up AP1 and not e.g. BACH1, which ranks similarly high in both lists A+C). Using correlation analysis, they show that c-Jun expression correlates with NGFR expression (Figure 6D). Drug treatment with a JNK inhibitor reduces both NGFR levels, but does not influence pERK levels, and reduces the number of proliferative cells (6H). Here, somewhat surprisingly after reading this story, there are quite a few NGFR low cycling cells in the JNK/BRAF double treatment, as opposed to the combinatorial MEK/BRAF treatment, where hardly any cells proliferate. I think the authors need to comment on this, as this goes against their hypothesis. The fact that the authors again add the effects of FAK/SRC in figure 6H/I is very confusing and disturbs the flow of the manuscript. Why not add it to Figure 5?

The authors go on to test now combinations (double and triple) on cell viability, NGFR expression, proliferation and signaling (Figure 7). They establish that the combination of the SRC, FAK, JNK inhibitors are reducing growth viability when compared to BRAF inhibitors or BRAF+MEK inhibitors. It remains a bit unclear if there is a real synergy between the additional inhibitors and the MAPK-pathway inhibitors, as the inhibitors themselves already had quite some effect (Figure 7A). For the combination it cannot be judged, as the raw data is not displayed.

Somewhat surprisingly, the authors then argue that combination of target therapies are not interesting as they tend to be toxic and then open a next arena in which they investigate drugs modulating epigenetics (which is then mentioned in the title, but in the paper it remains a minor point). Using a screen they get hits for two HDAC inhibitors and three BET inhibitor. For an unknown reason they then dropped the second most potent inhibitor and concentrate on Belinostat and three BET inhibitors, for which they show that they reduce both growth and cell cycle and NGFR expression. One of the BET inhibitors and one of the HDAC inhibitors show quite drastic cell killing. For some reason, again phenotypic data for targeted inhibitors is discussed and shown in Figure 8, which I guess should be moved to a more appropriate place in the manuscript.

Next the authors investigated the effect of cytokines HGF and TNF on viability, cell cycle and KI67. The data shows that in two cell lines different ligands have basically different effects on viability, proliferation, and NGFR expression. I find this section rather inconclusive.

Major comments

The manuscript needs some reorganization starting from Figure 6 to aid the reader, in the moment it is very hard to follow.

The authors should show directly that the NGFR^{high} state is actually slow cycling, and not adapting to drug treatment and just re-entering cell cycle.

The authors need to convincingly show with an experiment similar to the one presented in Figure 4D that NGFR high cells are actually resistant.

Minor comments

It would be very beneficial if the targets for the targeted therapies would be always indicated in the figure, otherwise a non-expert gets lost with so many different inhibitors.

The percentages of cells in the FACS dot-plots should be given, otherwise it is really hard to judge these.

Figure 6D

would an outlier robust correlation measure such as spearman correlation also yield a similarly strong correlation between Jun and p-ERK?

Figure 6D,B E,G and I

1. It is well known that JNK regulates Jun phosphorylation, in Figure 6D,B E,G and I, it seems to be assumed by the authors that JNK also chiefly regulates Jun mRNA level. Is this the case?

Page 16 lines 9-10 " ...in MZ7MEL and MMACSF cells, HGF or NRG1 increased the fraction of Ki-67 High cells...but did not change NGFR expression"

-> judging from figure 9C only MZ7MEL shows an increase in Ki67 high cells only for HGF whereas TNF increased NGFR High cells in MMACSF!!!

Why is the last panel in Figure C empty although some cells can be spotted in the FACS plot in Fig 9B for those cases.

Why were different concentrations of Vem used in Figure B for the two cell lines?

page 7 line 16-17:" ...cell cycle genes ... in COLO858 were upregulated again by t=48h (Figures 3A, 3B)..."

-> they are not upregulated but restored to their original level

page 7 line 22: In two out of eleven additional BRAF V600E/D melanoma cell lines tested (A375 and WM115,Figure 3C)...

-> in Figure 3C in addition to COLO858 and MMACSF only seven more cell lines are shown

Figures 5G/H

page 11 line 8-11 "These proteins... were present at higher levels, or were substantially up-regulated by vemurafenib in COLO858, A375 and WM115 ... relative to MMACSF and MZ7MEL "

1. pFAK is neither substantially upregulated nor higher than MMACSF, and Integrin β 1 in COLO858 and both Vem-sensitive cell lines seems not higher as MZ7MEL or much stronger induced as in MMACSF. These exceptions should be correctly pointed out and discussed
2. The more surprising it seems that FAK and SRC inhibitors can down regulate all three resistant cell lines NGFR in Fig5 I.

Figure 7C

1. arrows point to the lower FAK but should for consistency point in the space between the two FAK inhibitors (will not weaken the point of the authors)

Figure 7D

1. The encircled treatments seem to be close to the 0 of the first two PCs. Does this not mean they can not be well described by the loadings shown on the left?
2. It should further be pointed out that the red line in the first plot denotes the cumulative variance.

Figure 9

For easier comprehension the order in Figure A, B and C of cell line and treatment should be the same in all three figures!

Reviewer #2:

This manuscript studies the heterogeneous drug response to melanoma cells treated with RAF/MEK inhibition. The study identifies a fraction of slowly cycling NGFR-high de-differentiated cells, which can potentially develop a reversible drug-resistant phenotype. The authors also show that BET and HDAC inhibitors block the de-differentiation and restore the drug sensitivity.

While the study is certainly interest in showing subpopulation of cells having different drug responses, there are several shortcomings of this study. In many cases of gene expression analyses,

there are only subtle changes and there is a lack of consistence across the various figures in the manuscript. Secondly, the viability assays are primarily based on the short-term assays. The study would also greatly benefit from xenograft experiments, especially in the light of the suggestion that ECM and cytokine signaling are relevant for the emergence of the slow growing drug tolerant cell populations.

Major Comments:

Are these slowly-cycling cells having incomplete MAPK inhibition? Can the authors use a FACS-based assay to zoom in to the p-ERK level in slowly growing cells?

1. The study makes the link between NGFR expression and MAPKi-resistance. It would be interesting to investigate whether NGFR signaling is directly connected with MAPKi-resistance. For example, over expressing NGFR and/or add the ligand, and see whether these confer resistance to MAPK inhibitors, or whether knocking-down NGFR increases the sensitivity to MAPK inhibitors?
2. NGFR and Ki-67 stainings in Fig4E have very poor anti-correlation. This is not consistent with what the authors show in Fig3D.
3. The study identifies activation of the focal adhesion kinase (FAK) upon BRAF inhibition. However, of 5 melanoma cell lines tested, only Colo858 showed a significant activation of FAK. In general, the authors should comment on how often heterogeneous staining for NGFR is seen in primary tumors and relate this to the heterogeneity of their cell line panel in having this slow growth population emerge during drug resistance
4. In Fig 5I, NGFR is not significantly up-regulated upon BRAF inhibition, this piece of data is not consistent with the previous data, Fig3C, D
5. In Fig 8F, the drug combination effect of HDACi/BETi and BRAFi is quite weak. In COLO858, HDACi/BETi even sort of rescue the cells from high dose of MAPKi. An exposure of 48 hours to any drug that acts through chromatin effects seems very short and consequently unlikely to be mediated by chromatin-remodeling effects. Can the authors comment on this?
6. All the cell viability data in this study are relative short-term assay (72-96 hours); authors should also provide the long-term assay such as colony formation assay to support their conclusions. Incucyte experiments for some of the seminal observations would be very helpful, especially the experiments with epigenetic modifiers.
7. The authors make no reference to the work of the group of Jeff Settleman (Cell 141, 69-80, 2010), which identifies a similar slow growing drug resistant cell population. The authors should compare the features of their cells with those identified by Sharma et al. For instance, the role of KDM5A could be studied in the melanoma model also.

Reviewer #3:

Fallahi-Sichani et al used multiplexed signaling, single cell, and transcriptomic analysis to analyze early adaptive events (first 48 hrs) that may impact the emergence of a BRAF inhibitor inducible slow-cycling sub-population of de-differentiated melanoma cells. They concluded that a FAK-SRC-cJUN signaling axis induced by BRAF or BRAF/MEK inhibition promotes de-differentiation as marked by CD271/NGFR induction and drug-tolerant slow-cycling melanoma cells with mesenchymal features. The authors also concluded that inhibitors of FAK, SRC and BET bromodomain selectively target this therapy-induced slow-cycling subpopulation to enhance the Emax of killing.

The descriptive aspects of this study have been reported extensively in the literature. - cJUN has been implicated as a critical adaptive response to BRAF inhibitors. The same authors (Molecular Systems Biology 2015) have shown that BRAF inhibition induced c-JUN and JNK and

BRAF inhibitors enhanced the Emax of melanoma cell killing. Delmas et al (Oncotarget 2015) showed that BRAF inhibition transcriptionally and adaptively up-regulated cJUN, leading to RHOB and AKT induction. Ramsdale et al (Science Signaling 2015) reported that both intrinsic and adaptive BRAF inhibitor resistance are driven by increased expression of cJUN, which induces mesenchymal de-differentiation.

- That BRAF inhibitor can induce a reversible and slow-cycling drug tolerant subpopulation marked by loss of differentiation, induction of CD271, and histone modifications has been recently reported (Menon et al, Oncogene 2014; Zubrilov et al, Cancer Letters 2015).
- Integrin beta1/FAK/SRC have been shown to participate in BRAF inhibitor resistance (Hirata et al, Cancer Cell 2015; Girotti et al, Cancer Discovery 2013).
- The compound JQ1 has been shown to be synergistic with BRAFi or combined BRAFi and MEKi in growth-inhibiting BRAF mutant melanoma cells. The proposed mechanism was inhibition of bromodomain proteins such as BRD4 which are critical for MYC mRNA expression (Korkut et al, eLIFE 2015).

The mechanistic implications of BRAF combinatorial targets (FAK-SRC-cJUN and epigenetic targets) were under-developed. It was not clear how much SRC and cJUN activation was dependent on FAK activation. Do AP-1 dependent transcription and cell adhesion related survival constitute a feed-forward signaling loop? How critical was FAK in this loop? Knockdown/genetic approaches and clear demonstration of the targets of inhibitors (dose-response) used were missing. How significant was the effect of FAK/SRC inhibitors on NGFR levels (Figure 5I) responsible for the cell viability effects (Figure 7A)? Since FAK/SRC inhibitor co-treatment with BRAF inhibition in A375 and WM115 still left the cells with plenty of NGFR protein, why would co-treatment enhance the Emax? Most importantly, it appeared that the action of bromodomain and HDAC inhibitors such as JQ1 had little to do with the cJUN-NGFR link. For instance, in the prototypic line used in this study (Colo858), the HDAC inhibitor belinostat induced NGFR without BRAF inhibition (Figure S8B) and, in the presence of I-BET and JQ1, BRAF inhibitor still induced cJUN (Figure S8D). Also, the authors tried to show us that the small molecule inhibitors used against FAK-SRC-JNK were hitting the nominal targets in Figure 7D using A375. If one looks at Figure 5G and 5H, in A375, BRAF inhibitor did not induce p-FAK, alpha-TSP-1 or integrin beta1. In the cell line where the BRAF inhibitor marginally induced these levels (Colo858), the PCA looked extremely weak (Figure S7C).

Beyond the transcriptomic analysis for the two cell lines Colo858 and MMACSF, additional analysis of publicly available RNASeq data was performed. However, the results shown were selective or did not optimally highlight the significance of the Colo858 analysis. For instance, are the authors postulating that the transcriptomic alterations induced by BRAF inhibition in Colo858 (representative of adaptive resistance with an inducible slow-cycling population) are indeed happening within weeks 1 or 2 on treatment in patient-derived biopsies. If this were the case, then why did the GSEA data in Figure 5E only showed enrichment of one process. Did other processes/gene sets (shown in Figure 3B) simply did not meet the FDR cutoff? It is understandable that the authors could not perform correlation with three pairs of RNASeq from pre and post resistance tumor tissues (Figure 5F). However, what was the rationale for the selection of the particular genes' expression levels? The authors should show the genes such as those in Figure 5B and 5C or others from Figure 3 to support their claim that this in vitro BRAF inhibitor inducible subpopulation are present and detectable in acquired BRAF inhibitor resistance in patient-derived tissues. Two larger RNASeq data sets of this kind are also available now in the literature if the authors want to investigate this further.

How quantitative and linear are the signals in the authors' use of immunofluorescence? For example, the authors showed in Figure 3C that higher graded levels were readily detectable (A375, Colo858, WM115) but lower levels may not be (see the remainder six cell lines). The authors should probe NGFR levels and BRAF inhibitor effects by Western blots as well.

The authors should include data on cell lines treated with JQ-1 by itself in Figure 8G. As seen in Figure 8F, JQ-1 by itself can have negative effects on melanoma cell viability. This would suggest that the effects of JQ-1 in the presence of BRAF inhibition was not specific to the induction of alternate growth and differentiation state.

Data presented in Figure 9 are too preliminary and disconnected to the rest of the manuscript to warrant placing this at the end. Are the authors claiming that the 10% increase in viability with TNF

stimulation during BRAF inhibitor treatment in the one cell line MMACSF is the result of NGFR up-expression? By the same token, if the authors remove or sort out the NGFR subpopulation in A375, would the remainder population be more sensitive to BRAF inhibition?

On a broader level, while it seems important to achieve a greater Emax, what is the biologic significance of this slow-cycling subpopulation induced by BRAF inhibition. Does it merely serve as niche for expansion of truly drug-resistant clones or does it serve as the precursor to at least one form of acquired resistance? Does this NGFR high population give rise to the MITF low population that has been shown to resume proliferation in the presence of BRAF inhibition (Konieczkowsky et al *Cancer Discovery* 2014; Muller et al, *Nature Communication* 2014; Riesenberget al, *Nature Communication*, 2015)?

Reviewer #4:

Vemurafenib-treated cells were evaluated for cycling characteristic and for patterns of drug sensitivity. Vemurafenib induced 20% of COLO858 cells to enter a slow cycling state that could be reversed with drug washout. This subset of cells was resistant to apoptosis but did not reactivate MAPK signaling. Transcriptional profiling of vemurafenib-induced resistant cells identified a number of markers characteristic of less-differentiated neural crest cells, including p75. And, in populations, p75 expression marked slow cycling cells. Some human melanoma specimens were mosaic for p75 expression, which was complementary to MITF and Ki67. Transcription profiling of vemurafenib-treated responsive and resistant cell lines, and analysis of patient samples and cell lines, identified extracellular matrix and focal adhesion processes as candidates. Inhibitors targeting JNK, SRC, and FAK were especially effective in combination with vemurafenib on cell lines with tonic or vemurafenib inducible NGFR. A subset of epigenetic inhibitors downregulated NGFR expression, and these inhibitors combined for greater effect with vemurafenib. In combination with vemurafenib, FAK inhibitor and JQ1, each suppressed the slow-cycling population and increased the fraction of apoptotic cells.

Phenotypic interconversion is an important and common mechanism for cancer drug resistance. The manuscript established the means by which some melanoma cell lines develop resistance, and also addresses the important practical issue of how to combine targeted therapies to attack p75-expressing melanoma cultures. The use of transcriptional evaluation combined with selective drug screening is reasonable and includes considerable computational analysis appropriate for this journal. The wet bench work is technically excellent, and computational analyses are used appropriately.

Some biological issues are left on the table:

1. The role of p75 itself is not assessed; is it simply a biomarker for this population, or does it have a signaling or other function?
2. Immunoblotting supports the importance of ECM/FAK/SRC proteins, but the exclusive use of pharmacologic agents (with off-target activities) rather than genetic interventions to evaluate integrin/FAK/Src signaling leaves some uncertainty about mechanism.
3. The ability of any of the combinations to control tumor growth without excessive toxicity has not been evaluated. Ideally, effect of one of an integrin/FAK/Src inhibitor combination would be tested on p75-inducible and non-inducible cell lines in a xenograft mouse model.

Other questions.

1. Is there any information about how slow cycling is maintained?
2. The term "dedifferentiation", seems overly strong without more evidence on the phenotypes of mature and "dedifferentiated" cells, and their relationship to normal neural crest lineage compartments.

Response to Review of MSB-16-6796

“Adaptive resistance of melanoma cells to RAF inhibition via reversible induction of a slowly-dividing de-differentiated state”

As described in the cover letter, the revised version of our manuscript has been completely re-written, the figures have been redrawn and a substantial body of new data has been obtained from cell lines, xenograft tumors and human biopsies. Please find below the detailed, point-by-point response to the concerns raised by the reviewers.

Reviewer #1:

This manuscript of the Sorger group describes a very complex study in which the authors identify a subpopulation of melanoma cells, which are resistant to BRAF inhibition. This subgroup is characterized by high NGFR expression, and the authors explore drugs that block this cell-state such that a larger fraction of cells can be killed by BRAF inhibitors. The overall story and results are very interesting, but the story is told (especially in the second part of the manuscript) overly complicated and convoluted, making it very hard to follow and understand. I think also the title doesn't really reflect the content of the manuscript (why picking the BET inhibitor in the title which is a minor part of the manuscript?). And, most importantly, where I have a real issue is that the authors try to put forward the notion of the "slowly cycling" dedifferentiated NGFR high cells population that drive resistance. First, they don't show that NGFR high cells cycle slowly. What they show is that a subset of cells reenters cell cycle during treatment. Why don't they show that NGFR cells cycle slowly? Second, they show that NGFR is upregulated during treatment and also high basal NGFR expression is correlated with re-entering cell cycle. But why don't they show that the fraction of NGFR high cells is actually resistant?

1.1. We appreciate that the reviewer finds the story of the manuscript interesting but that the narrative, as presented, was difficult to follow. In response, we have completely re-written the manuscript, reorganized and re-drawn the figures and added additional data when needed to address specific questions and concerns. To the extent possible we have moved data that is supportive but not essential to the main thread of the paper to the supplements. We believe that these changes address all of the concerns raised in review, and made the manuscript easier to follow. It remains true, however, that this paper attempts a rather unfamiliar integration of single-cell imaging studies with more conventional fixed-cell dose response data.

1.2. We agree with the reviewer that the primary focus of this paper is its analysis of the heterogeneity of drug response and its implications for understanding drug-adapted, slow-growing cells. The co-drugging studies at the end of the paper are informative, primarily as a means of understanding the drug-adapted state and BET inhibitors are only one tool in the study. We have therefore removed BET inhibitors from the title of the manuscript and de-emphasized this drug class. In addition, although we now include xenograft studies in the revised manuscript, we have de-emphasized our previous argument that our data support co-drugging of human tumors with BET inhibitors or inhibitors of FAK or SRC kinases. We accept that large-scale tumor-recurrence studies in animals will be required to support such an assertion. As a result, the introduction and discussion to the paper have been completely re-written to emphasize the basic science.

1.3. In response to the reviewer's comment that more direct evidence is needed to demonstrate the slow growth of NGFR^{High} cells and their resistance to RAF inhibition, we have performed a new experiment that directly measure and compare growth rate and doubling time in freshly sorted NGFR^{High} and NGFR^{Low} cells following drug-induced NGFR induction in COLO858 melanoma cells (Figure 5). The data show that freshly sorted NGFR^{High} COLO858 cells cycle more slowly than NGFR^{Low} cells during the first 2-5 days after sorting as predicted. We have also compared directly the sensitivity of freshly sorted NGFR^{High} and NGFR^{Low} cell subpopulations to vemurafenib using a conventional three-day assay. The data were scored by measuring vemurafenib-induced growth rate (GR) inhibition, a method recently published in *Nature Methods* that is essential for correcting drug sensitivity measurements for confounders, differences in growth rates in particular (Hafner et al, 2016). Using this metric, we show directly that NGFR^{High} COLO858 cells are less sensitive to RAF inhibitor treatment than NGFR^{Low} cells (Figure 5).

Detailed description and comments:

The manuscript starts with reporting data from life-cell imaging of cell-cycle/apoptosis reporters in two melanoma cell lines. One cell line shows a subpopulation of cells that, when treated with the inhibitor, stop cycling for 5 days and then reenter cell cycle, while most other cells go into apoptosis. In the other cell line a similar number of cells is apoptotic, but does not re-enter the cell cycle. This is very interesting and a nice starting point for the paper. However, as stated above, I have some concern regarding the interpretation of it as a "slowly-cycling" state, which I am not convinced of. Much longer life-cell trajectories would be needed to show that they are indeed slowly cycling when treated with the drug, and not only transiently blocked.

1.4. In response to this concern we have now performed long-term live single-cell analysis using both high-resolution single-cell tracking and an inside-incubator microscope that minimizes extraneous perturbation of cell cultures. Imaging was performed long enough (7-8 days) that untreated cells had reached confluence and drug treated cells had either reached a steady number or experienced at least three logs of killing. Among the population of live cells tracked individually between days 4 and 8, approximately half of the cells did not divide at all and the other half divided only once with minimal estimated doubling time of 48-96 h (Figure 1G). This ratio of non-cycling versus cycling cells (approximately 1:1) and their doubling time are comparable with cells tracked within the first 4 days of treatment, suggesting that drug-adapted COLO858 cells were not transiently arrested but actually continued to grow slowly in the presence of vemurafenib. We have revised the manuscript to suggest that dividing and non-dividing cells probably co-exist in the population, consistent with data showing that total viable cell number is relatively constant over time.

Using a pre-treatment experiment, the authors then show convincingly that the cell line that shows cell cycle entry of a subpopulation become more resistant, when pre-treated for 24 hours with low doses of the drug and a drug cocktail of a MEK and BRAF inhibitor, while the other cell line shows no increased resistance. What is somewhat lacking is a biochemical control: Is the inhibitor acting in terms of low pERK levels when treated for 24hours + 72 hours? The only that is shown is shown for 48 hours treatment.

1.5. The reviewer raises an extremely important point: acquisition of the drug-adapted state is accompanied by continuous and efficient inhibition of MAPK signaling. To demonstrate this we now include the data on p-ERK changes over time (from 24 to 72 h) (Figures 2A, 2B, Appendix Figures S2A, S2B). The data shows that in vemurafenib-treated COLO858 cells p-ERK levels remained inhibited between $t = 24$ to 72 h. We have also revised to the manuscript in multiple places to make this important point clearer.

Next, transcriptome experiments were used to derive a hypothesis about the difference between the cell lines, and by GO-analysis they come up with the hypothesis that the resistant cells are in a less differentiated, more neuron like cell type. By measuring a candidate marker (NGFR), they show that three of their cell lines in a panel of 9 cell lines show high NGFR expression and an increased expression when treated with increased doses of BRAF inhibitor. FACS analysis shows that the cells surviving treatment show high NGFR expression and only few are KI67 positive, i.e. not in GO/G1. The data looks good and are in agreement with the idea that some NGFR high cells survive and do enter the cell cycle under treatment. They however do not

support the idea of a "reciprocal relationship between NGFR expression and proliferation". What the data shows is that there is basically hardly any relation between NGFR and KI67 in individual cells, and if there is some correlation than it seems positive.

1.6. We apologize for the confusion – our writing was sloppy. We have revised the manuscript to better describe the relationship between NGFR expression and cell cycle progression using the following experimental data, some of which is newly collected:

(i) Time-course single-cell analysis of COLO858 cells co-stained for NGFR and Ki-67 between 24-72 h: This analysis shows how drug-adapted slowly-growing NGFR^{High} cells arise (Figure 4D). 24 h of vemurafenib exposure induces a shift in the cell population from a predominantly (>80%) proliferative Ki67^{High}/NGFR^{Low} state toward an arrested Ki67^{Low}/NGFR^{Low} state. Arrested cells then begin to up-regulate NGFR and shift to a Ki67^{Low}/NGFR^{High} state (by t = 48 h) after which they gradually re-enter the cell cycle. By t = 72 h, >90% of cells are NGFR^{High} and ~38 % of the NGFR^{High} cells are also Ki-67^{High} (Ki67^{Low}/NGFR^{High}). These data are consistent with live single-cell imaging and RNA-seq analysis showing that drug-adapted NGFR^{High} cells emerge following a transient vemurafenib-induced arrest and then re-enter the cell cycle with a slower rate than drug-naïve cells.

(ii) Direct measurement of growth rate and doubling time (and also the rate of Ki-67 up-regulation) in freshly sorted NGFR^{High} and NGFR^{Low} cell subpopulations as described above (Figure 5).

Using FACS sorting, cells were then sorted into NGFR high and NGFR low cells, and re-cultured. This is an elegant way to determine if cells stochastically flip between NGFR high and NGFR low states and if so on what timescale. It seems that after 6-9 days, the cells originating from the two populations obtained similar levels of NGFR, suggesting that cells retain their state for about a week. Using NGFR/KI67 co-staining, the authors show how the cells relax to a low NGFR and high proliferative state. This is for the first time that there is some (albeit weak) anti-correlation between NGFR/cell cycle. Having that said, looking at the NGFR low state, cell cycle recovery seems very similar. I think if this is going to be a main point of the paper, this has to be strengthened by much more direct measurements and a thorough analysis. As it is, one can interpret the data that drug treatment up-regulates NGFR and blocks cell cycle, and that a subpopulation re-enters cell cycle. What is also puzzling for me is why they don't show drug sensitivity of the NGFR high vs. low cells at early time points. This would be a direct prove that NGFR high cells are resistant. Instead, they show in Fig. 4D that after 9 days there is no difference.

This is a very important point and, as mentioned above (see points 1.3 and 1.6), we have performed new experiments in which we directly measure and compare the growth rate, doubling time and drug sensitivity of freshly sorted NGFR^{High} and NGFR^{Low} cells (Figure 5). These experiments fully address the concerns of the reviewer.

Next, the authors analyze BRAF melanoma patient expression data, cell line data and show that NFGR level correlates different differentiation markers and cell martrix/contact pathways, suggesting that NGFR might control differentiation and cell matrix/adhesion. Cell culture work shows that vemurafenib treatment, in addition to inducing NGFR also regulates genes involved in these processes. They then screen different FAK and SRC inhibitors if they block NGFR induction and they tend to do so. From here on, the manuscript becomes hard to follow as now they basically screen for different pathways and drugs with different but overlapping methods and not a consistent drug panel throughout the next figures. I would guess the manuscript would largely benefit for concentrating on the main findings of Figures 6-9 or at least reorganizing those in targeted pathway inhibitors, targeted epigenetic drugs and cytokines, and either drop or move the rest to the supplement.

1.7. We accept that the paper was too confusing about these studies as previously written. We have re-organized the figures and text to follow the two cell lines for which the most extensive and complete data were collected (Figures 1-8) and have moved the study of particularly promising drugs on additional cell lines into Figure 9. All *in vivo* data has been consolidated in Figures 10 and 11. We hope that this makes the story easier to follow.

First they identify AP1/JUN as one of the potential transcription factors that may drive NGFR high state by using TFBS enrichment (it remains unclear why they follow up AP1 and not e.g. BACH1, which ranks similarly high in both lists A+C). Using correlation analysis, they show that c-Jun expression correlates with NGFR expression (Figure 6D). Drug treatment with a JNK inhibitor reduces both NGFR levels, but does not influence pERK levels, and reduces the number of proliferative cells (6H). Here, somewhat surprisingly after reading this story, there are quite a few NGFR low cycling cells in the JNK/BRAF double treatment, as apposed to the combinatorial MEK/BRAF treatment, where hardly any cells proliferate. I think the authors need to comment on this, as this goes against their hypothesis. The fact that the authors again add the effects of FAK/SRC in figure 6H/I is very confusing and disturbs the flow of the manuscript. Why not add it to Figure 5?

1.8. We apologize for the confusion regarding how the AP1/JUN transcription factors were selected as transcriptional regulators of the NGFR^{High} phenotype for the follow-up experiments. We summarized a fairly extensive and thorough analysis in an entirely inadequate and telegraphic paragraph. As described in the revised manuscript, the selection of AP1/JUN was made in two stages. In the first step, we used transcription factor binding site analysis on the promoters of genes differentially regulated between COLO858 and MMACSF cells to identify a list of potential transcriptional regulators that might induce the NGFR^{High} phenotype (Figure 6D). AP1 emerged as the top candidate based on *P* values. However, as the reviewer correctly points, other transcription factors (such as BACH1) were also identified among the statistically significant candidates. It is not possible to follow up such a large number of genes in a principled way. Thus, in the second step of the analysis, the list of candidates was further narrowed down by analyzing how vemurafenib influenced the expression of transcription factor genes (including AP1, BACH1, BACH2, etc.) in COLO858 and MMACSF cells (Figure 6E). This analysis identified JUN as the most differentially expressed transcription factor between COLO858 and MMACSF cells. It was up-regulated ~9-fold by 24 h, and ~23-fold by 48 h of vemurafenib exposure in COLO858 cells but remained almost unchanged in MMACSF cells. Other transcription factor candidates either did not change significantly in expression level in either of the cell lines (for example BACH1), or changed in both cell lines in the first 24 h in an undifferentiable manner. We have revised the manuscript to better explain this point and believe that the available data justify our focus on AP1 factors.

1.9. We have also revised the manuscript to better clarify the impact of JNK, FAK and Src inhibitors versus MEK inhibitors on the proportion of NGFR^{High} and proliferating cells. As the reviewer correctly pointed out, both RAF/MEK inhibitor combination and RAF/JNK inhibitor combination are able to reduce cell viability and proliferation in melanoma cells. However, the two combinations seem to act very differently from one another; the impact of MEK inhibitor on cell viability and proliferation is most likely via better inhibition of the ERK pathway and it leads to NGFR upregulation in COLO858, A375 and WM115 cells. In contrast, the impact of JNK, FAK and Src inhibitors on cell viability and proliferation is independent of ERK signaling in the three cell lines we tested; these drugs appear to act by blocking or eliminating NGFR^{High} cells. Particularly intriguing is the observation, across multiple lines, that the more thorough the MAPK blockade (e.g. with RAF and MEK inhibition using vemurafenib plus trametinib), the more powerful the adaptive response and the greater the added benefit of blocking the JNK/FAK/SRC pathway. We have tried to make these key points more clearly in the manuscript and have extended and improved the visualization of biochemical data using 2D landscapes (Figures 7B, 7C and EV4). These clearly show that MEK inhibitor trametinib and JNK, FAK and Src inhibitors are orthogonal in their activities: the former decreases Ki-67 levels but increases the fraction of NGFR^{High} cells, whereas

JNK, FAK and Src inhibitors in combination with vemurafenib all reduce both NGFR and Ki-67 levels. Overall these are key findings that not only show that JNK, FAK and Src inhibitors can sensitize NGFR^{High} cells to RAF inhibition but also predicts that they would be more effective when combined with RAF/MEK inhibitor combination where a larger fraction of residual cells are NGFR^{High}. We thank the reviewer for pointing out the need to revise the text and the order of figure panels to make the argument more clearly.

The authors go on to test now combinations (double and triple) on cell viability, NGFR expression, proliferation and signaling (Figure 7). They establish that the combination of the SRC, FAK, JNK inhibitors are reducing growth viability when compared to BRAF inhibitors or BRAF+MEK inhibitors. It remains a bit unclear if there is a real synergy between the additional inhibitors and the MAPK-pathway inhibitors, as the inhibitors themselves already had quite some effect (Figure 7A). For the combination it cannot be judged, as the raw data is not displayed.

First, with respect to the release of raw data, we are committed to providing all of the values for our extensive assays on multiple cell lines as source data. This will facilitate re-analysis by others since, as the reviewer points out, dose-response data are subtle and extensive, precluding a full description in the text.

1.10. We agree with the reviewer that JNK, FAK and Src inhibitors show different single-drug effects on different cell lines, a not unexpected finding, but a comparison of their effects in combination with vemurafenib or vemurafenib plus trametinib on NGFR^{High} (COLO858, A375 and WM115) versus NGFR^{Low} cell lines (MMACSF and MZ7MEL) reveals a significant and systematic difference in effectiveness in conventional 3-day cell viability experiments. Newly added “long-term” time-lapse live-cell imaging confirms true synergy for at least some of the combinations, JQ1 in particular since it is not measurably cytotoxic on its own at concentrations tested (Figures 8E, 8F, EV5 and Movies EV2 and EV3). Moreover, in agreement with conclusions from fixed cell data, time-lapse live-cell analysis shows a continuous reduction in the number of live cells in response to defactinib in combination with vemurafenib and more effectively in combination with vemurafenib plus trametinib, whereas neither defactinib, nor vemurafenib alone were able to reduce the number of cancer cells below that of the pre-treatment (day 0) condition (Appendix Figure S5F).

Somewhat surprisingly, the authors then argue that combination of target therapies are not interesting as they tend to be toxic and then open a next arena in which they investigate drugs

modulating epigenetics (which is then mentioned in the title, but in the paper it remains a minor point). Using a screen they get hits for two HDAC inhibitors and three BET inhibitors. For an unknown reason they then dropped the second most potent inhibitor and concentrate on Belinostat and three BET inhibitors, for which they show that they reduce both growth and cell cycle and NGFR expression. One of the BET inhibitors and one of the HDAC inhibitors show quite drastic cell killing. For some reason, again phenotypic data for targeted inhibitors is discussed and shown in Figure 8, which I guess should be moved to a more appropriate place in the manuscript.

1.11 We have completely eliminated this line of argument. It was based on the rather flimsy premise that getting three kinase inhibitors into melanoma patients was likely to cause unwanted toxicity. Not only have we now eliminated these therapeutic claims from the paper (agreeing that they are not sufficiently substantiated), but also we motivate analysis of epigenome-modifying drugs in a different and more historically accurate way. Namely, that drug persisters (DTPs – (Sharma et al, 2010)) have previously been shown to be sensitive to drugs that inhibit epigenome modifying enzymes such as histone demethylases or histone deacetylases. Moreover, the 9 day reset time we observe for NGFR^{High} adapted cells is reminiscent of epigenetic remodeling.

1.12. We apologize for the confusion that arose regarding how BET inhibitors were selected from the epigenetic targeting library screen. Among the 41 epigenome-modifying agents, 3 BET inhibitors were selected because they consistently suppressed NGFR and Ki-67 expression when combined with vemurafenib in different technical and independent biological replicates. The HDAC inhibitor pracinostat suppressed NGFR expression in the initial screen, but did not reduce cell proliferation (as measured by Ki-67). Belinostat was also identified in the initial screen, but its effect on NGFR expression was minimal for the COLO858 cell line when re-tested in a biologically independent experiment. We therefore focused on BET inhibitors, because three of them (JQ1, I-BET and I-BET151) were effective in suppressing the NGFR^{High} state and preventing adaptive resistance to RAF/MEK inhibition. The revised manuscript better clarifies how BET inhibitors were selected and to avoid confusion, we have removed unnecessary information about pracinostat and belinostat from the text. However, the full set of screen data collected for 41 chromatin-targeting small-molecules and three cell lines will be provided in full in the supplements (source data associated with figures).

Next the authors investigated the effect of cytokines HGF and TNF on viability, cell cycle and KI67. The data shows that in two cell lines different ligands have basically different effects on

viability, proliferation, and NGFR expression. I find this section rather inconclusive.

1.13. Based on the comments of several reviewers, we have removed this part of the study from the manuscript. We agree with the reviewer that the data is preliminary and can be removed without altering the key conclusions of the paper.

Major comments

The manuscript needs some reorganization starting from Figure 6 to aid the reader, in the moment it is very hard to follow.

We agree - please see comments above (points 1.1 and 1.7).

The authors should show directly that the NGFR^{high} state is actually slow cycling, and not adapting to drug treatment and just re-entering cell cycle.

We agree – and this has now been done. Please see points 1.3, 1.4 and 1.6.

The authors need to convincingly show with an experiment similar to the one presented in Figure 4D that NGFR high cells are actually resistant.

We agree – and this has now been done. Please see point 1.3.

Minor comments

It would be very beneficial if the targets for the targeted therapies would be always indicated in the figure, otherwise a non-expert gets lost with so many different inhibitors.

We have added the nominal targets for molecule kinase inhibitors in figure panels to aid with clarity.

The percentages of cells in the FACS dot-plots should be given, otherwise it is really hard to judge these.

We have added the percentage of cells in the density plots throughout the manuscript. It should be noted that these scatter plots are derived from the single-cell analysis of immunofluorescence microscopy images rather than FACS.

Figure 6D

would an outlier robust correlation measure such as spearman correlation also yield a similarly strong correlation between Jun and p-ERK?

Figure 6D in the original manuscript (Figure 9B now) shows the correlation between drug-induced changes in c-Jun and NGFR across the cell lines. This correlation is robust using both Pearson ($\rho = 0.86$, $P = 0.001$) and Spearman ($\rho = 0.83$, $P = 0.006$) metrics.

Figure 6D,B E,G and I

1. It is well known that JNK regulates Jun phosphorylation, in Figure 6D,B E,G and I, it seems to be assumed by the authors that JNK also chiefly regulates Jun mRNA level. Is this the case?

We did not actually assume that JNK regulates JUN mRNA levels in any part of the manuscript, although we probably injudicious with our working. We observed that vemurafenib-induced c-Jun up-regulation is associated with an increase in the phosphorylation of this transcription factor (see revised Figure 6G). As the reviewer correctly pointed out, phosphorylation of c-Jun – which is known to enhance its transcriptional activity – is regulated by JNK. Thus, we used JNK inhibitor to prevent JNK from phosphorylating c-Jun and to suppress the transcriptional activity of c-Jun. The on-target activity of JNK inhibitor JNK-IN-8 used in this study has been extensively evaluated using similar cell lines in our previous study (Fallahi-Sichani et al, 2015).

Page 16 lines 9-10 " ...in MZ7MEL and MMACSF cells, HGF or NRG1 increased the fraction of Ki-67 High cells...but did not change NGFR expression"

-> judging from figure 9C only MZ7MEL shows an increase in Ki67 high cells only for HGF whereas TNF increased NGFR High cells in MMACSF!!!

Why is the last panel in Figure C empty although some cells can be spotted in the FACS plot in Fig 9B for those cases.

Why were different concentrations of Vem used in Figure B for the two cell lines?

As mentioned above, we have decided to remove these experiments entirely, resolving these concerns.

page 7 line 16-17:" ...cell cycle genes ... in COLO858 were upregulated again by t=48h (Figures

3A, 3B)..."

-> they are not upregulated but restored to their original level

We thank the reviewer for this correction. We have revised the sentence in the manuscript.

page 7 line 22: In two out of eleven additional BRAF V600E/D melanoma cell lines tested (A375 and WM115, Figure 3C)...

-> in Figure 3C in addition to COLO858 and MMACSF only seven more cell lines are shown

We apologize for this mistake – we meant to refer to nine cell lines overall. We have made the correction in the revised manuscript.

Figures 5G/H

page 11 line 8-11 "These proteins... were present at higher levels, or were substantially up-regulated by vemurafenib in COLO858, A375 and WM115 ... relative to MMACSF and MZ7MEL "

1. pFAK is neither substantially upregulated nor higher than MMACSF, and Integrin β 1 in COLO858 and both Vem-sensitive cell lines seems not higher as MZ7MEL or much stronger induced as in MMACSF. These exceptions should be correctly pointed out and discussed
2. The more surprising it seems that FAK and SRC inhibitors can down regulate all three resistant cell lines NGFR in Fig 5 I.

1.14. We apologize for the confusion; we think that presentation of the p-FAK Western blot data in the original manuscript was biased by the way we had normalized the data. We had initially normalized p-FAK protein levels to the total FAK levels in each of the treatment conditions for each cell line (which makes sense as a relative measure of activity). Upon re-analysis, however, we noticed that total FAK levels are quite different between cell lines (as large as 4-fold) and that they also vary with vemurafenib treatment. For example, MZ7MEL cells express substantially lower levels of p-FAK relative to A375 cells, but they also express lower levels of FAK (Figure EV3B). Thus, normalizing p-FAK levels to total FAK levels obscures absolute changes in p-FAK. We now normalize p-FAK levels to HSP90 α/β and do this with TSP-1 and Integrin β 1 as well. The data are now more consistent and in-line with our text (because we will provide source data for all figures, interested readers will be able to judge the impact and significance of normalization for themselves).

1.15. Briefly, comparing quantified Western blots from the COLO858 and MMACSF cells (Figure 6F) shows that the p-FAK level (normalized to HSP90 α/β) increased by ~3.5-fold in COLO858 cells,

whereas it decreased slightly (~25%) in MMACSF cells. Integrin $\beta 1$ and TSP-1 levels (normalized to HSP90 α/β) increased in both cell lines but a larger increase was observed in the case of COLO858 cells; TSP-1 increased by ~25-fold in COLO858 cells and only ~2-fold in MMACSF cells. Integrin $\beta 1$ increased up to >5-fold in COLO858 cells but only ~2-fold in MMACSF cells.

1.16. Including additional NGFR^{High} cell lines A375 and WM115 and the NGFR^{Low} cell line MZ7MEL in this analysis also shows that A375 and WM115 cells express higher levels of TSP-1 and Integrin $\beta 1$ in comparison with MZ7MEL cells (Appendix Figure S6E). This is in agreement with the observation that A375 and WM115 cells express intrinsically high levels of NGFR under normal growth conditions. TSP-1 levels (normalized to HSP90 α/β) in A375 and WM115 cells were approximately 23- and 15-fold higher (respectively) than in MZ7MEL cells, and vemurafenib induced a further increase of this protein in A375 and WM115 cells. Integrin $\beta 1$ (normalized to HSP90 α/β) was expressed at approximately 2- and 1.5-fold higher levels in A375 and WM115 cells relative to MZ7MEL cells. Vemurafenib induced Integrin $\beta 1$ by ~2-fold in WM115 cells but reduced it by ~50% in MZ7MEL cells. Finally, p-FAK levels (normalized to HSP90 α/β) were approximately 2- and 6-fold higher in A375 and WM115 cells, respectively, relative to MZ7MEL cells. Vemurafenib further increased p-FAK in A375, but slightly reduced it in MZ7MEL. Overall, these data are compatible with the NGFR status of different melanoma cell lines, and consistent with the observation that FAK inhibitors suppress NGFR expression in all the three NGFR^{High} cell lines. At the same time we acknowledge that we only partly understand the differences among the melanoma cell lines with respect to signaling pathways and drug response. We have added text to this effect in the discussion.

Figure 7C

1. arrows point to the lower FAK but should for consistency point in the space between the two FAK inhibitors (will not weaken the point of the authors)

We have made this change in the revised manuscript.

Figure 7D

1. The encircled treatments seem to be close to the 0 of the first two PCs. Does this not mean they can not be well described by the loadings shown on the left?

2. It should further be pointed out that the red line in the first plot denotes the cumulative variance.

1.17. As mentioned in the original manuscript, we used PCA analysis of the biochemical data generated from the effects of drugs with similar or distinct nominal targets on cells as an indirect way of evaluating whether we could legitimately associate observed phenotypes (e.g. changes in NGFR expression) with on-target drug activity. However, in the revised manuscript we have used a more direct method - siRNA knockdown - to address this point (Figure 7A). The PCA analysis no longer adds significant insight and we have therefore removed it.

Nonetheless, to address the reviewer's concern, we note that in the PCA presented in the original manuscript (for the A375 cell line), PC1 captured drug-induced changes in p-ERK (with negative loading), and variation in NGFR, c-Jun and p-c-Jun (with positive loadings). Score vectors associated with different drugs projected along PC1 depending on how they affected each of these proteins. For example, the MEK inhibitor reduced p-ERK and increased c-Jun, p-c-Jun and NGFR levels and therefore had a large positive PC1 score. The JNK inhibitor inhibited c-Jun, p-c-Jun and NGFR with no effect on p-ERK, and FAK inhibitors suppressed NGFR with no effect on either p-ERK, c-Jun, or p-c-Jun. Thus, the score vector associated with the JNK inhibitor was associated with a larger negative number as compared with FAK inhibitors (that showed a PC1 score of close to zero). Importantly, the differences among the PC1 scores for each of these compounds (rather than the absolute score values) were associated with differential activity on the proteins assayed, which was the main point of PCA analysis.

Figure 9

For easier comprehension the order in Figure A, B and C of cell line and treatment should be the same in all three figures!

We have made this change in the revised figures.

Reviewer #2:

This manuscript studies the heterogeneous drug response to melanoma cells treated with RAF/MEK inhibition. The study identifies a fraction of slowly cycling NFGR-high de-differentiated cells, which can potentially develop a reversible drug-resistant phenotype. The authors also show that BET and HDAC inhibitors block the de-differentiation and restore the drug sensitivity.

While the study is certainly interest in showing subpopulation of cells having different drug responses, there are several shortcomings of this study. In many cases of gene expression analyses, there are only subtle changes and there is a lack of consistence across the various

figures in the manuscript. Secondly, the viability assays are primarily based on the short-term assays.

The study would also greatly benefit from xenograft experiments, especially in the light of the suggestion that ECM and cytokine signaling are relevant for the emergence of the slow growing drug tolerant cell populations.

As described in detail in the cover letter and response to Reviewer 1 (see points 1.1 and 1.7), we agree that the original manuscript was too dense and that the systematic analysis of drug response under different conditions (drugs, doses, cell lines, etc.) was insufficiently clear. We have extended the gene expression analysis (which in some cases shows very dramatic changes in mRNA levels – e.g. 20-fold), performed longer-duration viability assays, added xenograft experiments and eliminated data on the effects of cytokines on drug sensitivity.

Despite the deficiencies in presentation, the experimental data in our study are in fact self-consistent and also in-line with relevant literature. To help make this clear we have completely re-organized the figures and re-written the manuscript by focusing on a set of key inter-connected points derived from the data already present in the original manuscript and on data newly collected based on reviewers' comments.

2.1. With respect to the reviewer's critique about using short-term 3-day viability assays as opposed to longer term assays, we have added new data including: (i) Live single-cell imaging (Figure 1) which was performed to determine whether vemurafenib-adapted COLO858 cells continued to cycle slowly when exposed to drug for a period longer than 3-4 days, and showed that drug-adapted COLO858 cells actually continued to grow slowly in the presence of vemurafenib for at least 8 days. (ii) Time-lapse live cell analysis of cellular response to different drug combinations using an in-incubator IncuCyte system (as suggested by the reviewer). These data (Figure 8F) demonstrate continuous reductions in the number of live cells in response to co-drugging.

2.2. The newly included xenograft data were collected following 5-day treatment of mice with RAF inhibitor dabrafenib alone or in combination with BET inhibitor JQ1 (Figure 11). They complement tumor volume data collected by others on a JQ1 plus vemurafenib combination (this study was published in the middle of our xenograft studies and fortunately involves the same xenograft model as our study). By co-staining tumors from each treatment condition for NGFR and the proliferation marker Ki-67, we studied tumor heterogeneity with respect to expression or co-expression of these proteins (and compared with heterogeneity observed in patient tumor biopsies) as well as their changes following

different drug treatment conditions. Single-cell analysis of tumors showed that dabrafenib in combination with JQ1 could reduce both the fraction of NGFR^{High} and Ki-67^{High} cells by ~50% relative to tumors treated with dabrafenib only.

As noted in the cover letter, we have extensively revised the paper to remove excessive claims about the translational potential of this work and to focus it on the evidence of heterogeneous drug response and a slow-growing, drug-adapted cell populations. We agree that potential therapeutic relevance of the proposed drug combinations requires much more extensive analysis in animals and careful consideration of pharmacokinetic parameters.

Major Comments:

Are these slowly-cycling cells having incomplete MAPK inhibition? Can the authors use a FACS-based assay to zoom in to the p-ERK level in slowly growing cells?

2.3. This is an extremely important point that we also address above in point **1.5** of the response to Reviewer 1. In addition, as the reviewer suggested, we co-stained COLO858 cells for p-ERK and either of the proliferation markers p-Rb^{S807/811} or Ki-67 following 48-72 h exposure to vemurafenib. We did not see a significant difference in p-ERK levels between cells arrested in G0/G1 (p-Rb^{Low}/Ki-67^{Low} cells) or cells progressing to S/G2/M phases (p-Rb^{High}/Ki-67^{High} cells). Resistance is therefore unlikely to be a simple consequence of MAPK pathway reactivation. We have now added these data to the revised manuscript (Figures 2C, 2D, Appendix Figure S2C).

The study makes the link between NGFR expression and MAPKi-resistance. It would be interesting to investigating whether NGFR signaling is the directly connected with MAPKi-resistance. For example, over expressing NGFR and/or add the ligand, and see whether these confer resistance to MAPK inhibitors, or whether knocking-down NGFR increases the sensitivity to MAPK inhibitors?

2.4. Up-regulation of NGFR in COLO858 cells was detected approximately 48 h following vemurafenib exposure, around the time that the slowly-cycling cells began to appear. However, many of the other genes associated with neural differentiation were up-regulated as early as 24 h (Figure 4A). Because of this delay, we therefore did not initially suspect that NGFR itself could be a major cause of drug resistance. However, as the reviewer suggested, we performed RNAi knock-down experiments with NGFR and also exposed cells to NGF, a ligand for NGFR, in the presence and absence of

vemurafenib. NGFR knockdown did not enhance sensitivity of COLO858 cells to vemurafenib and exposure of cells to NGF also did not increase resistance. These data suggest neither NGFR itself nor NGF signaling are involved in drug resistance. Instead, NGFR expression appears to be a marker of a dedifferentiated cell state. We have added these data to the revised manuscript (Figures 7A and EV3C).

NGFR and Ki-67 stainings in Fig4E have very poor anti-correlation. This is not consistent with what the authors show in Fig3D.

2.5. We believe that this confusion has arisen because, in the original manuscript, specific regions of the biopsies were shown with high magnification to highlight the unusual and scattered presence of small patches of Ki-67^{High}/NGFR^{High} cells surrounded by larger NGFR^{High}/Ki-67^{Low} regions within pre-treatment tumor biopsies. We have redone our analysis using much larger portions of the tumor and now show it in a completely revised Figures 10A and 10B. This analysis shows that across the tumor there is still a strong anti-correlation between regions of high NGFR^{High} and Ki-67^{High} staining. Such an anti-correlation was also observed in xenograft tumors (Figure 11). In the original manuscript, we speculated that the presence of these small unexpected patches might imply tumor cells' switching between more or less differentiated and proliferative states. However, to avoid any confusion we have now removed this speculation and also the magnified figure panels from the manuscript. We have added quantitative, single-cell data analysis (Figure 10B).

The study identifies activation of the focal adhesion kinase (FAK) upon BRAF inhibition. However, of 5 melanoma cell lines tested, only Colo858 showed a significant activation of FAK. In general, the authors should comment on how often heterogeneous staining for NGFR is seen in primary tumors and relate this to the heterogeneity of their cell line panel in having this slow growth population emerge during drug resistance

This is an important point and we apologize for the confusion; we believe that presentation of the p-FAK Western blot data in the original manuscript was biased by the way we had normalized the data. We have redone the analysis and it is now in-line with our hypotheses. Please see detailed response in points **1.14**, **1.15** and **1.16** in the response to Reviewer 1. With respect to heterogeneous staining for NGFR, we observed NGFR^{High} regions in 4/11 of tested patient tumor biopsies. We also found NGFR^{High} cells in 2/9 cell lines that were tested in this study. However, it is notable that an additional cell line (COLO858) with minor basal NGFR levels can be induced by drug to an NGFR^{High} state. We have discussed this point in the Discussion of the revised manuscript.

In Fig 5I, NGFR is not significantly up-regulated upon BRAF inhibition, this piece of data is not consistent with the previous data, Fig3C, D

2.6. As depicted in Figure 3 of the original manuscript and Figures 4C, 4D of the revised manuscript, NGFR in COLO858 cells is up-regulated 3-7 fold depending on drug dose within the first 48 h of vemurafenib exposure. This up-regulation increases to ~25-fold by t = 72 h (the 72 h data have been added to the revised manuscript). We believe that this data has been consistent across different measurements presented in other figures (including Figure 5I of the original manuscript or Figure 7B of the revised manuscript that show NGFR changes following 48 h vemurafenib treatment in a biologically independent experiment). In the case of A375 and WM115 cells, which express very high basal levels of NGFR under normal growth conditions, vemurafenib treatment for 48 h increases NGFR by a substantial amount in absolute terms but only 25-50% in relative terms (depending on dose). This data is also consistent across different figures within the manuscript. Despite looking carefully, we have not been able to find any disagreement between NGFR protein measurements in different figures and assume that poor presentation led to the confusion. It is also noteworthy that the adaptive biology of cell lines in which NGFR is strongly induced, and those in which it is high under basal conditions, appear to be similar, as judged by co-drugging experiments (Figure 9).

In Fig 8F, the drug combination effect of HDACi/BETi and BRAFi is quite weak. In COLO858, HDACi/BETi even sort of rescue the cells from high dose of MAPKi. An exposure of 48 hours to any drug that acts through chromatin effects seems very short and consequently unlikely to be mediated by chromatin-remodeling effects. Can the authors comment on this?

2.7. We agree with the reviewer that the effects of HDAC/BET inhibitors on different melanoma cell lines is accompanied by variability at the level of molecular markers. This was not unexpected as these compounds were identified in a very simple screen involving induction of NGFR^{High} cells; their actual mechanisms of action still remains to be determined. However, we respectfully disagree with the reviewer about the strength of the effect of BET inhibitor combinations with RAF inhibitor or RAF/MEK inhibitor combination. We have now repeated cell viability measurements with additional biological replicates and confirmed that all BET inhibitors (+)-JQ1, I-BET and I-BET151 were effective in combination with vemurafenib or vemurafenib plus trametinib in enhancing cell killing consistently in the three cell lines tested (COLO858, A375 and WM115). The magnitude of BET inhibitor combination effects varied from one cell line to the next but could be as large as 10-fold reduction in cell viability.

JQ1 in particular, shows strong evidence of synergy in all lines tested and also has a combination effect in xenografts.

In the case of HDAC inhibitor belinostat, Figure 8B in the original manuscript showed substantial suppression of NGFR expression in the case of A375 and WM115 cells, but the effects were not as significant in COLO858 cells, for unknown reasons. Cell viability measurements were also consistent with this observation; belinostat was more effective in enhancing response of A375 and WM115 cells to RAF/MEK inhibition, but its inhibitory effect on COLO858 cells were independent of RAF/MEK inhibition or NGFR. Nevertheless, we had included the belinostat data in the main figure of the original manuscript.

However, to avoid confusion, and as described in response to Reviewer 1 (see point **1.12**), we have now moved HDAC data to the supplements and focus the revised Figures 8 and 9 on the strong and consistent effect of three BET inhibitors that work in all three NGFR^{High} melanoma cell lines tested by strongly suppressing both NGFR and Ki-67. As mentioned above, we have also added to the revised manuscript new data on one-week long time-lapse live cell analysis of COLO858 response to BET inhibitor JQ1 in combination with vemurafenib or vemurafenib plus trametinib using an IncuCyte system. The data clearly shows a persistent reduction in the number of live cells in response to JQ1 in combination with vemurafenib, whereas neither vemurafenib, nor JQ1 alone reduced the number of cancer cells below that of the pre-treatment (day 0) condition.

With respect to the reviewer's comment about the time-scale of the effect of epigenome-targeting compounds, we should mention that the time-scale of our data is consistent with previous studies that have looked into the impact of these compounds on cancer cells. Among these reports is a very influential paper by Sharma *et al* (Sharma et al, 2010) that is highlighted by the reviewer below; in this paper, HDAC inhibitors such as trichostatin A (TSA) cause the rapid and selective death of drug-tolerant cancer cells within 2-3 days. Studies on BET inhibitors also suggest a relatively rapid effect on cell growth/death (as early as 2-3 days) (Filippakopoulos et al, 2010). Thus we do not believe our studies with epigenome-targeting compounds differ substantially from the current state of the art.

All the cell viability data in this study are relative short-term assay (72-96 hours); authors should also provide the long-term assay such as colony formation assay to support their conclusions. Incucyte experiments for some of the seminal observations would be very helpful, especially the experiments with epigenetic modifiers.

Please see the response above (points 1.10, 2.1 and 2.7) in which we describe precisely these experiments.

The authors make no reference to the work of the group of Jeff Settleman (Cell 141, 69-80, 2010), which identifies a similar slow growing drug resistant cell population. The authors should compare the features of their cells with those identified by Sharma et al. For instance, the role of KDM5A could be studied in the melanoma model also.

2.8. We thank the reviewer for this comment – this omission was unintentional since we are of course very familiar with the Settleman’s paper. We have now revised our manuscript and cited the work by Sharma *et al* in multiple locations; we review its findings in detail in the revised discussion. As we now discuss, despite the general similarity with respect to the emergence of reversibly drug-insensitive cell subpopulations in both studies, there are considerable differences between them: The drug-tolerant cancer cell subpopulation (drug-tolerant persisters; DTPs) discovered by Sharma *et al* occur at a frequency of ~1% (or less) of the original cell population following exposure of drug-sensitive cancer cells to very high concentrations of drug (~100-fold greater than the IC₅₀ value). DTPs were found to be largely quiescent, but approximately 20% of them eventually resume normal proliferation in the long-term presence of drug, yielding clones of reversibly drug-insensitive cells. However, time-lapse live single-cell analysis of drug response in COLO858 cells shows that the slowly-cycling drug-adapted phenotype can emerge relatively rapidly at concentrations near IC₅₀. In addition, the slowly-cycling NGFR^{High} cell state can be induced in a larger fraction of cells. For example, >90% of COLO858 cells alive at the end of ~3 days of exposure to 1 μM vemurafenib express high levels of NGFR and ~50% of these cells cycle with a doubling time of ~3 days (COLO858 cells naturally double 3-4 fold faster).

Overall, combining our results with data from Sharma *et al* (Sharma et al, 2010) and other studies (Cohen et al, 2008; Flusberg et al, 2013; Gascoigne & Taylor, 2008; Ravindran Menon et al, 2015; Spencer et al, 2009), we speculate that individual tumor cells within a notionally genetically homogenous tumor cell population can exhibit naturally different levels of drug sensitivity (or drug tolerance). Such differences are epigenetically regulated and transiently stable and can be explained by natural fluctuations in the amount or activity of different proteins across individual cells. In addition to natural fluctuations, drug-induced adaptive responses can also influence and further complicate the cell-to-cell variability in drug sensitivity. The differences that we observe in epigenetically regulated drug resistance phenotypes (e.g. DTPs from Sharma *et al* versus slowly-cycling NGFR^{High} cells) can be explained by the differences in the timing and strength of selective pressure imposed by different drug treatment conditions. Whereas an acutely high drug concentration may lead to selection of a small

percentage of intrinsically highly-insensitive cell subpopulation (i.e. DTPs), a low or moderate (sub-killing) dose of drug can give a larger number of cells sufficient time for adaptation and acquisition of a less drug-sensitive state (such as the slowly-cycling NGFR^{High} state) that diminishes drug maximal effect. Regardless, both of these drug resistance states are transiently stable and therefore reversible with time upon drug removal. We have now discussed these points in the Discussion of the revised manuscript.

Reviewer #3:

Fallahi-Sichani et al used multiplexed signaling, single cell, and transcriptomic analysis to analyze early adaptive events (first 48 hrs) that may impact the emergence of a BRAF inhibitor inducible slow-cycling sub-population of de-differentiated melanoma cells. They concluded that a FAK-SRC-cJUN signaling axis induced by BRAF or BRAF/MEK inhibition promotes de-differentiation as marked by CD271/NGFR induction and drug-tolerant slow-cycling melanoma cells with mesenchymal features. The authors also concluded that inhibitors of FAK, SRC and BET bromodomain selectively target this therapy-induced slow-cycling subpopulation to enhance the Emax of killing.

The descriptive aspects of this study have been reported extensively in the literature.

- cJUN has been implicated as a critical adaptive response to BRAF inhibitors. The same authors (Molecular Systems Biology 2015) have shown that BRAF inhibition induced c-JUN and JNK and BRAF inhibitors enhanced the Emax of melanoma cell killing. Delmas et al (Oncotarget 2015) showed that BRAF inhibition transcriptionally and adaptively up-regulated cJUN, leading to RHOB and AKT induction. Ramsdale et al (Science Signaling 2015) reported that both intrinsic and adaptive BRAF inhibitor resistance are driven by increased expression of cJUN, which induces mesenchymal de-differentiation.

- That BRAF inhibitor can induce a reversible and slow-cycling drug tolerant subpopulation marked by loss of differentiation, induction of CD271, and histone modifications has been recently reported (Menon et al, Oncogene 2014; Zubrilov et al, Cancer Letters 2015).

- Integrin beta1/FAK/SRC have been shown to participate in BRAF inhibitor resistance (Hirata et al, Cancer Cell 2015; Girotti et al, Cancer Discovery 2013).

- The compound JQ1 has been shown to be synergistic with BRAFi or combined BRAFi and MEKi in growth-inhibiting BRAF mutant melanoma cells. The proposed mechanism was inhibition of bromodomain proteins such as BRD4 which are critical for MYC mRNA expression

(Korkut et al, eLIFE 2015).

3.1. We thank the reviewer for highlighting the potential connections that exist between our work and other studies on drug resistance in melanoma. We believe that we have cited all of these antecedents and agree that they set the stage for further analysis of adaptive drug response. Nonetheless, we believe that our manuscript presents a meritorious contribution to the literature on adaptive resistance in melanoma cells exposed to RAF/MEK-targeted therapy:

The major novelty of this work lies in its quantitative and systematic approach to understanding early drug adaptation in *BRAF*^{V600E/D} melanoma cells by using a combination of time-lapse analysis of live cells by tracking hundreds of single cells for multiple days as well as other forms of single-cell measurement and analysis that were performed on FACS-sorted cells or fixed cells derived from cultured or xenograft tumors (see the revised manuscript for new data). This provides a direct evidence on drug adaptation in individual cells and shows how heterogeneity contributes to emergence of sub-populations with differential sensitivity to RAF/MEK inhibitors. Our work connects many of the studies listed by the reviewer but, to the best of our knowledge, is substantially different in approach and conclusions.

As the reviewer points out, the emergence of reversible drug-tolerant subpopulations of melanoma cells marked by loss of differentiation and induction of NGFR has very recently been reported in a nice paper by Ravindran Menon *et al* (Ravindran Menon et al, 2015). However, this study was performed using fixed time-point and population-average measurements and primarily involved comparing drug-naïve cells with cells exposed to vemurafenib for times at which it is difficult to distinguish between adaptive resistance and selection of intrinsically drug-tolerant cells. The connection between drug adaptation and heterogeneity in the original culture could only be inferred indirectly. Moreover, the potential for NGFR^{High} cells to have a lower rate of proliferation could only be inferred indirectly by Ravindran Menon *et al* from pathway analysis of microarray data showing down-regulation of DNA replication pathways. In our studies, direct measurement of individual living cells connects drug response, cell cycle time and differentiation state within a heterogeneous population of cells.

Direct measurement of cell status is not simply a technical improvement over past practice; it yields fundamentally different data. Most notably, Ravindran Menon *et al* find that drug-adapted cells have highly active MEK/ERK signaling. In contrast, our data clearly shows that NGFR is induced, and drug adaptation occurs in cells in which MAPK signaling remains inhibited (as assayed by p-ERK levels at the single-cell level). Moreover, drugs we identify as blocking emergence of these cells do not reduce

p-ERK levels further. Finally, when MAPK signaling is efficiently inhibited using a combination of trametinib and vemurafenib (or dabrafenib), we find that the magnitude of NGFR induction increases rather than decreases (and co-drugging to block adaptation is more efficient). Whether drug adaptation does or does not involve reactivation of the MAPK pathway (the target of BRAF therapy) is a fundamental distinction in the field. Thus, our work does not represent a simple refinement on the studies of Ravindran Menon *et al*; it heads in a different direction. We explicitly discuss this in our revised manuscript.

As the reviewer pointed out, our study links our major findings from single-cell measurements and analyses to biochemical mechanisms of adaptive resistance that have been reported recently by us and other groups (particularly with respect to JNK activation). We strongly believe that aspects of our paper that link together apparently unrelated data in the literature represent a valuable contribution rather than a cause for concern. One of the aspirations (admittedly partially realized in most cases) is to provide a more comprehensive and contextualized understanding of complex biological phenomena. Nonetheless, to address the reviewer's concerns we now include a discussion of nearly all of the papers mentioned above.

The reviewer points our data showing that RAF inhibition induces the JNK/c-Jun pathway (Fallahi-Sichani *et al*, 2015). Understanding of the role of this pathway increased following subsequent publications that are highlighted by the reviewer, including Ramsdale *et al* (Ramsdale *et al*, 2015) and Delmas *et al* (Delmas *et al*, 2015). The current manuscript differs from all of these papers in linking the JNK/c-Jun mediated drug-induced adaptive response to the emergence of less-differentiated NGFR^{High} vemurafenib-tolerant subpopulation of cells. The NGFR^{High} state of melanoma cells has its own extensive literature, much of it involving histopathology and sequencing of human tumors. To our knowledge, our manuscript is the first to link these phenomena to each other and to show that they are vulnerable to pharmacological inhibition by a variety of kinase inhibitors and epigenome-modifying drugs.

Another example of connections between our work and previous reports, also pointed out by the reviewer, involve Integrin β 1/FAK signaling, which has been shown by Hirata *et al* (Hirata *et al*, 2015) to be involved in resistance of melanoma cells to RAF inhibitors. This study, however, focused on the role of melanoma-associated fibroblasts that confers resistance in melanoma cells in a paracrine fashion via reactivation of ERK signaling. Our results complement these findings and we think it is very interesting that a cell-autonomous adaptation also involving FAK signaling leads to emergence of resistance that is independent of stromal cells and ERK signaling, but induces the NGFR^{High} drug-tolerant state.

The mechanistic implications of BRAF combinatorial targets (FAK-SRC-cJUN and epigenetic targets) were under-developed. It was not clear how much SRC and cJUN activation was dependent on FAK activation. Do AP-1 dependent transcription and cell adhesion related survival constitute a feed-forward signaling loop? How critical was FAK in this loop? Knockdown/genetic approaches and clear demonstration of the targets of inhibitors (dose-response) used were missing. How significant was the effect of FAK/SRC inhibitors on NGFR levels (Figure 5I) responsible for the cell viability effects (Figure 7A)? Since FAK/SRC inhibitor co-treatment with BRAF inhibition in A375 and WM115 still left the cells with plenty of NGFR protein, why would co-treatment enhance the E_{max} ? Most importantly, it appeared that the action of bromodomain and HDAC inhibitors such as JQ1 had little to do with the cJUN-NGFR link. For instance, in the prototypic line used in this study (Colo858), the HDAC inhibitor belinostat induced NGFR without BRAF inhibition (Figure S8B) and, in the presence of I-BET and JQ1, BRAF inhibitor still induced cJUN (Figure S8D).

In response to these concerns, we have performed a large number of additional studies and have substantially revised the manuscript.

3.2. With respect to the on-target activity of kinase inhibitors, and the involvement of JNK and FAK kinases in particular, we have performed siRNA knockdown experiments. These confirm that FAK and JNK/c-Jun signaling induce the NGFR^{High} state and drug resistance. In particular, we exposed COLO858 cells to vemurafenib in combination with either *JUN* or *PTK2* (FAK) gene depletion by siRNA for 72 h. Both *JUN* and *PTK2* knockdown reduced vemurafenib-induced NGFR up-regulation by ~70% and reduced cell viability by 2-fold in comparison with cells transfected with control siRNA (Figure 7A). In contrast, we find no evidence from NGFR knockdown or NGF treatment that NGFR receptor is directly involved in drug resistance; instead, it appears to be a marker of the adaptive state. With respect to Src inhibitors such as dasatinib and saracatinib we hope that the reviewer agrees that knockdown experiments are not feasible – these drugs target multiple Src family kinases, and this is part of their mechanism of action.

3.3. With respect to the reviewer's concern about the significance of the effect of JNK, FAK and Src kinase inhibitors on NGFR levels in different cell lines and its relationship with enhancing E_{max} , we should mention that different cell lines expressed very different levels of NGFR both under normal growth conditions or following vemurafenib exposure. For example, in WM115 cells treated with 1 μ M vemurafenib for 48 h, NGFR was expressed at ~8-fold higher levels than in COLO858 cells under the

same treatment conditions. Nevertheless, the 48 h effect of the inhibitors of JNK, FAK and Src (and also BET inhibitors) on NGFR levels were highly comparable between the cell lines A375, COLO858 and WM115; this is clear when we look at the z-score normalized effect of each of these compounds on NGFR in each individual cell line. Z-score normalization emphasizes the variance in NGFR levels under different drug treatment conditions, rather than the differences in absolute levels of this protein between different cell lines (see Figures 7C, 8C, 9E, 9F). Furthermore, NGFR itself is not essential for drug resistance, as NGFR knockdown did not enhance sensitivity of melanoma cells to vemurafenib (see Figure 7A). Altogether, these data explain why co-treatment with either of the inhibitors of JNK, FAK or Src and RAF/MEK kinase inhibitors is similarly effective in enhancing drug maximal effect among the three cell lines that express different absolute levels of NGFR following 48 h combination treatments.

3.4. We agree with the reviewer that the effects of HDAC/BET inhibitors on different melanoma cell lines were accompanied by some variability at the level of molecular markers (for example, their effect on c-Jun). We explicitly discuss this variability in the revised manuscript. In the case of HDAC inhibitor belinostat, Figure 8B in the original manuscript showed that NGFR suppression by this compound was substantial in the case of A375 and WM115 cells, but not as significant in the case of COLO858 cells for an unknown reason. Cell viability measurements were also consistent with this observation; belinostat was more effective in enhancing response of A375 and WM115 cells to RAF/MEK inhibition, but its inhibitory effect on COLO858 cells were independent of RAF/MEK inhibition or NGFR.

At the same time, we think that analysis of HDAC inhibitors represents a distraction and we have moved all of this data to the supplement (please see points **1.12** and **2.7** in response to Reviewers 1, 2). In the current manuscript we highlight new studies showing that three BET inhibitors (+)-JQ1, I-BET and I-BET151 are consistently effective in combination with vemurafenib or vemurafenib plus trametinib in enhancing cell killing in the three NGFR^{High} cell lines tested (COLO858, A375 and WM115). The magnitude of BET inhibitor combination effects varied from one cell line to the next but could be as large as 10-fold reduction in cell viability. We have also added to the revised manuscript new data on one-week long time-lapse live cell analysis of COLO858 response to BET inhibitor JQ1 in combination with vemurafenib or vemurafenib plus trametinib using IncuCyte (Figure 8F). The data clearly shows a persistent reduction in the number of live cells in response to JQ1 in combination with vemurafenib, whereas neither vemurafenib, nor JQ1 alone were able to reduce the number of cancer cells below that of the pre-treatment (day 0) condition.

Also, the authors tried to show us that the small molecule inhibitors used against FAK-SRC-JNK were hitting the nominal targets in Figure 7D using A375. If one looks at Figure 5G and 5H, in

A375, BRAF inhibitor did not induce p-FAK, alpha-TSP-1 or integrin beta1. In the cell line where the BRAF inhibitor marginally induced these levels (Colo858), the PCA looked extremely weak (Figure S7C).

With respect to the reviewer's concern about protein levels of TSP-1, Integrin β 1 and p-FAK in A375 cells, quantified Western blot data clearly show that A375 cells expressed the highest levels of TSP-1 among all the cell lines (i.e. 20-30 fold higher than NGFR^{Low} cell lines MMACSF and MZ7MEL) and RAF inhibitor treatment increased TSP-1 levels by >50%. This is completely consistent with a high basal level of NGFR protein and a ~50% NGFR up-regulation following 48 h vemurafenib treatment. Integrin β 1 in A375 cells were also expressed at 3-6 fold higher levels than in NGFR^{Low} cell lines MMACSF and MZ7MEL.

With respect to the p-FAK Western blots, please see a detailed response above (points **1.14**, **1.15** and **1.16**).

With respect to the reviewer's concern about PCA analysis for COLO858, we have now removed the analysis and replaced it with more direct evidence from siRNA studies.

We also respectfully believe that the reviewer has misinterpreted the PCA data. As mentioned in the original manuscript, we used PCA analysis of the biochemical data generated from the effects of drugs with similar or distinct nominal targets on cells as an indirect way of evaluating whether we could legitimately associate observed phenotypes (e.g. changes in NGFR expression) with on-target drug activity. In the case of A375 cell lines, PC1 very clearly captured drug-induced changes in p-ERK (with negative loading), and variation in NGFR, c-Jun and p-c-Jun (with positive loadings). PC2, on the other hand, captured drug-induced changes in proliferation markers Ki-67 and p-Rb (with positive loadings), and variation in p-S6 (with negative loading). This clear separation of the molecular signals within the PC space made it very easy to distinguish different drug effects based on the score vectors associated with each drug along the same PC space. In the case of COLO858 cells, however, the orientation of loadings vectors associated with the measured molecular signals along the PC space was different from that in A375 cells. For example, drug-induced changes in Ki-67 and p-Rb were captured by both PC1 and PC2. Nevertheless, analysis of PC score vectors associated with each of the drug effects (considering the molecular signals associated with each PC) shows a clear separation between the effect of JNK, FAK and Src inhibitors versus inhibitors of MEK, PI3K, mTOR or CDK4/6. In summary, we do not agree that the PCA for COLO858 cells were weak, but only that they needed to be interpreted with respect to loadings vectors that orientated the signaling axes of the PC space in a way that was different from A375 cells.

In the light of new RNAi studies, we do not believe that PCA analysis adds significantly to the message of the paper beyond what has been presented in a much simpler way.

Beyond the transcriptomic analysis for the two cell lines Colo858 and MMACSF, additional analysis of publicly available RNASeq data was performed. However, the results shown were selective or did not optimally highlight the significance of the Colo858 analysis. For instance, are the authors postulating that the transcriptomic alterations induced by BRAF inhibition in Colo858 (representative of adaptive resistance with an inducible slow-cycling population) are indeed happening within weeks 1 or 2 on treatment in patient-derived biopsies. If this were the case, then why did the GSEA data in Figure 5E only showed enrichment of one process. Did other processes/gene sets (shown in Figure 3B) simply did not meet the FDR cutoff? It is understandable that the authors could not perform correlation with three pairs of RNASeq from pre and post resistance tumor tissues (Figure 5F). However, what was the rationale for the selection of the particular genes' expression levels? The authors should show the genes such as those in Figure 5B and 5C or others from Figure 3 to support their claim that this in vitro BRAF inhibitor inducible subpopulation are present and detectable in acquired BRAF inhibitor resistance in patient-derived tissues. Two larger RNASeq data sets of this kind are also available now in the literature if the authors want to investigate this further.

We thank the reviewer for pointing us to additional RNA-Seq datasets; we have now included publically available data from more patients in the revised manuscript (Figures 10D, 10E).

At the same time we respectfully disagree with the reviewer's comment that our transcriptomic analyses on patient data were selective. In the original manuscript, we showed therapy-induced changes in NGFR, MITF and a set of genes associated with extracellular matrix proteins and cell adhesion molecules, but this was based on an unbiased gene set enrichment analysis performed on the RNA-seq data from three melanoma patients with acquired resistance to RAF/MEK kinase inhibitors (Sun et al, 2014). A complete list of results from this analysis was included in the supplements (Table S3 in the original manuscript) –apparently missed by the reviewer – that identified ECM-receptor interactions and cell/focal adhesion among the top enriched biological processes and pathways in post-resistance NGFR^{High} tumors. This list also included other processes and pathways that were identified from analysis of COLO858 and MMACSF cells, including regulation of cell motion, axon guidance, regulation of actin cytoskeleton and others that highlight the significance of our cell line analysis.

We have repeated our analysis after combining two RNA-seq datasets, including total 21 patients with pre-treatment and post-resistance tumor biopsies (Figure 10D in the revised manuscript). As described

in the revised manuscript, this new analysis identified an increase in NGFR levels in ~62% of post-resistance biopsies in comparison with pre-treatment biopsies from the same patients. An unbiased gene set enrichment analysis on these tumors identified ECM-receptor interactions and focal adhesion among the top enriched pathways associated with acquisition of the NGFR^{High} state (Figure 10E and Datasets EV3 and EV4).

How quantitative and linear are the signals in the authors' use of immunofluorescence? For example, the authors showed in Figure 3C that higher graded levels were readily detectable (A375, Colo858, WM115) but lower levels may not be (see the remainder six cell lines). The authors should probe NGFR levels and BRAF inhibitor effects by Western blots as well.

Immunofluorescence staining followed by image analysis is routinely used by our group and others as a reliable and validated method for protein measurement and quantification (Loo et al, 2009; Millard et al, 2011). We use this method because, in contrast to cell population-average measurements such as Western blots, it has the advantage that it can reveal the extent of cell-to-cell variability in the amount of proteins and cross-correlation between them (e.g. Ki-67 vs. NGFR, or p-ERK vs. p-Rb) at a single-cell level. Nevertheless, in response to the reviewer's concern we have confirmed the correlation between population-average NGFR levels measured by immunofluorescence staining and Western blots in five cell lines before and after treatment with vemurafenib (see Appendix Figures S6A-S6C).

The authors should include data on cell lines treated with JQ-1 by itself in Figure 8G. As seen in Figure 8F, JQ-1 by itself can have negative effects on melanoma cell viability. This would suggest that the effects of JQ-1 in the presence of BRAF inhibition was not specific to the induction of alternate growth and differentiation state.

This is an important point that we have addressed with new experiments. Live-cell tracking of cells treated with JQ1 showed that this compound at a dose of 0.32 μ M induced a dramatic increase in tumor cell killing (>95%) when combined with 1 μ M vemurafenib. However JQ1 alone at a similar dose had minimal cytotoxic effect (indistinguishable from DMSO) on COLO858 cells (see Figures EV5, 8F and Movie EV2). The impact of JQ1 on reducing cell viability relative to DMSO-treated cells arises from its effect on proliferation as opposed to apoptosis. We have revised the manuscript to explicitly highlight this point.

Data presented in Figure 9 are too preliminary and disconnected to the rest of the manuscript to warrant placing this at the end. Are the authors claiming that the 10% increase in viability with TNF stimulation during BRAF inhibitor treatment in the one cell line MMACSF is the result of

NGFR up-expression? By the same token, if the authors remove or sort out the NGFR subpopulation in A375, would the remainder population be more sensitive to BRAF inhibition?

We agree with the reviewer with respect to the TNF data and have therefore removed this from the revised manuscript per reviewer's suggestion (Figure 9 of the original manuscript).

With respect to the reviewer's comment about sensitivity of NGFR-sorted cells, we believe that experiment sought by the reviewer is to test drug sensitivity in NGFR^{High} and NGFR^{Low} cells following vemurafenib treatment. We have performed precisely this experiment on COLO858 cells as described above (see points 1.3, 1.4 and 1.6). We compare directly the sensitivity of freshly sorted NGFR^{High} and NGFR^{Low} cell subpopulations to ~3-day vemurafenib treatment by measuring vemurafenib-induced growth rate (GR) inhibition in each of the subpopulations independently (Figure 5). Our analysis shows that NGFR^{High} COLO858 cells are more resistant to RAF inhibitor treatment than NGFR^{Low} cells.

On a broader level, while it seems important to achieve a greater Emax, what is the biologic significance of this slow-cycling subpopulation induced by BRAF inhibition. Does it merely serve as niche for expansion of truly drug-resistant clones or does it serve as the precursor to at least one form of acquired resistance? Does this NGFR high population give rise to the MITF low population that has been shown to resume proliferation in the presence of BRAF inhibition (Konieczkowsky et al Cancer Discovery 2014; Muller et al, Nature Communication 2014; Riesenbergs et al, Nature Communication, 2015)?

These are very important points that remain matters of speculation across the entire field of cancer biology. As the reviewer points out, eliminating slowly-cycling NGFR^{High} drug-resistant cells can substantially increase maximal effect of RAF/MEK kinase inhibitors. This is clear from conventional dose-response analysis of different drug combinations by using 3-day viability assays. We highlight in this manuscript that even cell lines such as COLO858 which are normally classified as "sensitive" to RAF inhibitors such as vemurafenib (with IC₅₀ < 0.1 μM and significant drug-induced cell death), also contain heterogeneous subpopulations of drug-resistant cells (such as slowly-cycling NGFR^{High} cells) that can quickly decrease drug effectiveness. This is most evident from time-lapse live cell and single-cell analysis of drug response in these cells; one week of treatment with vemurafenib at 1 μM (a concentration that is 10-fold higher than vemurafenib IC₅₀ for COLO858 cells) was not able to reduce the number of live COLO858 cells below that of the pre-treatment (day 0) condition. Therefore, our findings add to a growing body of research suggesting that tumor cells can reversibly undergo dynamic changes that create subpopulations of cells with different proliferative potentials and apoptotic sensitivity to drug and that such changes can be drug-induced. We speculate that such cells can be

both a source of residual disease and a precursor for acquired resistance by providing a pool for further genetic or epigenetic changes that eventually lead to tumor relapse. This is a great point for future studies, as discussed in the closing paragraphs of our discussion.

With respect to the connection between NGFR^{High} and MITF^{Low} cancer cell populations, we have studied the connection using cell lines and gene expression data from pre-treatment and post-therapy tumor biopsies. We found a significant and negative correlation between MITF and NGFR levels in vemurafenib-naïve tumors analyzed from TCGA. We also saw a similar correlation at the single-cell level in vemurafenib-naïve biopsies. However, only about half of the post-resistance tumor biopsies that exhibited an increase in NGFR levels relative to pre-treatment biopsies showed a decrease in MITF levels. Furthermore, the switch of melanoma cells to a slowly-cycling NGFR^{High} phenotype was not associated with a reduction in MITF expression in our studies (at least in COLO858 cells). This suggests that at least in drug-induced NGFR^{High} cells, the loss of MITF might not be the major cause of resistance.

We have addressed all of these points in the revised manuscript.

Reviewer #4:

Vemurafenib-treated cells were evaluated for cycling characteristic and for patterns of drug sensitivity. Vemurafenib induced 20% of COLO858 cells to enter a slow cycling state that could be reversed with drug washout. This subset of cells was resistant to apoptosis but did not reactivate MAPK signaling. Transcriptional profiling of vemurafenib-induced resistant cells identified a number of markers characteristic of less-differentiated neural crest cells, including p75. And, in populations, p75 expression marked slow cycling cells. Some human melanoma specimens were mosaic for p75 expression, which was complementary to MITF and Ki67. Transcription profiling of vemurafenib-treated responsive and resistant cell lines, and analysis of patient samples and cell lines, identified extracellular matrix and focal adhesion processes as candidates. Inhibitors targeting JNK, SRC, and FAK were especially effective in combination with vemurafenib on cell lines with tonic or vemurafenib inducible NGFR. A subset of epigenetic inhibitors downregulated NGFR expression, and these inhibitors combined for greater effect with vemurafenib. In combination with vemurafenib, FAK inhibitor and JQ1, each suppressed the slow-cycling population and increased the fraction of apoptotic cells.

Phenotypic interconversion is an important and common mechanism for cancer drug resistance. The manuscript established the means by which some melanoma cell lines develop

resistance, and also addresses the important practical issue of how to combine targeted therapies to attack p75-expressing melanoma cultures. The use of transcriptional evaluation combined with selective drug screening is reasonable and includes considerable computational analysis appropriate for this journal. The wet bench work is technically excellent, and computational analyses are used appropriately.

Some biological issues are left on the table:

The role of p75 itself is not assessed; is it simply a biomarker for this population, or does it have a signaling or other function?

This is an important question that we now resolve with RNAi and NGF treatment experiments, as described above in response to Reviewer 2 (point 2.4).

Immunoblotting supports the importance of ECM/FAK/SRC proteins, but the exclusive use of pharmacologic agents (with off-target activities) rather than genetic interventions to evaluate integrin/FAK/Src signaling leaves some uncertainty about mechanism.

We have performed siRNA knockdown experiments to address the reviewer's concern. For more details, please see the response to Reviewer 3 (point 3.2).

The ability of any of the combinations to control tumor growth without excessive toxicity has not been evaluated. Ideally, effect of one of an integrin/FAK/Src inhibitor combination would be tested on p75-inducible and non-inducible cell lines in a xenograft mouse model.

As mentioned in response to Reviewer 2, we have performed new xenograft experiments on dabrafenib alone or in combination with BET inhibitor JQ1 (Figure 11). We have also toned down many of the translational claims in the paper (see cover letter) and agree with the reviewer that a substantial body of additional work in animals is required to determine whether or not integrin/FAK/Src inhibitors might be effective in combination with MAPK therapy. Because of the favorable dosing and orthogonal toxicity, the xenografts reported in the new manuscript focus on JQ1. Please see more details in the response to Reviewer 2 (point 2.2).

Other questions.

Is there any information about how slow cycling is maintained?

We have performed extended live single-cell analysis to determine whether vemurafenib-adapted COLO858 cells continue to cycle in the presence of drug and whether they continue to cycle with a slow rate. Among the population of live cells tracked individually between days 4 and 8, approximately

half of the cells did not divide at all and the other half divided only once with minimal estimated doubling time of 48-96 h (Figure 1G). This ratio of non-cycling versus cycling cells (approximately 1:1) and their doubling time are comparable with cells tracked within the first 4 days of treatment, suggesting that drug-adapted COLO858 cells were not transiently arrested but actually continued to grow slowly in the presence of vemurafenib.

The term "dedifferentiation", seems overly strong without more evidence on the phenotypes of mature and "dedifferentiated" cells, and their relationship to normal neural crest lineage compartments.

We understand the reviewer's concerns and were ourselves uncertain how to address the issue of nomenclature. We use the term "de-differentiated" or "less-differentiated" melanoma cells based on general use of these terms in the literature and gene set enrichment analysis of the cell lines and patient tumor biopsies. "De-differentiated" is commonly used in the field for melanoma cells that express less of the melanocyte differentiation regulator MITF and pigmentation genes, and more neural crest markers such as NGFR (Landsberg et al, 2012;Lazova et al, 2010;Ravindran Menon et al, 2015;Reed et al, 1999;Riesenberg et al, 2015). We could not find a better term that could simply describe our observation; "stem-cell" like seemed to us a much worse term, although it too is commonly used.

References

Cohen AA, Geva-Zatorsky N, Eden E, Frenkel-Morgenstern M, Issaeva I, Sigal A, Milo R, Cohen-Saidon C, Liron Y, Kam Z, Cohen L, Danon T, Perzov N, Alon U (2008) Dynamic proteomics of individual cancer cells in response to a drug. *Science* **322**:1511-1516.

Delmas A, Cherier J, Pohorecka M, Medale-Giamarchi C, Meyer N, Casanova A, Sordet O, Lamant L, Savina A, Pradines A, Favre G (2015) The c-Jun/RHOB/AKT pathway confers resistance of BRAF-mutant melanoma cells to MAPK inhibitors. *Oncotarget* **6**:15250-15264.

Fallahi-Sichani M, Moerke NJ, Niepel M, Zhang T, Gray NS, Sorger PK (2015) Systematic analysis of BRAF(V600E) melanomas reveals a role for JNK/c-Jun pathway in adaptive resistance to drug-induced apoptosis. *Mol Syst Biol* **11**:797.

Filippakopoulos P, Qi J, Picaud S, Shen Y, Smith WB, Fedorov O, Morse EM, Keates T, Hickman TT, Felletar I, Philpott M, Munro S, McKeown MR, Wang Y, Christie AL, West N, Cameron MJ, Schwartz B, Heightman TD, La Thangue N, French CA, Wiest O, Kung AL, Knapp S, Bradner JE (2010) Selective inhibition of BET bromodomains. *Nature* **468**:1067-1073.

Flusberg DA, Roux J, Spencer SL, Sorger PK (2013) Cells surviving fractional killing by TRAIL exhibit transient but sustainable resistance and inflammatory phenotypes. *Mol Biol Cell* **24**:2186-2200.

Gascoigne KE, Taylor SS (2008) Cancer cells display profound intra- and interline variation following

prolonged exposure to antimetabolic drugs. *Cancer Cell* **14**:111-122.

Hafner M, Niepel M, Chung M, Sorger PK (2016) Growth rate inhibition metrics correct for confounders in measuring sensitivity to cancer drugs. *Nat Methods* **13**:521-527.

Hirata E, Girotti MR, Viros A, Hooper S, Spencer-Dene B, Matsuda M, Larkin J, Marais R, Sahai E (2015) Intravital imaging reveals how BRAF inhibition generates drug-tolerant microenvironments with high integrin beta1/FAK signaling. *Cancer Cell* **27**:574-588.

Landsberg J, Kohlmeyer J, Renn M, Bald T, Rogava M, Cron M, Fatho M, Lennerz V, Wolfel T, Holzel M, Tuting T (2012) Melanomas resist T-cell therapy through inflammation-induced reversible dedifferentiation. *Nature* **490**:412-416.

Lazova R, Tantcheva-Poor I, Sigal AC (2010) P75 nerve growth factor receptor staining is superior to S100 in identifying spindle cell and desmoplastic melanoma. *J Am Acad Dermatol* **63**:852-858.

Loo LH, Lin HJ, Steininger RJ, Wang Y, Wu LF, Altschuler SJ (2009) An approach for extensively profiling the molecular states of cellular subpopulations. *Nat Methods* **6**:759-765.

Millard BL, Niepel M, Menden MP, Muhlich JL, Sorger PK (2011) Adaptive informatics for multifactorial and high-content biological data. *Nat Methods* **8**:487-493.

Ramsdale R, Jorissen RN, Li FZ, Al-Obaidi S, Ward T, Sheppard KE, Bukczynska PE, Young RJ, Boyle SE, Shackleton M, Bollag G, Long GV, Tulchinsky E, Rizos H, Pearson RB, McArthur GA, Dhillon AS, Ferraro PT (2015) The transcription cofactor c-JUN mediates phenotype switching and BRAF inhibitor resistance in melanoma. *Sci Signal* **8**:ra82.

Ravindran Menon D, Das S, Krepler C, Vultur A, Rinner B, Schauer S, Kashofer K, Wagner K, Zhang G, Bonyadi Rad E, Haass NK, Soyer HP, Gabrielli B, Somasundaram R, Hoefler G, Herlyn M, Schaidler H (2015) A stress-induced early innate response causes multidrug tolerance in melanoma. *Oncogene* **34**:4448-4459.

Reed JA, Finnerty B, Albino AP (1999) Divergent cellular differentiation pathways during the invasive stage of cutaneous malignant melanoma progression. *Am J Pathol* **155**:549-555.

Riesenberg S, Groetchen A, Siddaway R, Bald T, Reinhardt J, Smorra D, Kohlmeyer J, Renn M, Phung B, Aymans P, Schmidt T, Hornung V, Davidson I, Goding CR, Jonsson G, Landsberg J, Tuting T, Holzel M (2015) MITF and c-Jun antagonism interconnects melanoma dedifferentiation with pro-inflammatory cytokine responsiveness and myeloid cell recruitment. *Nat Commun* **6**:8755.

Sharma SV, Lee DY, Li B, Quinlan MP, Takahashi F, Maheswaran S, McDermott U, Azizian N, Zou L, Fischbach MA, Wong KK, Brandstetter K, Wittner B, Ramaswamy S, Classon M, Settleman J (2010) A chromatin-mediated reversible drug-tolerant state in cancer cell subpopulations. *Cell* **141**:69-80.

Spencer SL, Gaudet S, Albeck JG, Burke JM, Sorger PK (2009) Non-genetic origins of cell-to-cell variability in TRAIL-induced apoptosis. *Nature* **459**:428-432.

Sun C, Wang L, Huang S, Heynen GJ, Prahallad A, Robert C, Haanen J, Blank C, Wesseling J, Willems SM, Zecchin D, Hobor S, Bajpe PK, Liefink C, Mateus C, Vagner S, Grenrum W, Hofland I, Schlicker A, Wessels LF, Beijersbergen RL, Bardelli A, Di Nicolantonio F, Eggermont AM, Bernards R (2014) Reversible and adaptive resistance to BRAF(V600E) inhibition in melanoma. *Nature* **508**:118-122.

I am pleased to inform you that your paper has been accepted for publication in *Molecular Systems Biology*.

NOTE:

- we would be grateful if you could send us the accession number for the RNA-Seq data. If possible, a stable resolvable link to the specific location where the single-cell data is accessible in LINCS would also be useful.

REFEREE REPORT

Reviewer #1:

The authors restructured their manuscript and provided extensive new data that addressed all my major concerns.

Corresponding Author Name: Peter K. Sorger, Mohammad Fallahi-Sichani

Manuscript Number: MSB-16-6796